# Network-based elucidation of colon cancer drug resistance mechanisms by phosphoproteomic time-series analysis

George Rosenberger [1,13], Wenxue Li [2,13], Mikko Turunen[1,13], Jing He[1,12,13], Prem S. Subramaniam[1], Sergey Pampou[1,3], Aaron T. Griffin [1,4], Charles Karan [1,3], Patrick Kerwin[1], Diana Murray[1], Barry Honig[1,5,6,7,8], Yansheng Liu [2,9] ✉ & Andrea Califano [1,5,6,8,10,11] ✉

Aberrant signaling pathway activity is a hallmark of tumorigenesis and progression, which has guided targeted inhibitor design for over 30 years. Yet, adaptive resistance mechanisms, induced by rapid, context-specific signaling network rewiring, continue to challenge therapeutic efficacy. Leveraging progress in proteomic technologies and network-based methodologies, we introduce Virtual Enrichment-based Signaling Protein-activity Analysis (VESPA)—an algorithm designed to elucidate mechanisms of cell response and adaptation to drug perturbations—and use it to analyze 7-point phosphoproteomic time series from colorectal cancer cells treated with clinically-relevant inhibitors and control media. Interrogating tumor-specific enzyme/substrate interactions accurately infers kinase and phosphatase activity, based on their substrate phosphorylation state, effectively accounting for signal crosstalk and sparse phosphoproteome coverage. The analysis elucidates time-dependent signaling pathway response to each drug perturbation and, more importantly, cell adaptive response and rewiring, experimentally confirmed by CRISPR knock-out assays, suggesting broad applicability to cancer and other diseases.

Cells receive and propagate exogenous signals via receptor-mediated signaling cascades, eventually resulting in the coordinated activation and inactivation of the transcriptional programs necessary to modulate cell state in response to environmental conditions. In multicellular organisms, for instance, this allows individual cells to orchestrate the gene regulatory programs necessary to progress through lineage differentiation trajectories[1] or to respond to changes in nutrient conditions[2]. Signals originating from the interaction of secreted (autocrine), microenvironment (paracrine), and distal (endocrine) ligands, and their cognate receptors, are transmitted via

[1]Department of Systems Biology, Columbia University Irving Medical Center, New York, NY, USA. [2]Yale Cancer Biology Institute, Yale University, West Haven, CT, USA. [3]J.P. Sulzberger Columbia Genome Center, Columbia University Irving Medical Center, New York, NY, USA. [4]Medical Scientist Training Program, Columbia University Irving Medical Center, New York, NY, USA. [5]Department of Medicine, Columbia University Irving Medical Center, New York, NY, USA. [6]Department of Biochemistry & Molecular Biophysics, Columbia University Irving Medical Center, New York, NY, USA. [7]Zuckerman Mind Brain and Behavior Institute, Columbia University, New York, NY, USA. [8]Herbert Irving Comprehensive Cancer Center, Columbia University Irving Medical Center, New York, NY, USA. [9]Department of Pharmacology, Yale University School of Medicine, New Haven, CT, USA. [10]Department of Biomedical Informatics, Columbia University Irving Medical Center, New York, NY, USA. [11]Chan Zuckerberg Biohub New York, New York, NY, USA. [12]Present address: Regeneron Genetics Center, Tarrytown, NY, USA. [13]These authors contributed equally: George Rosenberger, Wenxue Li, Mikko Turunen, Jing He. ✉e-mail: yansheng.liu@yale.edu; ac2248@cumc.columbia.edu

complex signal transduction cascades, whose tissue specificity depends on the availability of individual protein isoforms and on their ability to form functional complexes[3].

Dysregulation of these processes plays a critical role in human disease, especially in cancer, where signaling pathway mutations represent a hallmark of tumor initiation and progression[4]. This is exemplified by colorectal cancer (CRC), where progression from normal cells in the intestinal crypt to adenocarcinoma is determined by progressive accrual of genetic and epigenetic alterations in key signaling pathways, ultimately resulting in transformation[5]. Critically, despite similar histological presentation, we and others have shown that different CRC subtypes exist, due to signaling pathway-mediated integration of heterogeneous mutational landscapes[5], resulting in aberrant activation/inactivation of small Master Regulator protein modules[6]. Yet, the specific signaling mechanisms leading to concerted, aberrant activity of these regulatory modules and causally responsible for their time-dependent response and adaptation to drug perturbations are still largely elusive.

While their elucidation may provide more universal insights into tumor dependencies and response to treatment[6], systematic, proteome-wide elucidation of tissue-specific signaling networks has trailed the study of gene regulatory interactions. Although seminal progress has been made in recent years[7,8], the reconstruction and interpretation of signaling networks still represents one of the hallmark challenges in systems biology, with potential applications to both basic and translational research.

Signal transduction is mediated by reversible post-translational modifications (PTMs), often responsible for a rapid on/off switch in protein activity or ubiquitin-mediated proteasomal degradation. Among these, phosphorylation represents the most frequently studied event, due to its profound impact on protein conformation and function. In human cells, protein phosphorylation and de-phosphorylation is mediated by >500 kinases[9] and >200 phosphatases[10], respectively (KP-enzymes in the following). Although these enzymes have substrate specificity, determined by low to medium-affinity peptide-binding domains (PBDs), many substrates can be processed by multiple, sometimes closely related enzymes, resulting in considerable crosstalk. Auto-regulatory feedback loops, sub-cellular localization mechanisms, and context-specific availability of the cognate binding partners necessary for formation of active complexes further increase the complexity of these biological processes.

Enzyme-Substrate (ES) interactions have been broadly studied, including via low-throughput biochemical assays and structure determination[11], as well as by high-throughput methods using array-based[12], affinity purification coupled to mass spectrometry (AP-MS)[13,14], and computational biology approaches[15,16]. As a result, established repositories of ES interactions have been assembled, such as PhosphoSitePlus[17] and Pathway Commons[18], among others. However, none of these repositories addresses the context-specific nature of ES interactions and only comprise a small fraction of the total number of such molecular interactions. Furthermore, with some relevant exceptions[19–21], ES interactions have typically been studied at steady state, thus potentially failing to provide critical insight into the time-dependent signaling processes that underlie cell adaptation to endogenous and exogeneous perturbations.

A handful of reverse engineering methods for the mechanism-based interrogation of signaling pathways have been proposed, such as pARACNe (phospho-ARACNe)[22], KSEA (Kinase Substrate Enrichment Analysis)[23], INKA (Inference of Kinase Activity)[24], or PHONEMeS (PHOsphorylation NEtworks for Mass Spectrometry)[25]. However, in terms of accuracy and sensitivity, they still significantly trail behind equivalent methods for the dissection of regulatory networks[26].

To address these challenges, we here develop VESPA (Virtual Enrichment-based Signaling Protein-activity Analysis)—a phosphoproteomic-based machine learning methodology for the dissection of ES interactions and for measuring signaling protein activity—and apply it to study post-translational cell adaptation mechanisms that mediate CRC's resistance or lack of sensitivity (i.e., *insensitivity*) to clinically-relevant targeted drugs. Our proposed methodology presents four distinctive elements, including: (i) the ability to reconstruct and interrogate disease context-specific signaling networks de novo, based on phosphoproteomic profiles, (ii) the ability to measure the activity of signaling enzymes, including those that are poorly characterized in the phosphoproteomic profiles, based on the phosphorylation state of their substrates, (iii) the ability to deconvolute the time-dependent response of cancer tissues to inhibitors targeting signaling enzymes, and (iv) the ability to identify potential mechanisms presiding over drug resistance and cell adaptation. Systematic benchmarking, based on ES reference databases, assessing differential KP-enzyme activity of primary drug targets in cell lines with experimentally validated sensitivity to >200 targeted inhibitors, shows that VESPA substantially outperforms established approaches. In a proof-of-concept application, we design a large-scale drug perturbation experiment and use VESPA to elucidate the molecular mechanisms of CRC adaptation to drug treatments that mediate resistance or insensitivity in a highly context-specific fashion. VESPA analysis provides insight into the ability of CRC cell lines to adapt and "rewire" their signaling networks following drug perturbation. Critically, this reveals how specific cells may implement similar drug responses yet over highly different timeframes, while others may present highly idiosyncratic response mechanisms. Moreover, for drug resistant cells, this identifies signaling proteins responsible for the progression from initial drug perturbation to development of resistance. To assess its predictive nature, we experimentally validate these predictions using systematic CRISPR/Cas9-mediated knock-out experiments, confirming that VESPA predictions are indeed enriched in proteins that synergize with drug treatment in resistant cell lines, thus suggesting potential value towards identification of potential combination therapy opportunities.

## Results
### Conceptual workflow
VESPA comprises two steps. First, a *dissection* step (dVESPA) reconstructs tumor context-specific Signal Transduction Networks (SigNets), de novo, from phosphoproteomic and whole-proteome profiles of large-scale tumor cohorts (Fig. 1a). Such datasets—often comprising ≥ 100 samples, as required by the algorithm—are now broadly available, having been generated for many cancer subtypes by initiatives such as CPTAC. VESPA-inferred SigNets recapitulate the tumor context-specific nature of ES interactions, as well as their directionality and statistical confidence.

In a second step (mVESPA), SigNets are used to *measure* differential KP-enzyme activity in individual samples, based on the differential phosphorylation of their substrates (*signalon*), compared to a reference sample (Fig. 1b), for instance, to determine differential activity in drug vs. vehicle control-treated tissue. To infer enzyme activity, mVESPA leverages a probabilistic framework that integrates the differential phosphorylation state of its substrates, while accounting for potential confounding effects by other enzymes with potentially overlapping substrates (crosstalk). To improve performance for serine/threonine kinases (ST-Ks)—especially from low phosphoproteomic profile coverage—and to improve substrate coverage of tyrosine kinases (TKs), without requiring immunoprecipitation (IP) based enrichment methods, VESPA leverages a two-step hierarchical approach. An initial set activity profile layer is generated by KP-enzyme's substrate phosphostate analysis and is then refined by an additional network analysis step.

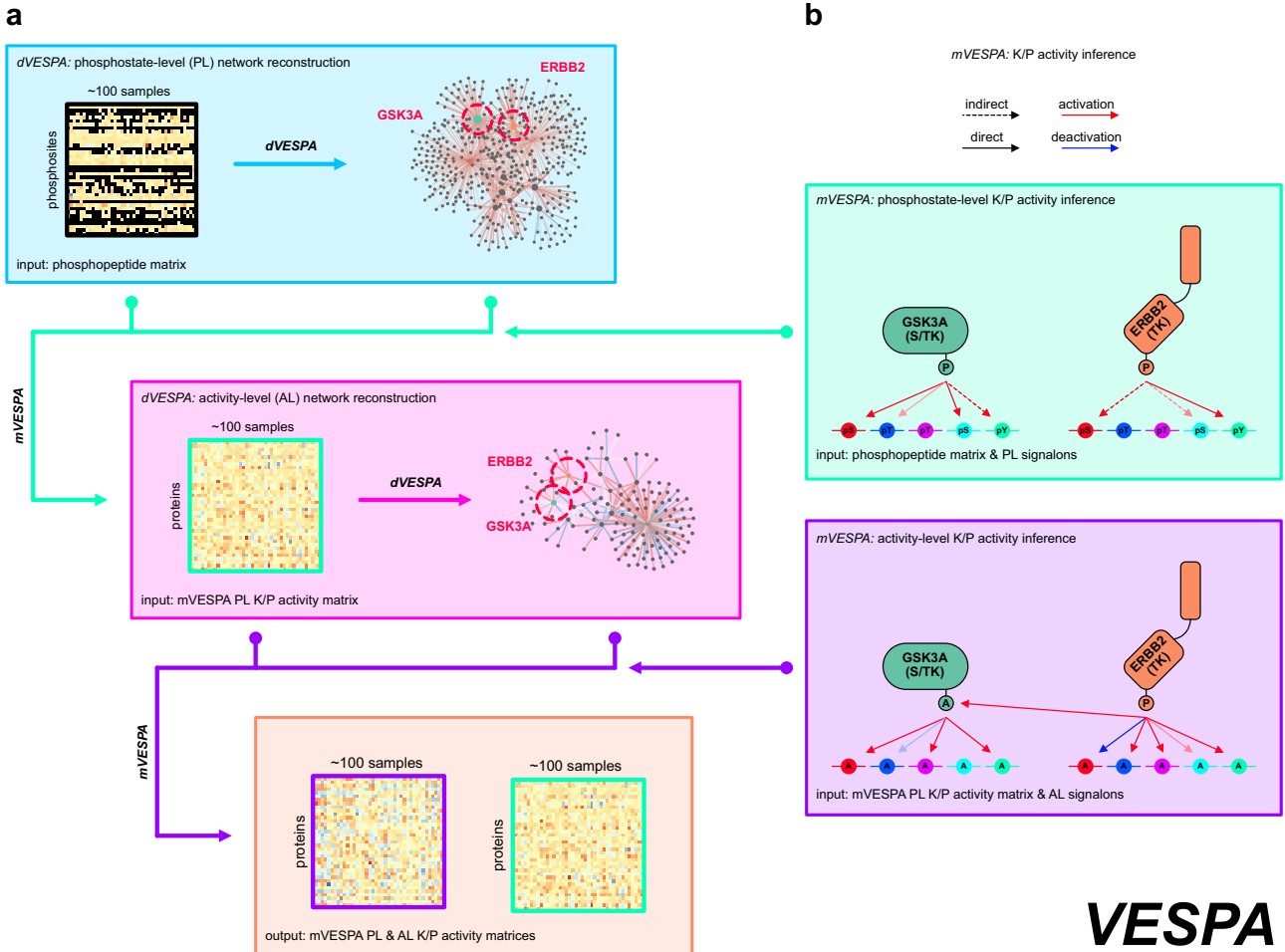

**Fig. 1 | Methodological overview. a** VESPA assesses the activity of protein kinases and phosphatases based on the phosphostate of their substrates. As an input, dVESPA requires a matrix representing the phosphopeptide or phosphosite abundance of a collection of samples representing different conditions of a specific cellular context, including missing values (black). The signaling network reconstruction module (blue box) analyzes this matrix to first identify candidate signal transduction interactions by assessing the significance of the mutual information between enzymatic regulator and candidate target phosphopeptides, second, remove indirect interactions by applying a signal transduction-specific form of the Data Processing Inequality (stDPI), and third, generate signalons for each KP-enzyme representing the probability of each interaction with a substrate and the mode of regulation (kinase activation: red, phosphatase deactivation: blue). **b** mVESPA first uses these signalons to assess KP-enzyme activity at the

"phosphostate-level" (green box). The resulting KP-enzyme activity matrix then becomes the input to an additional protein activity assessment step of dVESPA (pink box) which uses the (standard, non-signal transduction) Data Processing Inequality (DPI) to generate more abstract signalons (i.e. representing activation/deactivation instead of phosphorylation/dephosphorylation), which are then in turn used by mVESPA for activity-level inference (purple box). Methodological differences between phosphostate- and activity-level signaling networks. At the phosphostate-level, ST-Ks (e.g., GSK3A, green) are primarily associated with direct phosphorylation targets, whereas TKs (e.g., ERBB2, orange) can frequently not be directly associated with (unenriched) tyrosine-phosphorylated sites. On activity-level, more abstract "activation/deactivation" events can better associate targets for both ST-Ks and TKs.

Despite a superficial similarity of these steps to algorithms designed for the study of transcriptional networks, such as ARACNe[27,28] and VIPER[29], there are critical differences that were necessary to account for the unique structure and sparseness of phosphoproteomic profiles. These are summarized in the following.

Substrate inference: to extend the ARACNe algorithm[27,28] to phosphoproteomic data (see Methods), dVESPA assesses mutual information via a hybrid partitioning approach (hpMI) which supports use of continuous peptide intensities from quantitative proteomic workflows[30]. This addresses issues associated with missing values due to censoring[31,32], typical of bottom-up phosphoproteomic analyses (Supplementary Fig. 1a, Methods). Furthermore, to support the logic of three-way signaling interactions, as implemented by kinases and phosphatases measurable by standard MS-based phosphoproteomic methods, dVESPA introduces a signal transduction-specific version of the Data Processing Inequality (stDPI) (Supplementary Fig. 1b, Methods).

Critically, indirect interactions (e.g., $K_A \rightarrow S$, implemented as $K_A \rightarrow K_B \rightarrow S$) are eliminated if both direct interactions (i.e., $K_A \rightarrow K_B$ and $K_B \rightarrow S$) are detectable and have higher mutual information. If this is not the case, for instance because $K_B$ is poorly resolved in the dataset, then $K_A \rightarrow S$ will be identified as the "least indirect" interaction between $K_A$ and S. As a result, it is possible that some indirect interactions may be represented in the SigNet, especially if the phosphostate of the intermediary enzyme (i.e., $K_B$ in the above example) is noisy or undefined.

To complement ES interactions inferred de novo, dVESPA can incorporate context-free knowledge from reference databases—such as Pathway Commons[18], LinkPhinder[16], or the Hierarchical Statistical Mechanistic model (HSM)[15]. Each inferred interaction is associated with a *p-value* and a *directionality*—as determined by the proteins' enzymatic function (Methods).

Cross-talk correction: mVESPA includes the pleiotropy correction[29] method, which was designed to address potential issues

associated with overlap in the substrates of different enzymes (see Methods).

Site-specific activity inference: enzyme phosphostate is measured by mVESPA at both the *whole-protein* level—i.e., by integrating the state of all phosphosites—or at the *phosphosite-specific* level (Methods). The latter can help elucidate phosphosite-specific contributions to protein activity. Indeed, distinct phosphosites may result in different, potentially opposite contributions, ranging from ubiquitylation pathway activation to mediating critical dimerization or conformational changes, to sites providing no measurable contribution.

Hierarchical activity and model inference: Unless specifically enriched for, some substrates may be only sparsely represented, resulting in low-quality signalon inference. This is especially problematic for phospho-tyrosines. To address this challenge, mVESPA implements a two-step approach (Fig. 1, Supplementary Fig. 1c-d, Methods). In a first *phosphostate-level analysis (PL-analysis)* step, KP-enzyme activity is assessed from its signalon's phosphostate. In a second, activity-level analysis (*AL-analysis*) step, activity assessment is refined by using candidate substrates' activity rather than phosphostate, as assessed in step 1 (Methods). Indeed, since many TK substrates are ST-Ks, their activity may be assessed more accurately than their phosphostate. PL and AL-analyses are then integrated, using Stouffer's method, since substrate activity and phosphostate are assessed from statistically independent data (Methods).

Signalon optimization: If multiple datasets are used to generate signalons for a KP-enzyme, mVESPA will only use the most informative one, as assessed by the statistical significance of the KP-enzyme's differential activity, similar to the metaVIPER algorithm[33] (Methods).

Applicability to different dataset types: To be analyzed, phosphoproteomic datasets must fulfill several criteria: First, a minimum of 100 phosphoproteomic profiles[27]—ideally including whole protein measurements—should be generated from the same tissue context. Sufficient phosphoproteome coverage (>10,000 phosphosites) and quantitative consistency (>40%) is also required. These criteria are not limiting and are fulfilled by most CPTAC or DIA-based datasets. Lower proteome coverage will increase the number of indirect interactions and decrease the quality of activity measurements. Lower quantitative consistency or bias (*e.g.*, labeling, batch effects) may substantially reduce sensitivity. Consistently, datasets used for mVESPA-based enzyme activity analysis must be similarly quantitatively consistent (>40%) and have a substantial overlap (>50%) of measured phosphosites with the dataset used for dVESPA signalon inference. These requirements are also fulfilled by most CPTAC or DIA-based datasets.

## Generating a CRC-specific SigNet

Kinase inhibitors targeting a protein's active site typically modulate their targets' activity without affecting their phosphostate. As a result, drug target identification by proteomic methods is non-trivial. SigNet availability mitigates this issue by supporting enzyme activity assessment in drug vs. vehicle control-treated cells based on substrate's phosphostate. To apply this approach to colorectal cancer (CRC), we leveraged three proteomic and phosphoproteomic datasets, including (a) 97 profiles from the Clinical Proteomic Tumor Analysis Consortium (NCI/NIH) (CPTAC-S045[34]), (b) refined profiles obtained by normalizing the phosphosite abundance of CPTAC-S045 samples by the corresponding whole protein abundance (Methods), to help identify confounded KP→S relationships, as previously suggested[35] (CPTAC-S045N), and (c) 144 profiles from six CRC cell lines (HCT-15, HT115, LS1034, MDST8, NCI-H508 and SNU-61) harvested at three-time points (1 h, 24 h, 96 h) following perturbation with seven clinically relevant drugs and vehicle control media (U54-NET).

We used dVESPA to dissect independent SigNets from these datasets (Supplemental Data 1, Methods). Overall, consistent with the number of KP-enzymes expected to be expressed in any specific cellular context, signalons comprising 5 or more candidate substrates were reliably

inferred for 51.0% of human KP-enzymes, from at least one of the datasets. The first step (PL-analysis) produced a SigNet comprising 163,313 interactions, between 283 kinases, 88 phosphatases, and 7727 substrates. The second step (AL-analysis) identified 16,309 additional interactions, between 187 kinases, 37 phosphatases, and 371 substrates. To support more mechanistic analyses, we also generated a phosphosite-level network, comprising 1649 individual phosphosites. Collapsing phosphosites in the same peptide-binding domain— frequently correlated in both phospho-state and functional role—reduced this to the interactions between 918 non-redundant phosphosites (Methods). Each interaction was associated with a *mode of regulation* (i.e., substrate activation or deactivation by kinases and phosphatases, respectively) and *p*-value.

As expected, due to lower genetic background variability in selected cell lines, different MS measurement time per sample, and different depth of proteomic data acquisition methods (DDA-TMT vs. DIA-LFQ), CPTAC provided a more comprehensive phosphosite representation than cell line perturbations, specifically, 31,339 vs. 13,529 phosphosites in CPTAC-S045 and U54-NET, respectively. However, U54-NET signalons were often selected as more informative (Methods). Indeed, at the phosphostate-level, 47.2%, 43.4%, and 9.4% of the optimized signalons were derived from CPTAC-S045, U54-NET, and CPTAC-S045N dataset, respectively. Dataset specificity was even more skewed at the activity-level analysis, where U54-NET accounted for 46.4% of the optimized signalons, with CPTAC-S045 and CPTAC-S045N accounting for 38.4% and 15.2% of them, respectively.

A key advantage of mVESPA is that, once a SigNet is available, KP-enzymes' activity can be measured even if their phosphostate is undetectable. Indeed, VESPA could measure enzymatic activity for 158 of 371 (42.6%) of all KP-enzymes in the CRC SigNet that lacked phosphostate information. Furthermore, multiple dataset integration can effectively combine DIA's high throughput with the more comprehensive nature of the fractionated CPTAC profiling. Overall, despite the well-known sparseness of peptides and phosphopeptides detected by proteomic assays, mVESPA quantitatively assessed the activity of 371 KP-enzymes—i.e., around half of all known human KP-enzymes and around 66.7% of the KP-enzymes estimated to be expressed in CRC cells (Methods). In contrast, phosphostate information was available for only 42.7% of expressed KP-enzymes.

## Mutual information estimator benchmark

Typical phosphoproteomic profiles comprise between 20% and 80% missing values, making phosphopeptide-based MI estimation challenging. To address this issue, we introduce a hybrid-partitioning mutual information metric (hpMI, see Methods). We benchmarked its performance using the U54-NET dataset, compared to either removing proteins with missing data (depleted MI; dMI) or imputing values using random, low intensity noise (imputed MI; iMI). As ground truth, we used the interactions and priors predicted by the Hierarchical Statistical Mechanical (HSM)[15] modelling algorithm, which, albeit more limited in scope, represent the most faithful statistics mechanics model of these interactions. All MI scores are expected to recover well correlated ($\rho > 0.5$) interactions with few missing values (<20%). However, hpMI was particularly designed to also improve recovery of weakly correlated interactions ($\rho > 0.25$) with larger proportions of missing values. To illustrate this improvement in dependency of correlation of interactions and the proportion of missing values, we computed a score representing the recovery as the count of significant interactions as judged by the different MI estimators (BH-adjusted $p < 0.05$ estimated using a bootstrapped null model[28], see Methods), weighted by the corresponding HSM priors. When applied to data subsets of varying consistency, removing up to 80% of the data in 20% increments (Methods), the recovery can also be visualized in dependency of the correlation between the data points to illustrate the differences between dMI, iMI and hpMI estimators (Fig. 2a). As expected, for some well-sampled, highly correlated KP→S pairs, both dMI and

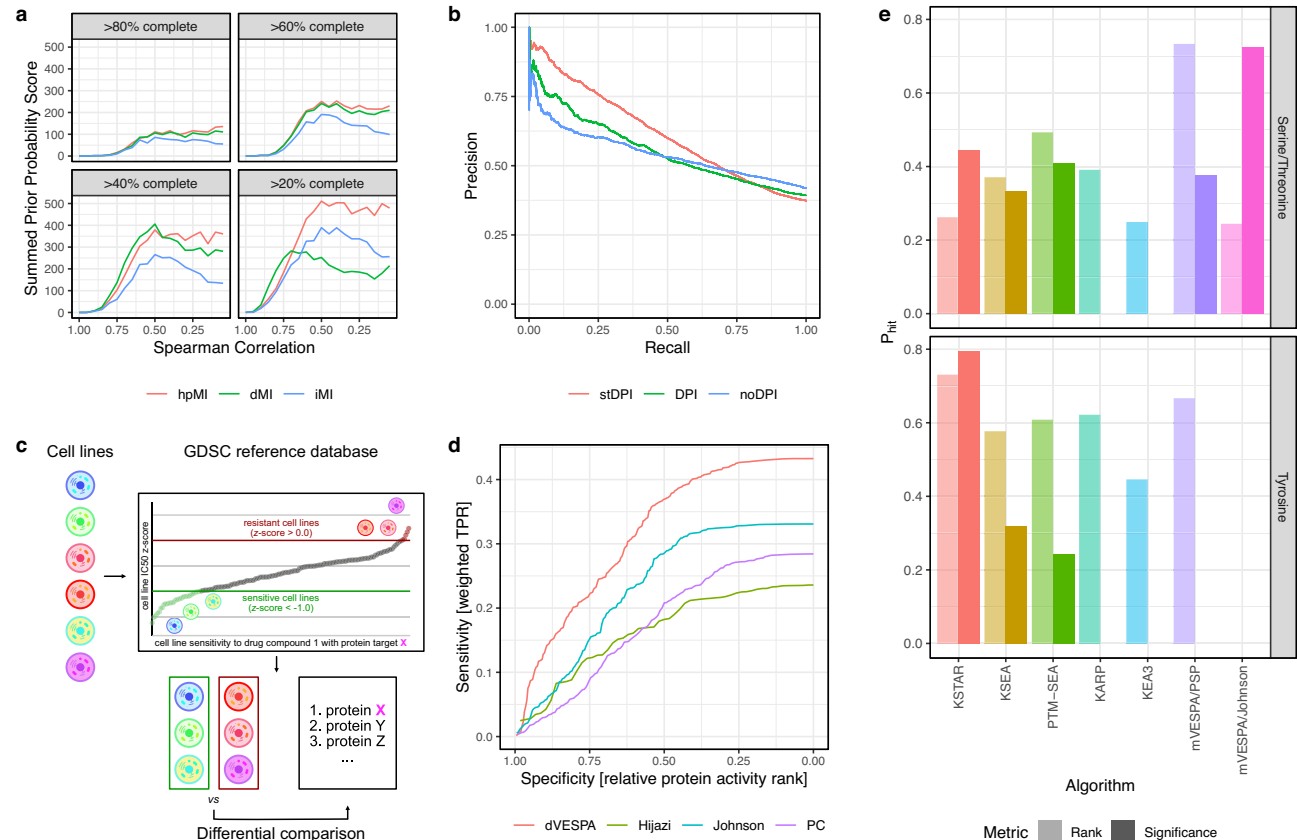

**Fig. 2 | Benchmark and validation of dVESPA and mVESPA. a** Comparison of different mutual information (MI) estimation strategies based on imputation (iMI), depletion (dMI) and hybrid partitioning (hpMI) and the MI – Spearman correlation relationship using the CPTAC-S45 dataset. MI was measured and Spearman correlation was computed for each KP-enzyme/target pair using data from the CPTAC-S45 dataset (Methods). **b** Precision-recall curves were used to evaluate the difference between regular Data Processing Inequality (DPI) and its adaptation to signal transduction networks (stDPI), as well as compared to not using the DPI step at all (noDPI). **c** Baseline profiles of six diverse CRC cell lines were acquired and used with the GDSC reference database to identify sensitive and resistant or insensitive cell lines for each drug. The most differentially active KP-enzymes, as induced by treatment with each drug, were then assessed using mVESPA. **d** The predictive performance of the analysis results of (**c**), comparing dVESPA and other reference networks (Pathway Commons (PC)[18], Hijazi et al.[20] and Johnson et al.[37]), was evaluated using receiver-operating-characteristics (ROC). For each differential

comparison, ROC metrics were computed, where the sensitivity represents the mVESPA scores, weighted by GDSC drug sensitivity, and the selectivity represents a normalized rank of the top VESPA hits (see Methods). The individual ROC curves were then averaged. Statistical comparison of the differential comparison AUC metrics was conducted using an unpaired, right-tailed Wilcox' tests. VESPA (red), comprising the mVESPA and dVESPA steps, significantly outperformed mVESPA when run using non-context-specific SigNets, including Johnson et al.[37] (blue), Hijazi et al.[20] (green) and Pathway Commons[18] (purple). **e** Benchmark against established algorithms and applicability to datasets with $N = 1$ samples. The KSTAR benchmark was extended according to the original publication. Algorithm performance on S/TKs or TKs is depicted, computed as the fraction of conditions (specific cell line perturbed by a specific drug) for which a perturbed kinase was assessed as differentially active ($P_{hit}$), i.e., either ranked in the top 10 most differentially active (translucent bars) or based on statistical significance (FDR < 0.05) (opaque bars). Source data are provided as a Source Data file.

iMI measured a statistically significant MI, supported by positive ground truth priors; however, hpMI inferred 102.4% more correct ES interactions than dMI, and 31.3% more correct ES interactions than iMI when using sparsely covered interactions (up to 80% missing values), particularly in case of lower ($\rho < 0.5$) KP → S correlation (Fig. 2a).

### Indirect interaction removal

To eliminate indirect interactions (e.g., KP → KP' → S) when a more statistically significant direct interaction (KP → S) exists, dVESPA uses a *signal transduction-adapted* version of the Data Processing Inequality (stDPI/DPI) originally proposed in[27,28] (Supplementary Fig. 1b, Methods). The DPI states that, in any system where information is not perfectly transferred (lossy)—thus including virtually all molecular networks—direct information transfer (i.e., KP → S) is always greater than indirect information transfer (KP → KP' → S). Application of this theorem allows effective indirect interaction removal.

To assess whether the stDPI improves indirect interaction removal, also compared to the original DPI formulation, we first generated a gold-standard dataset for ST-K proteins using the HSM[15]

algorithm (Methods). Specifically, ground truth interactions were selected based on HSM analysis of domains identified as primary determinants of ST-K → phosphopeptide specificity, including PDZ, SH3, WH1, and WW domains. As a negative gold standard, we used HSM predicted TK → S interactions, based on PTB, PTP and SH2 domains, since the dataset used for this benchmark (U54-NET) is not enriched for phosphotyrosine peptides and should thus not support their identification. It should be noted though, that the HSM gold standard data is not context-specific because its interactions, although biochemically plausible, might not be implemented in the cellular context of interest, thus reducing benchmark results. As such, only relative comparisons are possible.

dVESPA-based generation of a SigNet, using the U54-NET, with the PL-based methodology but without a prior reference network, was tested with each of the three DPI options: (a) no DPI, (b) regular DPI, and (c) stDPI. Inferred interactions were then compared to the gold standard datasets (Methods). Receiver operating characteristics (ROC) and precision-recall curves show that stDPI significantly outperforms the other two options (see Methods), including for stDPI vs. no DPI

($p < 2.2e\text{-}16$) and stDPI vs. DPI ($p < 2.2e\text{-}16$), see Fig. 2b for the specific receiver operating characteristics (AUROC) and area under the precision recall curve (AUPRC). For instance, at 25% recall, stDPI achieved 75.7% precision, compared to 65.2% for DPI and 60.3% for no DPI.

Taken together, these benchmarks confirmed that hpMI and stDPI —two distinct phosphoproteomic-specific components of dVESPA— significantly improve algorithm performance.

## mVESPA Benchmarking

To benchmark mVESPA, we extended a strategy previously introduced to benchmark the INKA (Integrative Inferred Kinase Activity) algorithm[24]. The Genomics of Drug Sensitivity in Cancer (GDSC) project reports on the sensitivity of >1000 human cancer cell lines to hundreds of drugs and small molecule compounds (i.e., drugs, for simplicity), including high-affinity kinase inhibitors[36]. When combined with a curated list[24] of the primary (i.e., high-affinity) targets of each inhibitor, this resource can be used to effectively assess relative kinase activities, as originally proposed in[24]. Specifically—within a specific tumor type and barring adaptive resistance mechanisms—higher enzyme activity should correlate, on average, with increased sensitivity to its high-affinity inhibitor(s). Higher correlation, across multiple cell lines, would thus indicate improved activity assessment, allowing comparative analysis of different protein activity prediction algorithms.

For this benchmark, we predicted the activity of protein kinases representing high-affinity targets of GDSC-tested inhibitors, by using dVESPA to analyze the baseline (i.e., unperturbed) phosphoprofiles of six CRC cell lines, profiled in triplicate (U54-BL), see Methods. To support the comparative analysis of multiple methods, we modified the benchmark to use differential rather than absolute protein activity ranks, see Methods.

As a first step, we assessed performance differences when using either dVESPA-inferred (i.e., context-specific) signalons or signalons reported by other sources, including generalized and contextualized reference databases. Specifically, we either restricted the comparative analysis to the protein kinases analyzed by all methods (intersection) or to all protein kinases (full) (Methods). The former is used to assess prediction accuracy, while the latter determines method-specific network coverage. These analyses show that dVESPA significantly outperformed the generalized reference databases obtained from Johnson et al.[37], Hijazi et al.[20], and Pathway Commons[18], for both ST-Ks and TKs activity inference (intersection: max $p < 2e\text{-}6$, full: max $p < 0.005$) (Fig. 2d, Supplemental Figs. 2a-3a, Supplemental Data 2-3, Methods). Furthermore, indirect interactions removal by stDPI/DPI showed a trend towards higher accuracy (intersection: $p < 0.156$) but higher network coverage (full: $p < 1.9e\text{-}6$), compared to using a contextualized reference network from LinkPhinder (LP)[16], and improved on both counts (intersection: $p < 0.003$, full: $p < 4.4e\text{-}4$) when using HSM[15] as a reference network. For TK enzymes, stDPI/DPI improved network coverage vs. LP (intersection: $p < 0.580$, full: $p < 4.7e\text{-}4$) but did not improve either metric compared to HSM (intersection: $p < 0.766$, full set: $p < 0.947$) (Methods, Supplemental. Figs. 2b-3b, Supplemental Data 2, 3).

We then benchmarked performance differences associated with each mVESPA component, including (a) signalon integration and optimization across multiple dataset (Supplemental Figs. 2c-3c, Supplemental Data 2, 3), (b) differences between phosphostate-level, activity-level and integrated analysis (Supplemental Figs. 2d-3d, Supplemental Data 2, 3), and (c) the effects of crosstalk correction (Supplemental Figs. 2e-3e, Supplemental Data 2, 3). Benchmarking only signalons with U54BL-measured phosphopeptides indicates that VESPA performs very similar on this subset when assessing all kinases in comparison to the full dataset, although with lower sensitivity. Further, it should be noted, that this result could also be confounded due to the bias of the benchmark towards well studied or experimentally better accessible KP-enzymes. While 83.9% of all comparisons

of the benchmark cover targets with U54BL-measured phosphopeptides, the fraction of CRC signalons that cover directly measured K/P-enzyme phosphopeptides is only 57.4%. Interestingly, when only considering TKs, inclusion of signalons without measured K/P-enzyme phosphoproteins, expectedly increased substantially (Supplementary Fig. 4).

Taken together, these analyses confirm the value of the individual improvements in mVESPA as well as their cumulative effect. Indeed, the latter produced the best overall performance and a statistically significant improvement over the current state-of-the-art (Fig. 2d). Based on these results, for all subsequent studies, we used stDPI for PL-based and regular DPI for AL-based signalon inference, respectively, followed by integration using Stouffer's method (Methods).

## Comparison to established algorithms and applicability to independent samples

To compare VESPA to other algorithms for the dissection of signal transduction networks—including KSTAR[38], KSEA[23], PTM-SEA[39], KARP[19], and KEA3[40]—we relied on the benchmarking dataset, tools, and evaluation criteria recently developed for the KSTAR algorithm[38]. The specific dataset comprises phosphoproteomic profiles following genetic or pharmacologic inhibition of 38 serine/threonine and 19 tyrosine kinase in multiple cell lines as derived from 15 individual studies[38]. Each algorithm was tested independently on perturbational profiles. Unfortunately, this dataset severely limits VESPA's performance, for two reasons. First dVESPA signalons, which provide the greatest contribution to the algorithm's performance, could not be used because their generation requires ≥ 100 independent phosphoproteomic profiles of the investigated biological system[27]. Second, key elements of mVESPA's analytical framework, such as the hierarchical approach and crosstalk correction, could not be used as they also require multiple profiles acquired by the same quantitative proteomic method.

As a result, we could compare existing algorithms only to a highly restricted version of VESPA that (a) used non-context-specific signalons from PhosphoSitePlus[17] (mVESPA/PSP), as also used by KSTAR, KSEA, and KARP, and Johnson et al.[37] (mVESPA/Johnson), (b) could not leverage the hierarchical PL/AL approach and (c) could not leverage the cross-talk correction (Methods). As a result, these analyses provide only a lower limit to VESPA's performance.

Despite these limitations mVESPA/PSP ($P_{hit} = 0.73$) and mVESPA/Johnson ($P_{hit} = 0.73$) outperformed all other methods ($P_{hit} \leq 0.49$) (Fig. 2e). As discussed in[38], $P_{hit}$ represents the fraction of experimentally inhibited protein kinases identified as differentially active, either based on rank (top 10 most inactivated kinases) or statistical significance (FDR < 0.05). When restricting the analysis to the much smaller set of TKs, mVESPA/PSP's ($P_{hit} = 0.67$) outperformed all other methods ($P_{hit} \leq 0.62$), except KSTAR ($P_{hit} = 0.79$). mVESPA/Johnson could not be assessed because the related dataset does not include TKs. Taken together, these data show that VESPA outperformed all existing algorithms on the analysis of ST-Ks, which comprise the vast majority of kinases, and all but KSTAR on the analysis of the much smaller set of TKs, even though the most critical component (i.e., the use of a context-specific network produced by dVESPA) could not be leveraged.

## Application of VESPA to the decryptM dataset

A recent study investigated the effects of drugs on PTMs using dose- and time-resolved proteomics, referred to as "decryptM"[21]. To demonstrate VESPA's applicability to this dataset, we applied the algorithm to the phosphoproteomic profiles for A431 epidermoid carcinoma cells (dependent on EGFR expression), perturbed by afatinib (targeting EGFR), gefitinib (targeting EGFR), and dasatinib (targeting SRC- and EPH-family proteins) with 10 different drug concentrations (Supplementary Fig. 5, Methods). Because epidermoid

carcinoma is not covered by CPTAC, we used a dVESPA-generated signaling network based on the CPTAC Lung Squamous Cell Carcinoma (LSCC) Discovery Study[41], with the caveat that our networks are not fully representative of A431 cell lines. We then used the VESPA approach to infer kinase activities for all covered KP-enzymes and focused interpretation on the known targets as listed by DrugBank. We considered a VESPA NES (z-score) of NES < −1.65 ($p < 0.05$) to be the threshold for significant inhibition.

Our analysis shows significant inhibition of EGFR for both afatinib and gefitinib treatments with median z-scores of -3.49 ($p = 0.0002$) and −2.03 ($p = 0.02$), respectively (Supplementary Fig. 5). ERBB2 was also significantly inhibited by afatinib, resulting in a median z-score of −2.24 ($p = 0.01$). Interestingly, only concentrations equal to or higher than 1 nM induced significant inhibition of the primary targets. For dasatinib, 11 out of 15 covered DrugBank targets showed negative activity, with only MAPK14 being significantly inhibited (z-score = −2.09; $p = 0.02$). Using orthogonal assays (kinobeads), the original authors of the decryptM study observed a wider distribution of drug-target affinities for dasatinib than for afatinib and gefitinib, supporting the notion that not all known drug targets might be effectively inhibited in all cellular contexts.

## Cell Line Selection for CRC Analysis

To study CRC-specific drug mechanism of action and cellular adaptation, we leveraged pharmacologic perturbations of cell lines selected to represent high-fidelity models of established CRC subtypes. Model fidelity was based on the overlap of Master Regulator (MR) proteins, representing critical determinants of transcriptional cell state, in each model vs. a collection of human tumor samples, using the OncoMatch algorithm[42,43]. We use this definition because we have shown that the mechanism of action of a drug in a tumor is well recapitulated in their OncoMatch-selected high-fidelity cell lines[43,44].

For this purpose, we first focused on eight CRC subtypes, as recently identified by MR-based stratification of the TCGA CRC cohort[45]. We then used the OncoMatch algorithm to identify Cancer Cell Line Encyclopedia (CCLE)[46] cell lines representing high-fidelity models of each subtype (Methods). When also accounting for other parameters—e.g., optimal growth in culture and suitability to high-throughput microfluidics—six cell lines were identified, including HCT-15, HT115, LS1034, MDST8, NCI-H508 and SNU-61. These represent 5 of the 8 CRC subtypes, with at least one cell line ranking in the top 5 for each subtype (Supplementary Fig. 6). As such, three tumor subtypes lack ideal representation in CCLE and could not be studied.

We then proceeded to assess whether these cell lines were also matching subtypes identified by phosphoproteomic cluster analysis, as determined by OncoMatch analysis of their KP-enzyme differential activity. The latter was assessed by VESPA analysis of 97 clinically annotated CRC samples in the CPTAC-S045 cohort[34]. To perform the analysis, we first generated phosphoproteomic profiles from each unperturbed cell line, in triplicate, by label-free DIA. At 1% peptidoform and protein FDR, the analysis identified and quantified the state of 9813 phosphosites on 18,012 unique peptide precursors mapping to 3320 proteins (Methods). We will refer to this dataset as the "U54-BL". At the peptide precursor-level, the dataset/matrix completeness—i.e., the fraction of runs where peptide precursors were confidently detected and quantified—ranged from 77.3% to 83.1% per cell line, while the average completeness over all cell lines and replicates was 54.2%. CPTAC samples are profiled via a tandem mass tag (TMT)-based workflow; as such, they present even deeper coverage, with 31,339 phosphosites from 6383 proteins, and a matrix completeness of 40.2%. However, due to the data-dependent acquisition (DDA) and TMT-labelling approaches used for data collection, these profiles present considerable batch effects. To optimally compare cell lines to tumor samples, we identified a subset of 8617 shared phosphosites, presenting equivalent completeness (Methods). We then used VESPA to

assess protein activities, as previously described (Fig. 3, Methods). The analysis yielded an activity matrix comprising 381 common KP-enzymes for both tumor samples and cell lines (Supplementary Figs. 7–9, Supplemental Data 4).

Activity-based analysis of the CPTAC dataset, using K-medoids clustering[45], identified three main clusters ($VC_1$ – $VC_3$) (Methods), while Random Forest-based, recursive feature elimination identified the KP-enzymes with the greatest independent contribution to subtype classification (Fig. 3, Supplemental Data 5-6, Methods). KP-enzyme-based OncoMatch analysis confirmed that most of the selected cell lines matched one of these three subtypes. Specifically, HCT-15 and HT115 matched $VC_1$, NCI-H508, LS1034 and SNU-61 matched $VC_2$ and MDST8 matched $VC_3$. Notably, one replicate of HT115 was assigned to $VC_2$ instead of $VC_1$.

For completeness, we also assessed whether the six cell lines could recapitulate four subtypes ($CMS_1$ – $CMS_4$) identified by transcriptomic analysis of the Consensus Molecular Subtype (CMS) dataset, as reported by the Colorectal Cancer Subtyping Consortium (CRCSC)[47] (Methods). The analysis revealed broad consistency between CMS and VESPA classification (Fig. 3a, colored, non-white labels). Specifically, $VC_1$, $VC_2$, and $VC_3$ samples were significantly enriched in $CMS_1$, $CMS_2$, and $CMS_4$ samples, with $CMS_3$ samples split between $VC_1$ and $VC_2$, likely as a result of the finer-grain stratification achieved by transcriptional analysis, which reflects epigenetics differences that may not affect signal transduction. OncoMatch analysis identified the NCI-H508 and LS1034 cell lines as high-fidelity models for $CMS_2$ samples, SNU-61 for $CMS_3$, and MDST8 for $CMS_4$, confirming that the cell line panel identified by our analysis broadly represents patient-relevant subtypes (Fig. 3a, b). Note that HCT-15, and HT115 could not be confidently classified into one of the CMS clusters. A recent study[48] produced similar results when matching CRC cell lines to CMS clusters; while MDST8, NCI-H508, LS1034, and SNU-61 were well classified, HT115 produced an ambiguous matching, and HCT-15 was not reported, suggesting finer-grain subtype identification by MR-based analysis.

Gene set enrichment analysis[49] (GSEA) using the Reactome database[50] further supported these results, based on several signaling pathways that were uniquely enriched in the three VESPA clusters ($p < 0.05$, Benjamini-Hochberg (BH)-corrected, see Methods) (Fig. 3c, Supplemental Data 7). For instance, we identified enrichment of VEGFA-VEGFR2 Pathway in $VC_3$, a hallmark of the $CMS_4$ subtype[47], which was further supported by the activation of RHO GTPases involved in WAVE complex regulation, a key regulator of actin-remodeling, invasiveness and EMT-like processes[51] (Fig. 3c). This was recapitulated by the MDST8 cell line in our panel, representing an established EMT model[52].

In summary, except for $CMS_1$, for which no representative cell lines could be identified, the six cell lines selected for our study effectively represent the major CRC subtypes inferred by either transcriptional or phosphoproteomic analysis.

## Generation of drug perturbation profiles

To assess drug mechanism of action (MoA), CRC cell adaptive mechanisms leading to drug resistance, and potential treatment-mediated rewiring of signaling pathways, we performed a longitudinal drug perturbation assay, supporting quantitative analyses across drugs, cell lines, and time points (Methods). To achieve a reasonable experimental complexity, we focused on seven clinically relevant compounds, based on their ability to target complementary, CRC-relevant pathways. With the exception of WIKI4 (a TNKS & TNKS2 inhibitor), these represent FDA-approved drugs for the treatment of CRC and related cancer types, including alpelisib (PIK3CA), imatinib (ABL1/3 & c-Kit[53]), linsitinib (IGF1R[54]), osimertinib (EGFR-T790M), ralimetinib (p38 MAPK), and trametinib (MEK1 & MEK2). Although some of these compounds were designed to target genes harboring specific

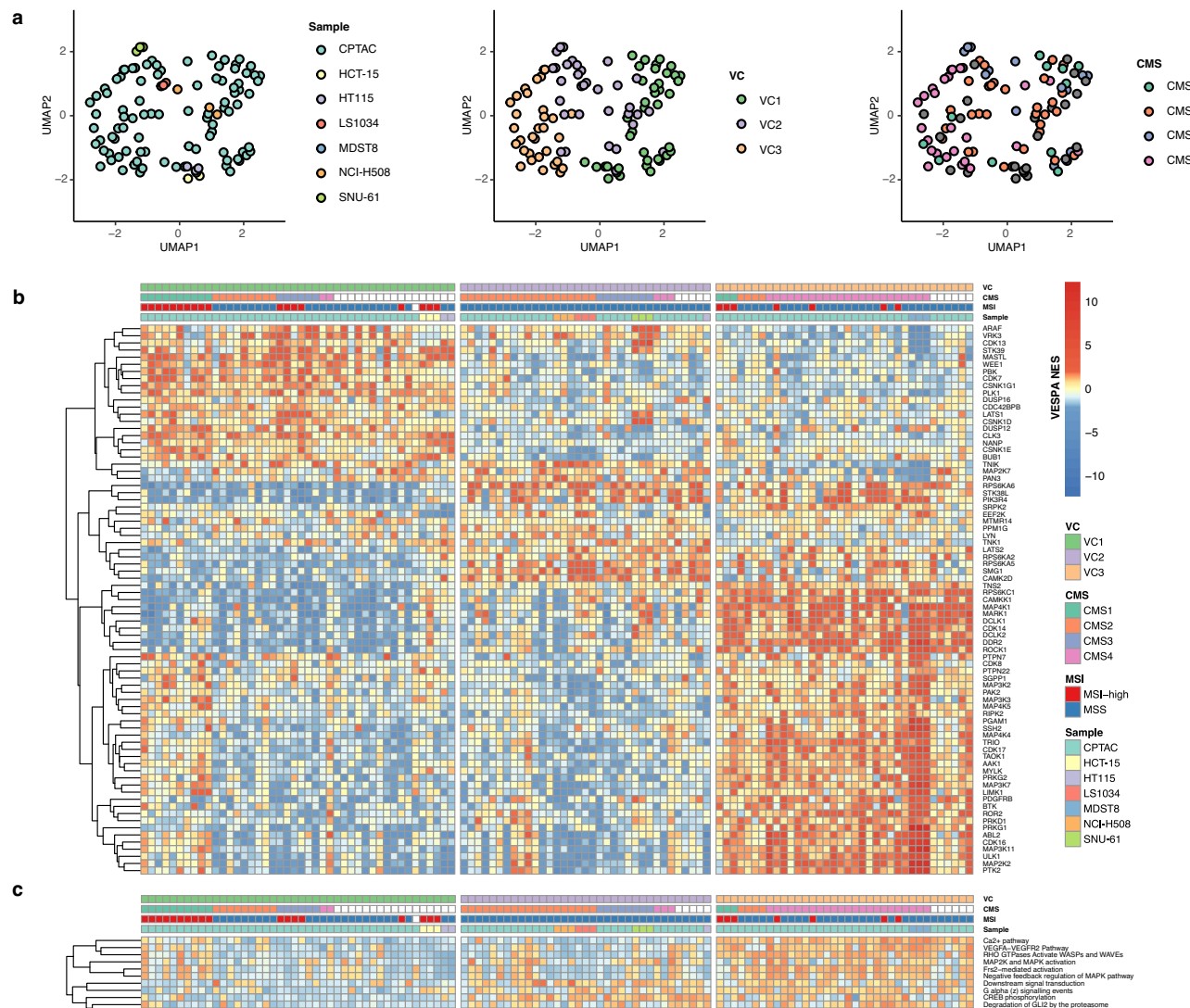

**Fig. 3 | Representation of CRC subtypes by cell line models. a** Uniform Manifold Approximation and Projection (UMAP) embedding of KP-enzyme activity color-coded according to different classification systems (phosphoproteome-based VESPA; VC and the CRC Consensus Molecular Signature; CMS). **b** The most informative proteins and their VESPA inferred normalized enrichment scores (NES) were selected for visualization (full datasets: Supplementary Fig. 7–9). CPTAC clinical profiles and cell lines were grouped according to the Consensus Molecular

Classifier (CMS), VESPA clusters (VC), and microsatellite instability (MSI). Samples are grouped according to VC. **c** Gene Set Enrichment Analysis (GSEA) using a signal transduction-specific subset of the Reactome database. Only terms significant in at least one sample (FGSEA ES-test two-tailed BH-adj. $p < 0.05$) are shown. The colors represent GSEA NES and are linked to the legend in b). Source data are provided as a Source Data file.

mutations (e.g., osimertinib[55] and alpelisib[56]), we used a mutation-agnostic approach to the analysis, since targeted drugs can also inhibit wild-type proteins[56] or have off-target effects on unrelated proteins[57]. In the case of alpelisib, the cell line panel represents both mutated (HCT-15, HT115, NCI-H508) and wild-type (LS1034, MDST8, SNU-61) PIK3CA genes. Osimertinib is an effective EGFR inhibitor, which, in contrast to erlotinib, is not affected by EGFR-T790M mutations[58]. As such, the presence of T790M mutations was not considered in the analysis.

Assessing drug MoA requires careful selection of an optimal, physiologically achievable concentration in vivo, at which the MoA is manifested with minimal activation of cell stress and death pathways, as well as off-target effects, representing critical confounding factors. Consistent with our prior studies[59,60], we thus selected the highest sublethal concentration of each compound, as defined by the lowest of (a) the reported $C_{max}$ (maximum tolerated serum concentration in vivo) and (b) the 48 h $IC_{20}$ in the most sensitive cell line from our

panel, as experimentally determined by 10-point dose-response curves (Methods). Concentrations were also capped at ≤ 0.5 µM, consistent with maximum levels achievable in tissues. Imatinib, osimertinib, rali-metinib, and WIKI4 were thus titrated at 0.5 µM, while alpelisib, linsi-tinib, and trametinib were titrated at 0.12 µM ($IC_{20}$), 0.14 µM ($IC_{20}$), and 0.036 µM ($C_{Max}$), respectively (Methods).

Differentiating between sensitive and resistant cell lines is also non-trivial[61]. For example, as determined by the GDSC reference data, the frequently applied threshold of $IC_{50} \leq 1.0$ µM, would yield a resis-tant phenotype for 23 of 27 of our cell line/drug combinations (Sup-plementary Fig. 10a)[61]. To select a more relative threshold, often used to assess sensitivity from GDSC data, we used z-score thresholds based on transforming log($IC_{50}$) values over all measured datapoints for specific drugs or cell lines. To identify sensitive and resistant cell line/drug pairs, we selected those with z-score < −1.0 and z-score > 1.0, respectively, with combinations between these values labeled as unknown (Supplementary Fig. 10b). The analysis identified trametinib-

treated MDST8, LS1034, and NCI-H508, and linsitinib-treated LS1034, NCI-H508, as well as alpelisib-treated HCT-15 cells as sensitive, while linsitinib-treated SNU-61, HCT-15, and HT115, as well as trametinib-treated NCI-H508 were identified as resistant. Surprisingly, trametinib-treated NCI-H508 was identified as both sensitive and resistant in different datasets (GDSC1 vs. GDSC2, respectively).

We generated phosphoproteomic profiles by DIA-based proteomics analysis of each cell line, at seven-time points (ranging from 5 min to 96 h) following perturbation with each of the seven inhibitors and vehicle control (DMSO) at the previously selected concentration (methods). This allowed assessing quantitative effects of KP-enzyme activity following short (5 min, 15 min), intermediate (1 h, 6 h), and long-term (24 h, 48 h, 96 h) treatment. Cumulatively 336 phosphoproteomic profiles were acquired by label-free DIA, for quantification and statistical validation at peptidoform-level[62] (Methods). We will refer to this dataset as "U54-DP". To minimize cross-sample statistical dependencies that would affect the mutual information estimator in dVESPA, we generated a reduced "U54-NET" dataset comprising only samples that were sufficiently separated in time, specifically the samples collected at 1 h, 24 h, and 96 h, respectively.

In total, 27,813 peptidoform precursors, 14,376 phosphosites, and 3786 phosphoproteins were identified and quantified at 1% global-context peptidoform and protein FDR[63] (Supplemental Data 8). Across all perturbations and time points, our workflow achieved high consistency on peptidoform-precursor level, on a cell line by cell line basis (48.7–55.6%), whereas the global completeness across all 336 runs of 36.6% indicates considerable biological inter-cell-line heterogeneity and different response to drug perturbations.

After data preprocessing—including normalization and missing value imputation (Methods)—we used VESPA to assess KP-enzyme differential activity in each cell line, at each time point, following treatment with each drug vs. vehicle control, using the integrated phosphostate (PL) and activity (AL) level analysis. The resulting matrices (Supplementary Fig. 11 (PL sorted), 12 (PL clustered), 13 (AL sorted), 14 (AL clustered)) represent the differential activity of 381 KP-enzymes across 336 sample conditions vs. vehicle control-treated, with positive and negative NES values indicating either increased or decreased enzymatic activity (Fig. 4a, Supplemental Data 9–11). As expected, cell line identity was the dominant factor in the unsupervised cluster analysis, when activity was computed at the phosphostate-level (PL-analysis) (Supplementary Fig. 12, Supplemental Data 10). This suggests that drug response is strongly dependent on the cellular state. However, as expected, when activity was assessed by activity-level (AL-analysis), unsupervised clustering improved stratification based on activation of different signaling pathways (Supplementary Fig. 14 (AL clustered), Supplemental Data 11), as assessed by Reactome enrichment analysis (Fig. 4b, Supplemental Data 12). This is consistent with the improvement of mVESPA activity inference when using the AL-level analysis, as already shown.

As a first-level validation, we assessed whether the primary (i.e., high affinity-binding) targets of each drug were differentially active in drug vs. vehicle control-treated cells. There are multiple caveats, however. First, the use of the maximum sublethal concentration is likely to induce only partial inhibition of the target protein; in addition, different mechanisms including pump, and feedback loops, may prevent target inhibition in resistant cells. We used VESPA to assess the time-dependent effect of each drug on its established high-affinity targets, as reported and specified in DrugBank[64] and ProteomicsDB[65] (Fig. 5, Methods). For drugs with > 5 primary targets, we selected the five with the highest average inhibition across all cell lines. The analysis confirmed that even though our experiment was designed for a different purpose, primary targets were inhibited for some drug and cell line combinations, albeit with highly variable temporal kinetics, ranging from 5 min to 96 h before maximum inhibition was achieved, potentially due to activation of cell adaptative mechanisms.

Further supporting the cell-line-specific effect of each drug, primary target inhibition across cell lines was highly variable even for the same drug. For instance, following ralimetinib treatment, activity of its high-affinity target MAPK13 was inversely correlated to that of MAPK14 in LS1034, MDST8, and SNU-61 cells yet positively correlated in other cell lines (Fig. 5). Critically, comparative analysis shows that abundance of phosphopeptides mapping to a drug's primary targets was often less informative than VESPA-measured KP-enzyme activity, often because sites determining enzyme activation were not directly measured or their measurement was noisy (Supplementary Fig. 15). In addition, changes in phosphosite abundance would only be relevant for enzymes that autophosphorylate.

Equally important, analysis of phosphosite-specific signalons provided critical clues for the identification of those determining enzyme activation. Most drugs inhibit enzyme function by binding to an enzymatically important part of the protein conformation rather than by modulating the phosphosite state directly; however, for kinases that auto-phosphorylate, the site determining its active vs. inactive state (*activating site*) would also be affected. Indeed, the analysis revealed that signalons associated with activating sites were often affected by the targeted inhibitors, while signalons associated with other sites were not affected (Supplementary Fig. 16, Supplemental Data 13). For example, MAP2K2:S222 phosphorylation was previously identified as an activating site[17]. Consistent with the literature, our data shows that trametinib-mediated MAP2K2 inhibition often resulted in lower S222-specific, time-dependent, VESPA-inferred activity. In contrast, the time series profile of MAP2K2:S23 was correlated with drug activity only in some cell lines (Supplementary Fig. 16). Interestingly activity of MAP2K1:S298—a distinct, previously reported activating site[17]—was anti-correlated with that of MAP2K2:S222, following trametinib treatment of HCT-15, HT115 and NCI-H508 cells, suggesting a cell line-specific compensatory mechanism. A similar pattern could also be observed for the correlation between MAPK14:Y182 activity and the activity of both MAPK13:S350 and MAPK13:T265, following ralimetinib treatment of HCT-15, HT115 and LS1034 cells (Supplementary Fig. 16). Additional established active sites targeted by specific drugs include EGFR:S991, EGFR:S1071 and EGFR:Y1092 (osimertinib), MAP2K1:S298 MAP2K2:S222, RIPK3:S227 (linsitinib) and INPPL1:S132 (imatinib)[17]. Taken together, these data show that VESPA analysis of data generated by drug perturbation assays can help elucidate subtype-specific drug MoA and cell adaptation mechanisms.

## Context-specific signaling network adaptation and rewiring

A primary goal of our experimental design was to study context-specific signaling network buffering/rewiring, as induced by drug treatment, to help elucidate mechanisms of cell adaptation. For this purpose, we combined VESPA-based inference of KP-enzyme activity with the DeMAND algorithm[59], a previously published methodology that was highly effective in identifying sub-networks dysregulated by a drug (Methods).

First, we used DeMAND to assess dysregulation of (a) the activity-level-based, CRC-specific SigNet—comprising 14,390 high-confidence interactions between 329 proteins— and (b) 915 high-likelihood (LR ≥ 0.5), non-phosphorylation-related interactions between 198 of the 329 proteins from the STRING database[66] (Methods). Indeed, since phospho-state may affect protein conformation and thus the ability to form complexes, it is reasonable to expect that integration of additional non-phosphorylation-related protein-protein interactions should further improve the analysis[67,68]. For each of the two network models, the DeMAND analysis was performed by replacing gene expression time series (as in the original implementation) with VESPA-assessed, KP-enzyme activity time series (Methods). Results from the two analyses were then integrated (Methods, Supplemental Data 14, 15).

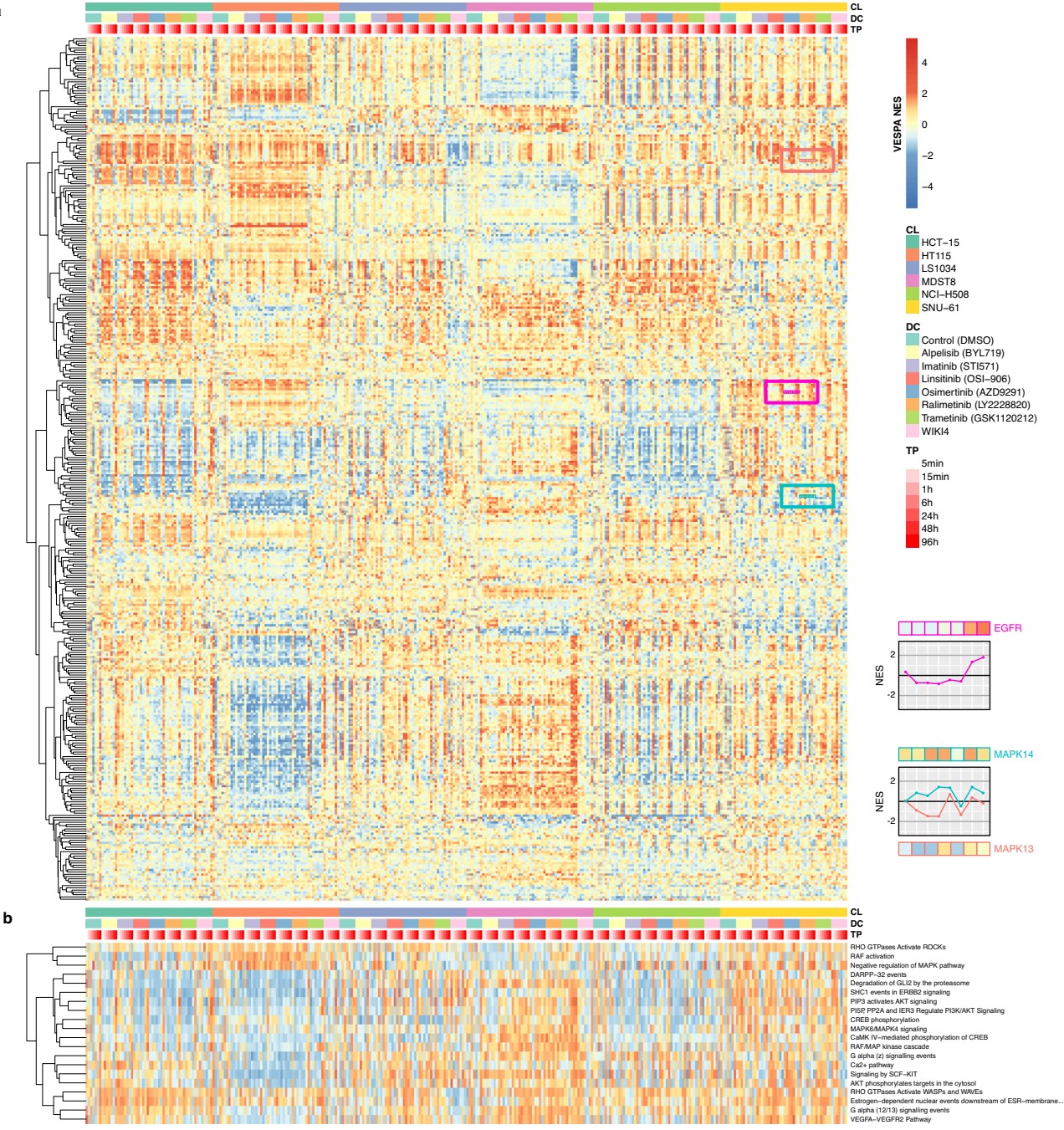

**Fig. 4 | Targeted drug perturbations in CRC cell lines. a** Visualization of VESPA's inferred normalized enrichment scores (NES) across the full drug perturbation dataset (336 samples), comprising six CRC cell lines (CL), 7 drug perturbations (Drug compound; DC) and vehicle control (DMSO), across 7 time points (TP). **b** Gene Set Enrichment Analysis (GSEA) using a signal transduction-specific subset of proteins in the Reactome database. Only terms significant in at least one sample (FGSEA ES-test two-tailed BH-adj. $p < 0.05$) are shown. The colors represent GSEA NES and are linked to the legend in **a**. Source data are provided as a Source Data file.

To assess both global (i.e., most conserved across all cell lines) and cell-line-specific drug MoA, two analyses were performed: For the former, we used data from all drug vs. vehicle control-treated cell lines, across all time points. For the latter, the analysis was performed on a cell line-by-cell line basis. The global analysis identified 62 proteins that were significantly dysregulated by the seven drugs ($p < 0.05$, BH-corrected), with an average of 12 to 21 proteins per drug (Supplemental Data 14). Hierarchical clustering of DeMAND-inferred MoA profiles identified cell lines presenting either congruent or divergent MoA for the same drug (Fig. 6a). Interestingly, some proteins—including established colorectal cancer risk factors, such as PRKCZ[69], BMP2K[70],

and MAPK14[71]—was highly dysregulated by virtually all drugs, across most cell lines, suggesting that the signaling logic of the cell plays a critical role in canalizing the effect of drug targeting distinct pathways.

To assess early vs. late effects of each drug, which may recapitulate potential cell adaptive mechanisms, we plotted the VESPA-assessed activity of the proteins identified as most dysregulated by DeMAND, at the early (5 min, 15 min, 1 h) (Fig. 6b) vs. late (24 h, 48 h, 96 h) (Fig. 6c) time points (Methods). As shown, for each drug, responses clustered into 1 to 3 sub-signatures (with most showing 2) indicating that drug response is mediated by distinct CRC-specific signaling networks. For instance, at the early time points, NCI-H508

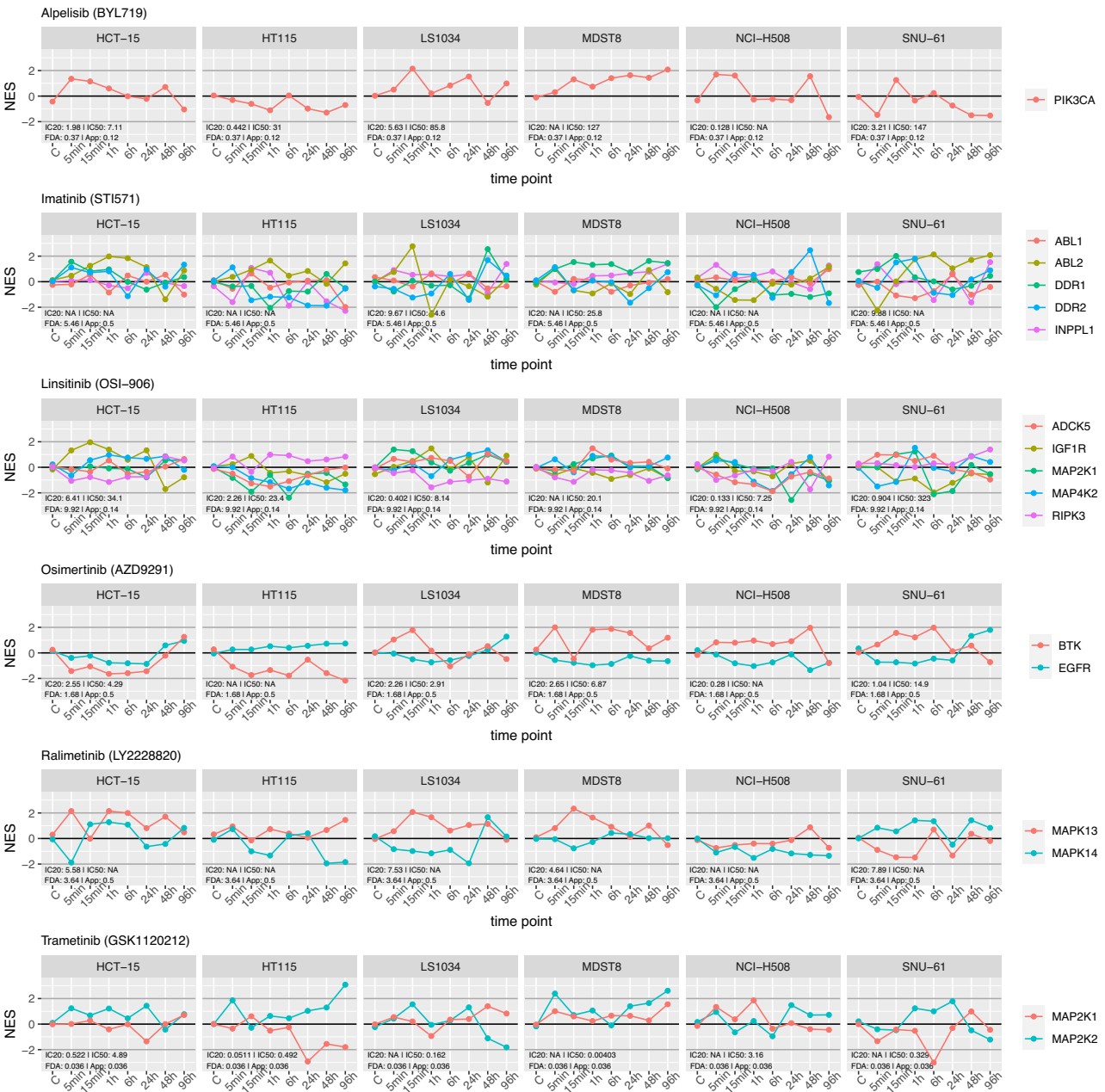

**Fig. 5 | Time-dependent response of known primary targets to drug perturbation.** VESPA normalized enrichment scores (NES) of the top 5 most downregulated proteins among known primary targets are visualized, and grouped according to drug perturbations and cell lines. Source data are provided as a Source Data file.

and LS1034, both classified as high-fidelity CMS$_2$ models, behaved similarly following 3 of the 7 treatments (imatinib, linsitinib, and ralimetinib) but not following the other 4.

As an illustrative example, two main clusters were identified in the early time points for osimertinib, including either NCI-H508, HCT-15, and HT115 (cluster 1) or MDST8, LS1034, and SNU-61 (cluster 2) (Fig. 6b). To illustrate how network rewiring affects drug MoA, we thus visualized the propagation of signaling activity dysregulation over time on the most drug-dysregulated sub-networks of HCT-15 and HT115, as representative of the two clusters (Fig. 6d). While the activities of key dysregulated proteins—BUB1, ERBB2, LYN, PRKCZ—are very similar at the early time points (Fig. 6b), they clearly diverge in HCT-15 and HT115 at the late time points (Fig. 6c). Their time course profiles show that activity of the primary drug target (EGFR) was not significantly affected, likely because it is not highly activated at baseline (Fig. 6a). However, for HT115, the established off-target[65] BTK was

significantly dysregulated, especially based on its interaction with ERBB2—a lower-affinity target of Osimertinib[57]—which is inactivated at the early time points in both cell lines, but re-activated at the 48 h and 96 h time points in HCT-15 cells. Similarly, the mitotic checkpoint serine/threonine kinase BUB1—which interacts with EGFR, BTK, ERBB2, LYN, and PTK6—was strongly activated in HCT-15 cells up to 24 h, suggesting that resistance/survival of this CRC cell line could be attributed, to some extent, to its increased signaling activity[72]. Together with the late time point activation of LYN and PRKCZ (Fig. 6d), these represent the main drug response differences between the two cell lines. Interestingly, LYN is an established mediator of EGFR inhibitor resistance, due to its involvement in EGFR's nuclear translocation[73]. In contrast, PRKCZ is mainly associated with cancer cell response to nutrient deprivation in intestinal tumorigenesis[74], suggesting that, following osimertinib treatment, HCT-15 cells undergo metabolic adaptation to induce drug-resistance.

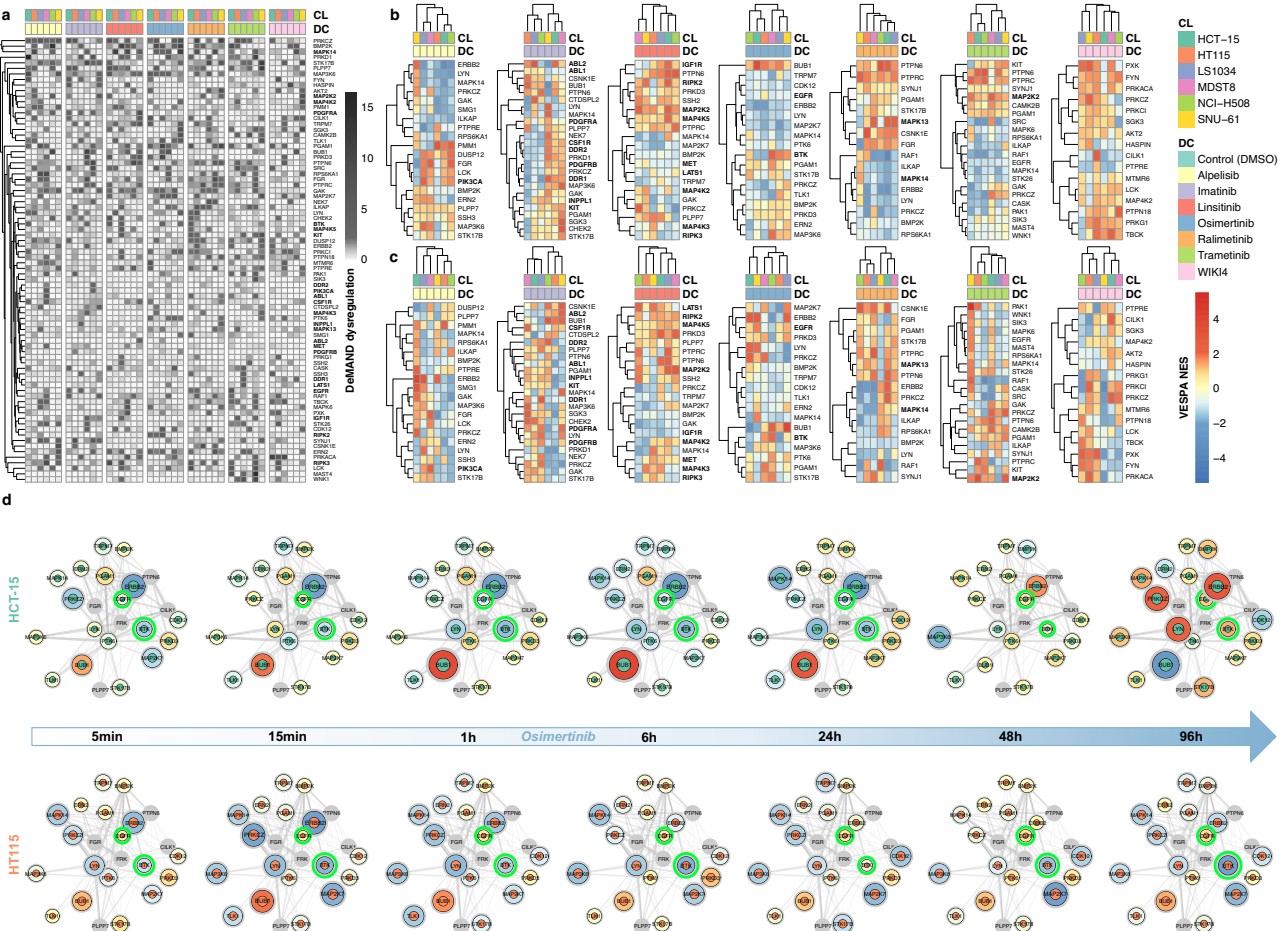

**Fig. 6 | Context-specific nature of signal transduction networks. a** Analysis of VESPA-inferred protein activity by DeMAND identifies KP-enzyme whose interactions with other proteins are most dysregulated across cell lines (CL) and drug compound perturbations (DC). The heatmap color scale represents the statistical significance of the DeMAND-assessed dysregulation (-log10(BH-adj. *p*); one-tailed empirical *p*-values computed by DeMAND). Only known drug targets (bold) and proteins with significant score (black: BH-adj. *p* < 0.05; one-tailed empirical *p*-values computed by DeMAND) in the cell-line-unspecific DeMAND analysis are visualized. **b** Dysregulated proteins described above were selected and grouped according to drug perturbations. The heatmap depicts VESPA-inferred activities of the aggregated early time points. **c** Dysregulated proteins described above were selected and

grouped according to drug perturbations. The heatmap depicts VESPA-inferred activities of the aggregated late time points. **d** Network dysregulation and drug mechanism of action (MoA) for the EGFR inhibitor osimertinib. Nodes indicate the most affected regulators with inner circle colors indicating cell line type and outer circle color and node size indicating VESPA activity. VESPA activity color legend is shared with subfigures **b** and **c**. Edges identify dysregulated, undirected interactions between KP-enzymes (Methods). Line thickness represents the statistical significance of each dysregulated interaction. Proteins highlighted in green indicate known primary/secondary targets. Source data are provided as a Source Data file.

At the early time points, ralimetinib also shows similar MoA across all cell lines. However, at the later time points, divergent response ensues in two cell line clusters, including NCI-H508, HT115, and LS1034 ($C_1$) and SNU-61, HCT-15, and MDST8 ($C_2$). As shown by two representative cell lines, HT115 ($C_1$) and SNU-61 ($C_2$), the primary ralimetinib targets (MAPK13 and MAPK14) show inverse temporal perturbation profiles, suggesting the emergence of critical cell adaptation mechanisms in $C_2$ cells (Supplementary Fig. 17). While MAPK14 inhibition in HT115 cells induced consistent inactivation of downstream MAPK targets at the later time points, MAPK13 inhibition in SNU-61 resulted in either activation or inactivation of downstream targets, likely due to negative feedback loop.

In summary, DeMAND analysis shows that sub-network dysregulation is subtype-specific and presents distinct temporal patterns, as also shown by graphical representation (Supplemental Data 16). Moreover, VESPA-assessed, time-dependent protein activity profiles, can be effectively used to investigate differential mechanism of action and cell adaptation mechanisms induced by either pre-existing,

context-specific signaling network wiring or by network rewiring (cell adaptation) following drug treatment.

## Cell adaptation-mediated drug resistance

Drug resistance mechanisms in cancer are among the most critical issues preventing the long-term efficacy of targeted drugs. While multiple studies have focused on the discovery of genetic events associated with drug-resistant clones[75], elucidation of dynamic network-based adaptation without clonal selection is emerging as a promising avenue to understand and modulate therapeutic efficacy[8,76].

VESPA-based activity analysis of drug perturbation time series can help investigate the adaptive response of kinases and phosphatases. We define cell adaptation as the dysregulation of signaling networks following drug perturbation in resistant vs. sensitive cell lines, as assessed at late time points (24 h, 48 h, 96 h) vs. vehicle control treated samples (Methods).

As previously shown, late-time-point effects were dominated by cell line identity (Fig. 7a, Supplemental Data 17,18). For instance, all

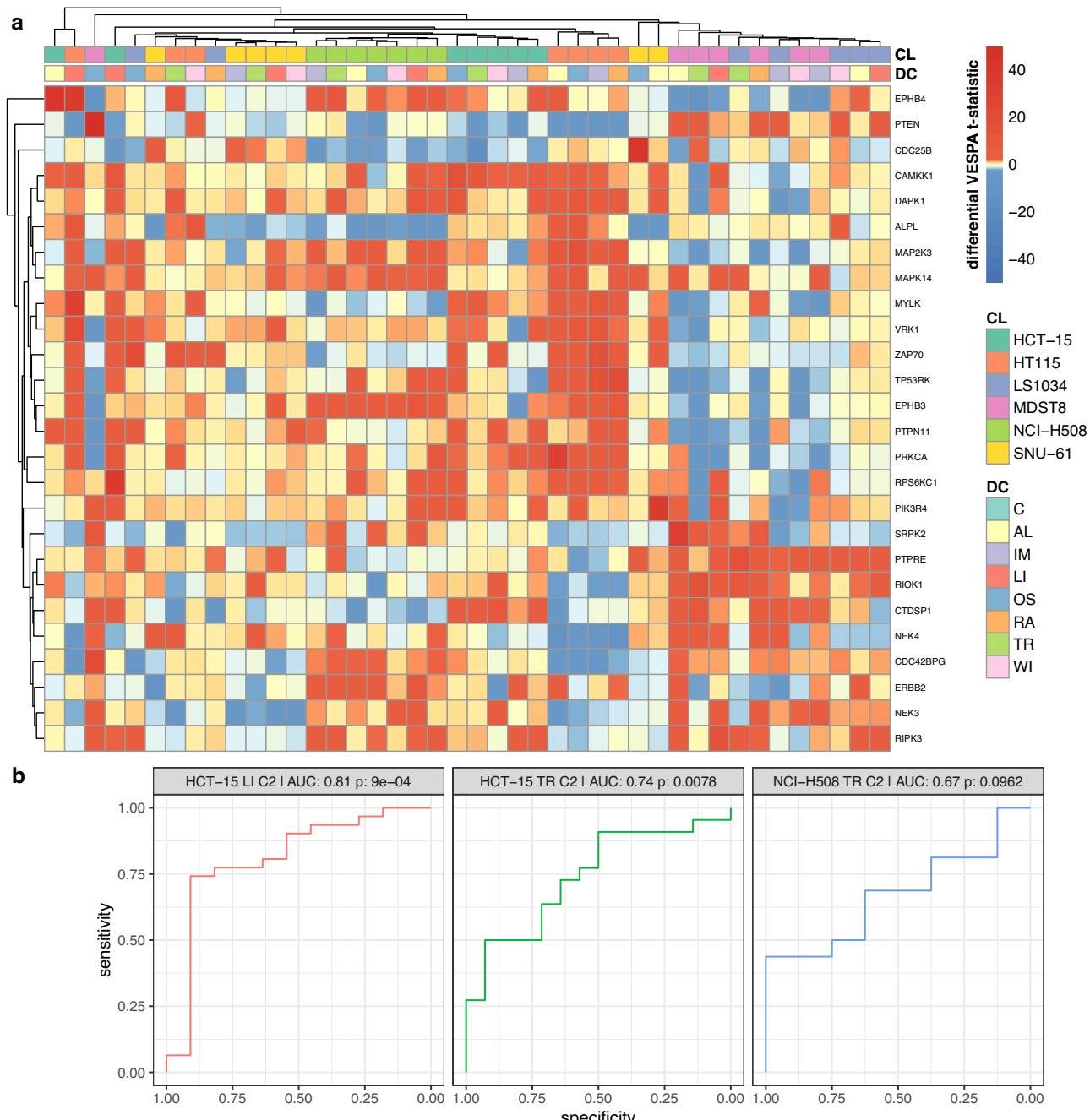

**Fig. 7 | Context-specific adaptive stress resistance mechanisms. a** Effect of drug perturbation vs. vehicle control at late time points is shown for each cell line (CL) and drug compound (DC). The effect is assessed based on the differential VESPA paired two-tailed *t*-test *t*-statistic between drug perturbations at three late-time-point (24 h, 48 h, 96 h) vs. vehicle control (DMSO) treated samples. Only statistically significant KP-enzymes are shown based on multiple-testing corrected $q < 0.05$ and avg(t-statistic) > 0, after integrating all *p*-values across all comparisons by Stouffer's method (Methods). **b** Enrichment of predicted KP-enzymes in proteins validated by CRISPRko assays is shown using receiver operating characteristics (ROC), area under the curve (AUC), and statistical significance (one-tailed Mann-Whitney-U test). Results are shown for HCT-15 cells treated with linsitinib (LI; C2: 4.0 μM) and trametinib (TR; C2: 0.7 μM) vs. DMSO, as well as NCI-H508 cells treated with trametinib (TR; C2: 0.01 μM) vs. DMSO perturbation. Source data are provided as a Source Data file.

drug treatments in MDST8 and LS1034 (resistant), except for osimertinib and ralimetinib treated LS1034 (sensitive), induced increased activity of a KP-enzyme cluster—including SRPK2[77], PTPRE[78], RIOK1[79], CTDSP1[80], NEK4[81], CDC42BPG[82], ERBB2[83], NEK3[84] and RIPK3[85]—previously associated with colorectal tumorigenesis and/or drug resistance. As such, association of several of these enzymes with the MAPK/ERK or STAT3 signaling pathways, as well as their inhibition by the EGFR inhibitor osimertinib and p38 MAPK inhibitor ralimetinib in

LS1034 cells, suggests that this protein cluster may be a key mediator of drug resistance.

A similar, cell line-specific cluster of activated and inactivated proteins was also observed in HT115 cells, following treatment with all drugs (resistant), except trametinib and WIKI4 (sensitive), including CAMKK1, DAPK1[86,87], MAP2K3[88], MAPK14[88], MYLK, VRK1[89], ZAP70[90], TP53RK[91], PTPN11[92], and RPS6KC1. Most of these proteins were previously associated with resistance mechanisms in colorectal cancer.

## Experimental Validation by CRISPR/Cas9-mediated Silencing

Three additional cell lines, HCT-15, NCI-H508, and SNU-61, also exhibited cell-line-specific responses to drug perturbations, albeit with less distinctive signatures. To validate the candidate resistance factors identified by VESPA and to systematically assess whether targeting of the predicted resistance factors would rescue chemosensitivity in insensitive cells, we performed a pooled CRISPR knock-out (CRISPRko) screen assay, targeting all annotated human kinases and phosphatases expressed in the cell lines used in our assays with four different guides per target gene (Methods, Supplemental Data 19). To select cell lines resistant to specific drugs, we used the previously described GDSC-based approach (Supplementary Fig. 10b, Methods): For linsitinib, we selected HCT-15 (z-score = 1.12) and SNU-61 (z-score = 1.55). However, the drug concentration in SNU-61 was too low to allow detecting statistically significant CRISPRko-mediated sensitization. As a result, data from this cell line was not included in the analysis. For trametinib, we selected HCT-15 (z-score = 0.89) and NCI-H508 (z-score = 1.13), even though the combination of NCI-H508 and trametinib resulted in discrepant sensitive (GDSC2) and resistant (GDSC1) responses within the two datasets.

To validate the predictions of proteins mediating cell adaptation and drug resistance, we performed CRISPRko screens in HCT-15 cells treated with linsitinib for 10 population doublings (C1: 1.0 μM, C2: 4.0 μM) and trametinib (C1: 0.1 μM, C2: 0.7 μM), as well as in trametinib treated NCI-H508 cells (C1: 0.005 μM, C2: 0.01 μM). DMSO was used as vehicle control to assess guide RNA (gRNA) depletion. The initial (drug/DMSO-free) time point-samples (T0) for these screens were collected approximately 5–7 days after the sgRNA lentiviral transductions and puromycin selection. To pick the correct drug concentrations for the pooled CRISPRko screens, we performed a long-term (10 population doublings) growth test for each cell line and their corresponding drug(s) with multiple different drug concentrations (Methods). For the CRISPRko screens, we picked two drug concentrations for each cell line, which appeared to only have a perturbation, but not a full inhibition effect, analogously to the phosphoproteomic perturbations (Methods). The only exception was the cell line NCI-H508, where we had to use a lower drug concentration for the long-term pooled CRISPRko screening, due to drug toxicity manifesting after 96 h time point (last time point of the short-term assay). Differential sgRNA abundance analysis was performed using DESeq2 (Methods, Supplemental Data 20). Sequencing quality was excellent, with an average alignment ratio of 90.98% (Supplementary Fig. 18). Differential expression analysis of DMSO vs. T0 samples identified known essential genes for CRC with an area-under-the-curve (AUC) of 0.96 for both NCI-H508 and HCT-15 (Supplementary Fig. 19, Methods).

For tumor suppressors that can also act as resistance or insensitivity factors, such as DAPK1 or PTPN11, the nature of perturbation or knock-out will substantially bias their activity and function[93]. It was recently suggested that tumor suppressor genes, or genes whose knock-out imparts a growth advantage on cells, could cause recurrent drug suppressor hits in drug-gene interaction CRISPRko screens, and thus a source of a systematic bias and false positives in drug-perturbed CRISPRko screens[93]. There is thus a potential discrepancy in the experimental design of the VESPA predictions and the CRISPRko experiment, where VESPA predicts KP-enzyme late-timepoint activity and potential involvement in resistance or insensitivity mechanisms, whereas the CRISPRko experiment assesses their gene essentiality starting from timepoint 0 in combination with drug perturbations for altogether 10 population doublings. For this reason, we excluded knock-outs of known tumor suppressors[94] from the analysis (Supplementary Fig. 20-21, Methods). Other confounding factors, specifically the involvement of proteins in cell regulatory mechanisms outside of the scope of their primary KP-enzyme function characterized by VESPA, could also explain the bias of these comparisons, specifically a proportion of the false negative predictions. For example, MAP3K7 was found to have both lower VESPA activity and phosphoprotein abundance, while being an essential gene. This discrepancy could potentially be explained by the centrality of MAP3K7 as regulator of cell death, being involved in both NF-κB and in NF-κB-independent pathways such as oxidative stress and receptor-interacting protein kinase 1 (RIPK1) kinase activity-dependent pathways[95].

To compare candidate resistance factors predicted by VESPA with the ground truth from CRISPRko assays, we conducted separate analyses for each cell line and drug perturbation using receiver operating characteristics (ROC) (Methods). Gene essentiality (log-fold-change perturbation vs. control, including negative (i.e., essential) and positive (i.e., non-essential) values), is expected to be inversely correlated to VESPA-assessed activity (t-statistic perturbation vs. control; positive: increased activity, negative: decreased activity).

The analysis strongly supports the relevance of VESPA's predicted resistance factors in combination with the drug perturbations (Fig. 7b). ROC was found to be particularly significant for HCT-15 perturbed by linsitinib and trametinib (AUC = 0.81, p = 9e−04; AUC = 0.74, p = 7.8e−3, respectively), yet lower significance for trametinib treated NCI-H508 cells (AUC = 0.67; p = 0.0962), potentially caused by the differences in drug concentrations in the two CRISRP-ko experiments (C1: 0.005 μM, C2: 0.01 μM) vs. the drug perturbation assays used to generate the phosphoproteomic profiles ($C_{max}$: 0.036 μM). Correlation analysis further shows that VESPA can identify high numbers of true positive candidates with only a few false positives (Supplementary Fig. 21), an essential requirement for diverse applications.

We further used the CRISPRko validation experiments to assess the VESPA-DeMAND-predicted resistance factors, as well as measured phosphoprotein abundances. While the VESPA-DeMAND-predicted resistance factors achieved almost similar performance to the results obtained only by VESPA (Supplementary Fig. 22,23), we found measured differential phosphoprotein abundance to not be predictive or correlate with the CRISPRko validation experiment, supporting the increased predictive power of VESPA inferred K/P-enzyme activities over phosphoprotein abundances (Supplementary Fig. 24,25).

In summary, VESPA analysis of phosphoproteomic time-series following drug treatment was effective in identifying candidate resistance factors that could be exploited in combination therapy approaches.

## Discussion

Most drug targeting kinases or phosphatases fail due to the cell's ability to implement an adaptive response that re-wires the underlying signaling network to buffer the drug effects. Compared to recent studies on these mechanisms focusing on adaptation to a specific target[96], the aim of this study is to introduce a methodological approach to study and validate the context-specific wiring and time-dependent, drug-mediated adaptive re-wiring of signaling networks across different subtypes of a specific tumor and in response to drugs targeting multiple targets. While the study is focused on CRC, it is designed to be fully generalizable to other tumor and non-tumor-related context, limited only by data availability.

To accomplish these goals, we complemented large-scale, tumor-specific phosphoproteomic profile repositories generated by CPTAC with a comprehensive experimental design to generate *perturbational phosphoproteomic profiles* from six CRC cell lines representing distinct tumor subtypes, at seven time points following treatment by seven targeted drugs and vehicle control. Compared to other recent studies, e.g. profiling 60 inhibitors against three diverse cell lines[20], or investigating 31 cancer drugs in 13 diverse cell lines at multiple drug concentrations and time points[21], the focus of our study was to create a highly focused dataset allowing quantitative elucidation of cell adaptive response across multiple CRC subtypes—as recapitulated by selected cell lines—drugs, and time points. Such a large-scale assay required a flexible and scalable approach. The recent development of

new data-independent acquisition (DIA) strategies[97,98] and corresponding computational analysis methods[62,99,100], provided an opportunity for the comprehensive and consistent quantification of the phosphoproteomic profiles, requiring less than 3 weeks of instrument time, confirming the scalable nature of the proposed methodology. Although this unfractionated, label-free approach provides substantially lower coverage compared to fractionated, label-based CPTAC studies, we reasoned that, for the specific questions addressed in this study, the quantitative consistency of the sample set may be more important than the depth of proteome coverage. Further, we design an algorithm, VESPA, that can leverage signaling networks inferred from the comprehensive CPTAC datasets to support improved analysis of focused drug perturbation profiles.

Borrowing from previous approaches[23,24,39,67,68,101], VESPA postulates that kinase and phosphatase activity is better measured based on the phosphostate of their substrates than on their own phosphostate. However, as discussed, critical changes were necessary to adapt this framework to analyzing highly sparse and noisy phosphoproteomic profiles, including a reformulation of the Data Processing Inequality approach used to remove a majority of indirect signaling interactions.

Compared to established pathway databases, VESPA dramatically increases the number of KP → S interactions per signaling protein (e.g., going from an average of 70 in Pathway Commons to an average of 500 by dVESPA analysis) and was able to generate signalons appropriate for activity measurement for almost twice the number of KPs in Pathway Commons (i.e., 371 vs. 211). Critically, for several KPs, the activity could be assessed in phosphosite-specific fashion—thus improving mechanistic understanding of signaling transduction—and could be corrected for signaling crosstalk. Cross-talk represents a critical property of cellular signaling, which can only be addressed with the context-specific, comprehensive signaling networks generated by VESPA and its analytical framework based on the original VIPER algorithm[29]. Finally, the hierarchical approach in VESPA significantly improved the assessment of tyrosine kinase activity, by addressing the reduced sensitivity of phosphoproteome profiling methods to tyrosine phosphorylation. The use of methods for phosphotyrosine pull-down may further improve VESPA's performance.

Overall, extensive benchmarks, including at the level of the individual algorithmic improvements, show that even foregoing the use of dVESPA-inferred signaling networks, VESPA significantly outperforms previously published methodologies. Such basic implementation, however, is associated with a lower limit on algorithm performance, since we also show that use of de novo signaling networks dramatically improves performance.

Although VESPA is applicable to most CPTAC or DIA-based datasets, several requirements must be fulfilled to make use of its full potential: Phosphoproteome coverage (>10,000 phosphosites), quantitative consistency (>40%) and sufficient sample number for network reconstruction (>100 independent samples) are typically required.

Selected cell lines for perturbational profile generation effectively recapitulate the major CRC subtypes identified by either transcriptomics or phosphoproteomic CRC sample analyses, in TCGA and CPTAC, respectively (Fig. 3a, b), as well as the subtypes reported by the Consensus Molecular Classifier (CMS) of the Colorectal Cancer Subtyping Consortium (CRCSC). Consistent with this selection, mechanisms of adaptive response stratified with cell lines representing the same or most related subtypes.

Further showcasing the flexibility of the proposed framework, we leveraged drug perturbation profiles for three different purposes, (1) determining temporal activity dynamics of established high-affinity drug targets, (2) assessing context-specific wiring/re-wiring of signaling pathways following drug perturbation, and (3) identifying context-specific adaptive stress resistance or insensitivity mechanisms. The temporal activity analyses showed that phosphosites of primary

targets can rarely be measured consistently or fail to show a direct response. In contrast, VESPA-inferred signaling activity could, in some cases, even resolve the activity associated with phosphorylation of individual phosphosites (Supplementary Fig. 16).

DeMAND-based network dysregulation analysis further illustrates the value of using context-specific signaling networks. Based on VESPA's ability to dissect signaling networks de novo, in context-specific manner, DeMAND was able to provide a more direct, mechanism-based assessment of drug-mediated signaling network dysregulation and thus of adaptive responses mediated by other KP-enzymes compared to the original implementation (Fig. 6d, Supplementary Fig. 17).

Differential analysis of late vs. early time point KP activity effectively identified candidate proteins mediating adaptive response and drug resistance, thus providing valuable clues for pharmacologic targets that may rescue drug sensitivity (Fig. 7a). While many of the proteins identified by the analysis had already been validated as resistance factors in CRC, CRISPRko assays targeting kinases and phosphatases confirmed significant enrichment of algorithm predictions in proteins representing causal determinants of drug resistance. This suggests that VESPA may provide a useful tool to elucidate mechanisms underlying drug resistance and cell adaptation.

Although our study focuses on the phosphoproteomic profiles, the signaling activities inferred by VESPA are ideally suited and directly compatible with upcoming methods for causal integration of multiomic profiles, e.g. via TieDIE[68] or COSMOS[67]. In addition, the methodology is fully generalizable and can be used to generate SigNets for many tumor contexts that have already been characterized by CPTAC and related studies. VESPA is directly compatible with popular upstream bottom-up proteomic workflows and can be easily adapted for various experimental designs. The algorithmic components are available as platform-independent open-source software under a non-commercial usage license. We further provide de novo inferred signaling networks and inferred kinase and phosphatase activities for a variety of published CPTAC datasets (see Data Availability).

## Methods
### VESPA

**Data preprocessing.** The primary input to the VESPA algorithm is a set of quantitative, proteotypic/unique peptide-level phosphoproteomic profiles from bottom-up mass spectrometry experiments. The data format is a matrix (hereafter referred to as Proteomic VESPA input Matrix, PVM), whose columns represent: (a) the "gene_id" (UniProtKB entry name without species, e.g. "EGFR"), (b) the "protein_id" (UniProtKB entry identifier, e.g. "P00533"), (c) the "peptide_id" (free text unique peptide identifier from upstream software), (d) the "site_id" (unambiguous combination of gene_id, protein_id and phosphosite, separated by ":", e.g. "EGFR:P00533:S229"), (e) the "modified_peptide_sequence" (free text modified peptide sequence from upstream software), (f) the "peptide_sequence" (free text unmodified peptide sequence from upstream software), (g) the "phosphosite" (unambiguous phosphosite identifier, e.g. "S229"), (h) the "run_id" (free text sample or MS run identifier), and (i) the "peptide_intensity" (float log2-transformed peptide intensity from upstream software) of each detected peptide/phosphopeptide. To avoid any ambiguities and to allow for data transferability, all peptide sequences, phosphosites and protein names and identifiers are expected to be mapped to UniProtKB. If a phosphosite is represented by multiple peptide precursors, the most consistently detected peptide precursor is used. If a peptide precursor contains multiple phosphorylated sites, redundant entries for each distinct phosphosite are added. Each dataset (e.g., CPTAC sample cohort or study) should be stored in a separate PVM to ensure that differences in experimental design or batch effects can be accounted for in downstream steps.

Protein abundance level normalization can be very important for clinical datasets, such as CPTAC, where some proteins might have

more variable protein abundance distributions, which could potentially confound corresponding phosphopeptide abundances and thus lead to wrong associations of co-regulated proteins. VESPA can incorporate protein abundances at three different stages of the workflow:

**Protein abundance normalization.** Of note, as opposed to CPTAC samples, we did not measure baseline peptide abundances suitable for protein abundance inference for the U54 samples, since this would have doubled the number of required LC-MS/MS runs. However, by normalizing each drug-treated sample against the corresponding vehicle control-treated ones, we expect that this will not significantly affect results. Further, since most signalons comprise of phosphosites representing dozen to hundreds of independent phosphoproteins, the analysis is robust against changes in protein abundance of individual proteins. As such, we recommend using protein abundance-normalized profiles only if the investigated mechanisms are expected to be substantially confounded, e.g. the auto-phosphorylation feedback loops of tyrosine kinases when comparing drug-perturbed to baseline samples.

**Protein abundance as proxy for KP-enzyme signalons.** Instead of using KP-enzyme phosphopeptides as proxy for the enzyme component of the signalon, protein abundances can optionally be used for the inference of signaling networks. This is particularly useful when variability in gene expression influences the activity of a KP-enzyme, for which phosphopeptides were not measured, for example, tyrosine kinases. This mode should thus only be used instead of "protein abundance normalization" described above, but not together. The file format is similar to PVM; however, the columns "peptide_id" (free text unique protein identifier), "site_id" (unambiguous combination of gene_id, protein_id and "PA" (protein abundance), separated by ":", e.g. "EGFR:P00533:PA"), "modified_peptide_sequence" (free text unique protein identifier), "peptide_sequence" (free text unique protein identifier), "phosphosite" ("PA" (protein abundance)), and "peptide_intensity" (float log2-transformed protein intensity from upstream software) are different.

**Signalon optimization.** Since protein abundance measurements are themselves noisy and thus have limited accuracy, the normalization step may introduce additional bias. Since dVESPA supports the use of de novo, inferred signalons, both protein-abundance normalized and unnormalized phosphoproteomic profiles can be included in the network dissection step, thus allowing optimal signalon selection on an individual KP-enzyme basis.

The "vespa" R-package provides fully automated import functionality for the OpenSWATH[102], IonQuant[103], MaxQuant[104], and the CPTAC[105] file formats. Support for other file formats can be easily added by supporting their reference implementation. During data import, peptide sequences are first mapped to a user-provided UniProtKB/SwissProt FASTA database to ensure consistent mapping of phosphosites and identifiers. Peptide intensities are (optionally batch-wise) quantile normalized and centered.

**Signaling Network Inference.** Bottom-up proteomic experiments spanning dozen to hundreds of samples are affected by both biological variability and technical noise. For phosphoproteomic profiles, accounting for technical noise and artifacts is especially challenging, because different sample preparation workflows, phosphopeptide enrichment strategies, labelled or label-free quantification, biochemical peptide fractionation, data acquisition techniques, and signal processing can all have dramatic effects on critical variables—e.g., phosphoproteome coverage, depth, and consistency—resulting in missing values. As a result, consistent with the assumptions for mutual information estimation, different datasets cannot be combined to

generate an integrated signaling network. Rather each dataset must be analyzed independently to avoid different biases in different datasets from introducing massive technical artifacts. Networks are generated via the Snakemake[106] workflow ("vespa.net") consisting of the "vespa" and "vespa.db" R-packages and the "vespa.aracne" algorithm. This includes the following steps:

**Data preprocessing.** The PVM is first transformed to a peptide-level quantitative matrix, with missing values designated as NA, and then rank-transformed, in a peptide-wise manner, while retaining all missing values. To restrict the number of potential interactions between KP-enzyme and substrate phosphopeptides, several options are available: (a) a list of KP-enzymes is used to define KP-enzyme and substrate phosphopeptides, where all combinations between them are allowed, (b) a list of activating (kinases) and deactivating (phosphatases) KP-enzymes is used, where kinases require positive correlation with substrates and phosphatases reiquire negative correlation, and c) reference network, where a list of KP-enzyme/substrate interactions, with optional priors, is supplied based on the literature or other algorithms. For options (a) and (b), an additional, optional list of candidate substrates can be supplied if a subset of phosphopeptides should be ignored by the analysis.

**Mutual information estimation by hybrid adaptive partitioning.** Peptides measured by bottom-up proteomics have individual limits of detection (LOD) and limits of quantification (LOQ), usually resulting in censored values. Missing values in some samples might thus arise not due to technical effects (e.g., stochastic data-dependent acquisition or batch effects) but can contain information about those peptides not reaching LOD/LOQ abundance levels. To make use of this information and to estimate mutual information (MI) between two sparse abundance rank vectors representing the phosphopeptide abundance of a KP-enzyme (KP) and a substraite (S) across multiple samples, a hybrid adaptive partitioning algorithm was implemented in "vespa.aracne". Specifically, the space defined by the two vectors is split into four quadrants containing: 1. All data points with no missing KP and S values, 2. data points missing both KP and S values, 3. data points missing only KP values, and 4. data points missing only S values. For quadrant 1, MI is estimated by an adaptive partitioning algorithm (ARACNe-AP)[28]. For quadrants 2-4, MI is estimated separately without adaptive partitioning. The MI of all quadrants is then combined and normalized, providing a more robust metric to assess the relationship between KP-enzymes and targets.

**Selecting a statistical significance threshold for mutual information.** To estimate a threshold for statistical significance, a null model is generated by permuting the rank-transformed quantitative peptide matrix, including missing values. The MI probability density for all candidate interactions is computed and an MI threshold for a user-definable family-wise error rate (default: FWER = 0.05) is estimated.

**Bootstrapped network reconstruction.** For each candidate KP-enzyme/substrate interaction MI is computed by bootstrapping the Hybrid Adaptive Partitioning estimator over $N$ random samplings (default: $N = 200$) of the PVM matrix and removing interactions with MI below the statistical significance threshold at each bootstrap step. To remove putative indirect interactions, the Data Processing Inequality (DPI)[27] or its signal transduction-specific version (stDPI) can be applied.

**Signal Transduction Data Processing Inequality (stDPI).** The molecular mechanisms involved in phosphorylation-based signaling networks can be very diverse and involve changes in phosphostate, binding to activating proteins, allosteric activation, among many others. In contrast, standard serine/threonine phosphopeptide enrichment and bottom-up proteomics, as used for example in CPTAC

 

studies, can only be used to measure sparse phosphopeptide abundances for some proteins. To account for this limitation, we implemented stDPI, a more biochemically constraint version of DPI. Specifically, this mode assumes that for kinases, phosphopeptide or protein abundance must positively correlate with substrate abundance. On the other hand, phosphatase abundance must negatively correlate with substrate abundance. This permits only two out of four possible DPI "triangles" to be valid for assessment, including (i) kinase-kinase-substrate or (iv) phosphatase-kinase-substrate relationships, but not (ii) phosphatase-phosphatase-substrate or (iii) kinase-phosphatase-substrate, as shown in Supplementary Fig. 1b. This is because (ii) and (iii) would have an opposite effect on the substrate. If we assume for example that a phosphatase dephosphorylates its substrate (direct), a putative indirect phosphatase-phosphatase-substrate interaction would have an inverted effect on the substrate. It should be noted though that the assumptions of stDPI are tailored to standard serine/threonine phosphopeptide enriched data, where the phosphate groups involved in molecular mechanisms are typically not directly measured, e.g. as is the case for tyrosine kinases and phosphatases. If tyrosine-enriched phosphopeptide measurements are available, standard DPI should be used instead, as this will also allow for other mechanisms, including phosphatase activation by dephosphorylation, resulting in positive correlation between tyrosine phosphatase and their substrates.

**Consensus network generation.** Finally, a consensus network is generated from the individual bootstrap runs as introduced in ARACNe-AP. Specifically, the statistical significance of each interaction is estimated based on a Poisson distribution generated from all bootstrap runs and only statistically significant interactions are retained (default: $p < 0.05$, Benjamini-Hochberg-corrected, BH). Two networks are generated by the analysis, one where individual phosphosite-phosphosite interactions are considered and the other where the abundance of all phosphosites in the same protein are combined.

**Signalon generation.** Based on the final consensus network generation, the set of substrates (full proteins or individual phosphosites) regulated by a KP-enzyme (signalon) is generated for further use by the mVESPA algorithm, which extends the VIPER algorithm[29] to signaling networks. In this step, peptide identifiers are mapped back to site identifiers to ensure transferability between different datasets. For each interaction, a probabilistic weight is computed by normalizing its estimated MI by the maximum MI estimated across the entire network. For each interaction, optional priors from reference networks are normalized by the maximum prior specific to each KP-enzyme. The mode of regulation is then determined as described previously[29] by fitting a three-Gaussian mixture model, representing repressed (Spearman $\rho \ll 0$), activated ($\rho \gg 0$), and non-monotonically regulated ($\rho \cong 0$) targets. Spearman's correlation coefficient is computed using only fully quantitated datapoints. Finally, signalons are trimmed to include only the top N (default: $N = 500$) substrates, based on their probabilistic weight, until the threshold T is reached, optionally weighted by a reference network's priors:

$$T = \sum_{1}^{N} \frac{likelihood}{\max(likelihood)}^2 \tag{1}$$

Only signalons with at least M substrates (default: $M = 5$) are used by mVESPA.

**Activity-level network reconstruction.** So far, phosphostate-level signalons were generated using stDPI, associating KP-enzyme and substrate phosphopeptide abundances. mVESPA can then be used to infer phosphostate-level activity using these signalons. For VESPA's hierarchical approach, additional activity-level networks are reconstructed as described above for phosphostate-level networks, however now using the phosphostate-level activities inferred in the previous step as input instead of phosphopeptide abundances. Further, instead of stDPI, standard DPI is used to abstract the second signaling network to a functional instead substrate-based representation of the system.

## KP-enzyme Activity Inference

**Phosphostate-level inference.** To infer KP-enzyme activity based on the phosphostate of their substrates (phosphostate-level analysis), we use signalons generated by phosphostate-specific dVESPA analysis and either the PVM used for their generation or an independent PVM comprising a phosphoprofiles from a set of context-related samples, as the main input to the "viper"[29] R-package. Signalons must comprise at least M substrates (default: $M = 5$) to be considered for the analysis. The PVM matrix can be divided into a set of samples for which differential KP-enzyme activity must be assessed compared to a second set of control samples. Alternatively, the entire PVM matrix can be used as a control, thus assessing differential KP-enzyme activity compared to the centroid of the entire sample set. First, the PVM is transformed to a quantitative matrix, with missing values imputed as row-wise minimum with the addition of random values from a white noise distribution (R-package "jitter", range set to the difference between the two lowest values per row) to break ties due to identical values. The parameters for the "viper" activity inference function can be tailored for different applications and support the same experimental designs as the original implementation. A bootstrapped null model can be used, using the "viperSignature" function, to assess differential protein activity in each specific sample compared to the reference dataset (i.e., either a subset of the PVM or the entire PVM).

**Activity-level inference.** The same analysis as described in the previous paragraph is performed with the following differences: (a) activity-level signalons, generated as described in previous sections, are used and (b) the PVM matrix is replaced by a matrix of phosphostate activity levels, as inferred from the first analysis.

**Integrated inference.** Phosphostate- and activity-level activities, as assessed by mVESPA, are then integrated using Stouffer's method.

**Crosstalk correction.** Signalons for two KP-enzymes may present substrate overlap, thus resulting in situations where activation of one enzyme may result in the other enzyme also appearing activated. To address this challenge, VESPA leverages the pleiotropy correction[29] originally introduced in the VIPER algorithm and included in the "viper" function[29]: All signalon pairs affected by cross-talk are generated that fulfill two conditions: Specifically, consider two KP-enzymes, A and B, whose signalons comprise shared substrates and are significantly enriched ($p < 0.05$) in a phosphopeptide abundance signature of interest. In that case, the contribution of the shared substrates to the activity of the KP-enzyme with the lower differential activity (e.g., B), is assessed by computing:

$$CDE = \log_{10}(pB) - \log_{10}(pA) \tag{2}$$

$CDE$ is penalized by $CDE^{CI/NT}$, where the cross-talk index (CI) is a constant (default: $CI = 20$) and NT is the number of signalon pairs where signalon A is one component and vice versa.

**Signalon optimization.** VESPA signalons are typically generated based on the analysis of multiple dependent or independent phosphosites, one more phosphoprofile datasets, and potentially using different priors from reference databases or predictive algorithms. To select the best signalon for each phosphosite and/or protein, we use the approach introduced by the metaVIPER algorithm[33], where the

signalon producing the highest differential activity is selected, based on the assumption that incorrect signalons can only reduce the NES computed by the enrichment analysis.

Phosphopeptides can frequently harbor multiple phosphosites, signalons generated at a site-specific level can be redundant. To generate a non-redundant set, VESPA identifies and removes highly correlated signalons associated with the same phosphopeptide using the "findCorrelation" function from the R-package "caret" with a specified correlation cutoff (default: $C = 0.5$).

**Integrated generation of signalons on phosphostate- and activity-level.** The "vespa.net" Snakemake workflow automates the process of implementing all the above-described steps to generate optimized phosphostate- and activity-level signalons starting from one or more input PVMs (and optionally related protein abundance matrices). As discussed, the analysis requires an additional PVM representing a reference phosphoproteomic dataset with respect to which the differential activity is assessed. For instance, the samples representing vehicle control-treated cells can be used as a control set to assess KP-enzyme differential activity in drug-treated samples. In alternative, the entire PVM matrix can be used as a reference dataset.

**Application to target datasets.** After running "vespa.net" and generating phosphostate- and activity-level signalons, the "viper" function of "vespa" is used to compute KP-enzyme activity based on the inferred network. These frameworks provide a flexible toolkit suitable to several applications, as discussed in the main text. The tutorial dataset ("vespa.tutorial") illustrates the use cases of this study and describes the required parameters.

## Cell culture

The six CRC cell lines used in this study (HCT-15, HT115, LS1034, MDST8, NCI-H508, SNU-61) were previously selected to ideally represent the clinical phenotypes covered by TCGA as assessed by their transcriptional state inferred by VIPER, while also fulfilling practical culture condition considerations[45]. The cell lines were obtained from ATCC (American Type Culture Collection) (HCT-15: ATCC#CCL-225, LS1034: ATCC#CRL-2158, NCI-H508: ATCC#CCL-253), the Korean Cell Line Bank (KCLB) (SNU-61: KCLB#00061), and the European Collection of Authenticated Cell Cultures (ECACC) (MDST8: ECACC#99011801, HT115: ECACC#85061104) and cultured using prescribed conditions to the amounts as described below. No authentication was conducted after purchase from the vendors. All cell lines were routinely tested for *Mycoplasma* contamination and were kept in a 37 °C humidity-controlled incubator with 5.0% $CO_2$.

## IC_{20} determination

As discussed, to avoid off-target effect and activation of stress and cell death pathways, that may confound the analysis of a drug's mechanism of action, cells were treated with a drug concentration representing its 48 h $IC_{20}$. To assess this value, cell lines were first plated into 384-well plates, in 50 μL total volume, and incubated at 37 °C. After 16 hours plates were removed from the incubator and compounds were transferred into assay wells (100 nL) in triplicate, according to a 10-point dilution curve starting at 10 μM. Plates were then returned to the incubator. After 48 hours plates were again removed from the incubator and allowed to cool to room temperature prior to the addition of 100 μL of CellTiter-Glo (Promega Inc.) per well. Plates were then mechanically shaken for 5 minutes prior to readout on the EnVision Multi-Label Reader (Perkin Elmer Inc.), using the enhanced luminescence module. Relative cell viability was computed using matched Thimerosal control wells as reference. $IC_{20}$ was estimated by fitting a four-parameter sigmoid model to the titration results. The high-throughput screening table for the $IC_{20}$ screen is available in Supplemental Table 1.

## Drug perturbation profile generation

Each cell line was treated with seven different drugs, as well as vehicle control (DMSO). Each cell line was plated in 6-well plates in numbers that would approach confluency by 96 h for the fastest-growing cell line. After allowing overnight attachment, cells were treated with each drug at a concentrations C selected to be (a) $C \le 0.5\ \mu M$, (b) $C \le C_{max}$, the maximum approved drug concentration, and (c) $C \le IC_{20}$ value of the most sensitive cell line in the panel, as discussed above: Based on this logic, the following concentration were used: alpelisib (BYL719): 0.12 μM, imatinib (STI571): 0.5 μM, linsitinib (OSI-906): 0.14 μM, osimertinib (AZD9291): 0.5 μM, ralimetinib (LY2228820): 0.5 μM, trametinib (GSK1120212): 0.036 μM, WIKI4: 0.5 μM. DMSO was titrated at 0.5%. Cells were then harvested at multiple time points, including 5 min, 15 min, 1 h, 6 h, 24 h, 48 h, and 96 h, lysed and processed as described below for the generation of phosphoproteomic profiles. Each sample was run in triplicate. To generate baseline (i.e., untreated) phosphoproteomic profiles, cell lines were grown in 150 mm × 25 mm dishes to about 80% confluency and split into 3 batches. At the time of harvest, cells were washed 3x with PBS, pelleted, snap-frozen by liquid nitrogen, and stored at -80 °C.

## Proteomic sample preparation

For frozen cell pellets, cells were lysed on ice, by adding 10 M urea containing a complete protease inhibitor cocktail (Roche) and Halt™ Phosphatase Inhibitor (Thermo); pellets were then resuspended and processed for tryptic digestion. For cells in 6-well plates, plates were washed 3x with pre-cooled PBS and cells in wells lysed on ice immediately in 10 M urea containing complete protease inhibitor cocktail (Roche) and Halt™ Phosphatase Inhibitor (Thermo) and lysates stored at -80 °C until for further analysis. Lysates were processed for tryptic digestion as follows. Cell pellets/lysates underwent sonication at 4 °C for 2 min, using a VialTweeter device (Hielscher-Ultrasound Technology), and then centrifuged at $18,000 \times g$ for 1 h to remove the insoluble material. A total of 300-500 μg supernatant proteins (determined by BioRad Bradford assay) were transferred to clean Eppendorf tubes. Supernatant protein mixtures were then reduced by 10 mM tris-(2-carboxyethyl)-phosphine (TCEP) for 1 h at 37 °C and 20 mM iodoacetamide (IAA), in the dark for 45 min, at room temperature. Then, five volumes of precooled precipitation solution containing 50% acetone, 50% ethanol, and 0.1% acetic acid were added to the protein mixture and kept at −20 °C overnight. The mixture was centrifuged at $18,000 \times g$ for 40 min. The precipitated proteins were washed with 100% acetone and 70% ethanol with centrifugation at $18,000 \times g$, 4 °C for 40 min, respectively. Protein pellets were dried in SpeedVac for 5 min. 300 μL of 100 mM $NH_4HCO_3$ was added to all samples, which were digested with sequencing grade porcine trypsin (Promega) at a ratio of 1:20 overnight at 37 °C. After digestion, the peptide mixture was acidified with formic acid and then desalted with a C18 column (MarocoSpin Columns, NEST Group INC). The amount of the final peptides was determined by Nanodrop (Thermo Scientific). About 5% of the total peptide digests were kept for total proteomic analysis of the cell line baseline profiles.

## Phosphoproteomic sample preparation

From the same peptide digest above, ~95% of the peptides from each sample were used for phosphoproteomic analysis. Phosphopeptide enrichment was performed using the High-Select™ Fe-NTA kit (Thermo Scientific, A32992), according to the kit instruction. Briefly, the resins of one spin column in the kit were divided into five equal aliquots, each used for one sample. The peptide-resin mixture was incubated for 30 min at room temperature and then transferred into the filter tip (TF-20-L-R-S, Axygen). The supernatant was removed after centrifugation. Then the resins adsorbed with phosphopeptides were washed sequentially with 200 μL× 3 washing buffer (80% ACN, 0.1% TFA) and 200 μL × 3 $H_2O$ to remove nonspecifically adsorbed peptides.

The phosphopeptides were eluted off the resins by 100 μL × 2 elution buffer (50% ACN, 5% $NH_3 \cdot H_2O$). All centrifugation steps above were conducted at 500 $g$, 30 sec. The eluates were collected for speed-vac and dried for mass spectrometry analysis.

## Mass spectrometry data acquisition

For each proteomic ($N = 18$) and phosphoproteomic ($N = 354$) sample generated above, DIA-MS analysis was performed on 1 μg of peptides, as described previously[107,108].

Briefly, LC separation was performed on EASY-nLC 1200 systems (Thermo Scientific, San Jose, CA) using a self-packed analytical PicoFrit column (New Objective, Woburn, MA, USA) (75 μm × 50 cm length) using C18 material of ReproSil-Pur 120 A C18-Q 1.9 μm (Dr. Maisch GmbH, Ammerbuch, Germany). A high-throughput, 75-min measurement with buffer B (80% acetonitrile containing 0.1% formic acid) from 6% to 37% and corresponding buffer A (0.1% formic acid in $H_2O$) during the gradient was used to elute peptides from the LC. The flow rate was kept at 300 nL/min with the temperature-controlled at 60 °C using a column oven (PRSO-V1, Sonation GmbH, Biberach, Germany).

The Orbitrap Fusion Lumos Tribrid mass spectrometer (Thermo Scientific) instrument coupled to a nanoelectrospray ion source (NanoFlex, Thermo Scientific) was calibrated using Tune (version 3.0) instrument control software. The spray voltage was set to 2000 V and the heating capillary temperature at 275 °C. All the DIA-MS methods consisted of one MS1 scan and 40 MS2 scans of variable isolated windows[108], with 1 m/z overlapping between windows. The MS1 scan range is 350–1650 m/z, and the MS1 resolution is 120,000 at m/z 200. The MS1 full scan AGC target value was set to be 2.0E5, and the maximum injection time was 100 ms. The MS2 resolution was set to 15,000 at m/z 200 with the MS2 scan range 200–1800 m/z, and the normalized HCD collision energy was 28%. The MS2 AGC was set to be 5.0E5, and the maximum injection time was 50 ms. The default peptide charge state was set to 2. Both MS1 and MS2 spectra were recorded in profile mode. Detailed MS settings can be inspected through raw files provided via ProteomeXchange.

## Mass spectrometry data analysis

Raw data files were processed and converted to mzXML by ProteoWizard[109] (version 3.0), enabling centroiding (using the vendor-provided algorithm) on MS1 and MS2 levels. For peptide identification and quantification, an integrated Snakemake workflow consisting of DIA-Umpire[110,111] (version 2.1.6), MSFragger[112] (version 2.3.0), the Trans-Proteomic Pipeline (PeptideProphet[113,114], PTMProphet[115], iProphet[116], version 5.2.0), EasyPQP (version 0.1.6), OpenSWATH[102] (OpenMS[117], version 2.5.0), PyProphet[63,118] (version 2.1.4) and TRIC[119] (msproteomicstools, version 0.11.0) was used.

A UniProtKB/Swiss-Prot protein sequence database was used for MSFragger. The spectral library was controlled to 1% PSM-, peptide- and protein-level FDR in global context and the best site-localization per phosphosite was selected. EasyPQP exported a global library, as well as a sample-specific library for each run.

OpenSWATH was run using the sample-specific high-confidence library for mass calibration and non-linear retention time alignment with enabled IPF[62] module for peptidoform-level confidence estimation. PyProphet with enabled IPF module and using the XGBoost classifier[120] was used for statistical validation. Peptides and proteins were filtered to 1% FDR in global context. TRIC was used for feature alignment using the IPF peptidoform-level scores in run-specific context, aligning detected peptides by lowess with a seed FDR of 1% to a maximum of 5%.

For quantitative protein abundance inference, the R-package "iq"[121] (version 1.9), implementing the MaxLFQ algorithm[104] for DIA-based datasets, was used with default parameters.

The full workflow, all used parameters, and software distributed as Docker containers that enable accurate reproduction of the analysis are provided with the dataset via ProteomeXchange.

## CRISPRko validation experiment

**Cell culturing.** The following cell culture conditions were used: (1) HCT-15 – RPMI 10% FBS + pen/strep, (2) NCI-H508 - RPMI 10% FBS + pen/strep, (3) 293 T – DMEM 10% FBS + pen/strep.

All cell lines were routinely tested for *Mycoplasma* contamination and were kept in a 37 °C humidity-controlled incubator, with 5.0% $CO_2$.

**Optimizing drug concentrations for pooled CRISPRko screens.** Drug concentrations were optimized for each cell line to ensure ideal long-term CRISPRko screen readouts. The time to reach 10-population doublings depended primarily on characteristics of each cell line and could take 25 to 40 days. Trametinib and linsitinib perturbations were tested with 5 concentrations (10 μM, 1 μM, 0.1 μM, 0.01 μM, and DMSO only) and the cellular growth effect was assessed for each of those concentrations for each of the cell lines in a long-term growth assay. The DMSO concentration was optimized for 0.15%.

Cells were grown and underwent drug treatment in 15 cm plate format, splitting the cells whenever they became approx. 80-90% confluent. When the DMSO-plate reached 10-population doublings, the total number of cell divisions was counted for each of the above-mentioned drug treatment plates. Final concentrations for the pooled CRISPRko-screens were selected to represent drug concentrations which had only a modest effect on cell division rate (approx. 10-20% slower cell divisions compared to DMSO), similarly as previously suggested[122].

**CRISPRko library design.** For CRISPRko screening we designed the target gene list to include all human kinases (obtained from UniProt: pkinfam.txt) and phosphatases (obtained from reference[10]). All these genes were targeted with 4 sgRNAs/gene. For guide designs we used CRISPick[123,124].

**CRISPRko oligo synthesis and library cloning.** Oligo libraries (4404 oligos) were ordered from Twist-biosciences in following format:

cttgtggaaaggacgaaacaccgNNNNNNNNNNNNNNNNNNNNNNgtttAagagctagaaatagcaagttTaaataaGgct.

The following Twist oligo pool amplification conditions were used:

Concentrations: 1 μl Twist oligo library (1 ng/ul), 10 μl 5x KAPA HIFI buffer, 1 μl dNTPs, 1 μl KAPA, 2 μl sgRNA_insert_dd_F (10 μM), 2 μl sgRNA_insert_dd_R (10 μM), 2.5 μl 20xSYBR, 30.5 μl $H_2O$.

Cycles: 95 °C 3 min, 98 °C 20 sec, 56 °C 15 sec (done with qPCR, stopped before saturation), 72 °C 20 sec, 72 °C 5 min, 4 °C ∞

sgRNA_insert_dd_F:CTTGTGGAAAGGACGAAACACCG
sgRNA_insert_dd_R:AGCCTTATTTAAACTTGCTATTTCTAGCTCTTAAAC.

After PCR, the insert was gel purified (GeneJet) and Gibson cloned into BsmBI-digested modified lentiGuide-Puro.3xBsmBI (Addgene #196709). For this study, the 3rd BsmBI-site was mutated from the vector.

Gibson cloned insert + vector was Isopropanol precipitated and large-scale electroporated into Lucigen Enduro competent cells. The bacterial colonies were scraped from 10 x 24, 5 cm x 24, 5 cm agar plates so that the estimated library complexity was > 1000 colonies/sgRNA.

**CRISPRko library viral packaging.** 13 million 293 T cells were seeded for each 15 cm dish previous night of the transfections. The following morning the viral transfections were conducted as follows: 22.1 μg sgRNA-library containing lentiGuide-Puro or modified lenti-Cas9-

sgHPRT1, 16.6 µg PsPAX2 (Addgene 12260), 5.5 µg PMD2G (Addgene 8454), 1660 µl of sterile $H_2O$.

After mixing the plasmids 1106 µl of Fugene HD (Promega) was added to the mix.

The transfection mixture was briefly vortexed and incubated 10 minutes at room temperature before adding dropwise to 293 T cells. Altogether 3 x 15 cm plates were transfected for sgRNA-library containing lentiGuide-Puro and 1 x 15 cm plates were transfected with modified lenti-Cas9-sgHPRT1 (Addgene #196713). For this study, the sgHPRT1 part was removed from the lenti-Cas9-sgHPRT1-vector.

The transfection mixture was removed the following day and the virus was collected at 48 h and 72 h after initial transfections. To remove cellular debris, the virus-containing supernatant was centrifuged 500 x *g* for 5 min and filtered by using 0.45µm PES filters (Millipore). The lentivirus was concentrated by using Lenti-X concentrator (Clontec), aliquoted, and stored at −80 °C.

**Generation of Cas9 expressing CRC cell lines.** Cas9 expressing cell lines were generated as follows: Concentrated lenti-Cas9-lentivirus was transduced to CRC cell lines (in the presence of 8 µg/ml polybrene) with estimated MOI 0.3. The virus was removed the following day and 4 µg/ml Blasticidin was added to the cells. Blasticidin selection was continued as long as the control cells (non-transduced) were viable.

**CRISPRko screening.** sgRNA containing lentiviruses were transduced into Cas9 expressing CRC cell lines (in 15 cm plate-format) in quadruplicates (in presence of 8 µg/ml polybrene), at an estimated MOI = 0.2. After 24 h, the lentivirus-containing media was removed, cells were washed with PBS, and puromycin-containing media (3 µg/ml) was added to the cells for 48–96 h until all control cells (not virus-infected) were dead. After this the cells were cultured for two additional days, allowing plates to reach approx. 80% confluency. At this point, cells were divided into 3 parts; 1/3 going into −80 °C as time point 1 to assess sgRNA representation baseline, 1/3 to continue to culture with DMSO and 1/3 to continue to culture with either with Linsitinib or Trametinib. Cells were always maintained at >1500 cells per guide throughout the screens and finally harvested after 10 population doublings to assess gene essentiality. The exact time (in days) for this varied for DMSO/Linsitinib/Trametinib with different cell lines. After the screen, the genomic DNA from the first and the last timepoints (DMSO & Drug perturbed) were extracted by using Blood and Cell culture DNA Maxi kits (Qiagen).

**Preparation of the sequencing library from genomic DNA.** NGS library preparations were performed as follows: Briefly, 40 µg of gDNA, theoretically corresponding to 6 million diploid cells, was used as PCR template in 4 parallel NGS PCR1 reactions (10 µg template DNA per reaction) using ExTaq DNA polymerase (Takara bio). After 18 cycles, the 4 replicate reactions were pooled together. 2 µl of pooled NGS PCR1 product was used as a template for NGS PCR2 which was run with qPCR with index primers and stopped before the amplification started to saturate. The resulting products of approx. 360 bp were gel purified (GeneJet), pooled together and Next generation sequenced.

NGS_PCR1 master mix:10 µg gDNA, 0.75 µl ExTaq, 10 µl 10 x ExTaq Buf, 8 µl dNTPs, 0.5 µl CRISPRko_PCR_1R (pool of 5 (100 µM)), 0.5 µl CRISPRko_PCR_1F (100 µM) to 100 µl $H_2O$.

PCR1 protocol (18 cycles): 98 °C 1 min, 98 °C 10 sec (18 cycles), 58 °C 30 sec (18 cycles), 72 °C 30 sec (18 cycles), 72 °C 10 min, 4 °C ∞.

NGS_PCR2 master mix: 2 µl DNA (from 1st PCR), 0.375 µl ExTaq, 5 µl 10 x ExTaq Buf, 4 µl dNTPs, 0.5 µl CRISPRko_PCR_2F (100 µM), 0.5 µl CRISPRko_PCR_2R(index) (100 µM), 1.25 µl 20xSYBR, 36.4 µl $H_2O$.

PCR2 protocol: 98 °C 1 min, 98 °C 10 sec, 60 °C 30 sec (done with qPCR, stopped before saturation), 72 °C 30 sec, 72 °C 10 min, 4 °C ∞.

**CRISPRko Oligos used for NGS library preparation.** CRISPRko_PCR_1F: TGGAGTTCAGACGTGTGCTCTTCCGATCTTCTACTATTCTTTCCCCTGCACTGT

CRISPRko_PCR_1R:CTTTCCCTACACGACGCTCTTCCGATCT(1-5nt_stagger)TGTGGAAAGGACGAAACACCG

CRISPRko_PCR_2F: AATGATACGGCGACCACCGAGATCTACACTCTTTCCCTACACGACGCTCTTCCGATCT

CRISPRko_PCR_2R(index): CAAGCAGAAGACGGCATACGAGATNNNNNNNGTGACTGGAGTTCAGACGTGTGCTCTTCCGATCT

**High-throughput screening reporting table.** The high-throughput screening table for the CRISPRko screen is available in Supplemental Table 2.

**Data processing & statistical analysis**
For all data analysis steps, "viper" (version 1.22.0), "vespa" (version 1.0.2), "vespa.db" (version 1.0.2), and "vespa.aracne" (version 2.2) were used. "vespa.net" (version 1.0.2) was executed using the corresponding Docker images of the algorithms converted to Singularity images. All software tools are available from the corresponding repositories as referred below.

**Inference of a CRC-specific signaling network.** To generate a CRC-specific signaling network, we obtained the processed phosphoproteomic and total proteomic profiles from the CPTAC study S045[34] (referred to as "CPTAC-S045"). To account for potential confounding factors originating from protein abundance levels, we further generated a derived dataset, referred to as "CPTAC-S045N", where phosphopeptide abundance was normalized by the corresponding protein-level intensity values. The datasets were imported from CCT and CPTAC formats and converted to PVM by the corresponding "vespa" functions without further processing except mapping of identifiers. Only tumor samples were used across all analyses.

The phosphoproteomic dataset generated in this study (referred to as "U54") was imported from the OpenSWATH file format and converted to PVM by the corresponding "vespa" function, with quantile normalization grouped by cell line and centering enabled. The baseline profiles of the six cell lines measured in triplicates ("U54-BL"), as well as drug perturbations across three distinct time points (1 h, 24 h, 96 h; "U54-NET") and the full-time series ("U54-DP") were exported as separate PVM files.

These three PVM matrices (CPTAC-S045, CPTAC-S045N, and U54-NET) were used as input to the "vespa.net" workflow. By default, separate signalons were generated using the stDPI/DPI, LP[16] (published dataset), HSM/P[15] (published dataset), and PC[18] (version 12) methods. For all analyses, the PVM of U54 BL was used to generate optimized signalons. For all analyses, except for benchmarking, stDPI/DPI-based signalons were used.

To estimate the fraction between KP-enzymes covered by a phosphopeptide or a signalon in our study and the total number of different KP-enzymes present in cells, we measured the overlap with expressed KP-enzymes, based on gene expression profiles. Pre-processed RNA-seq profiles for the six CRC cell lines were obtained from CCLE[46]. Counts were normalized to TPM and identifiers mapped to SwissProt/UniProtKB. KP-enzymes were considered to be expressed when their average TPM across the six cell lines was ≥ 10 TPM, consistent with the expression of the first quantile of KP-enzymes covered by at least one phosphopeptide in the U54BL dataset (kinases: 9.96 TPMs; phosphatases: 8.99 TPMs).

**Benchmark and validation of VESPA**
**Benchmark signaling network generation.** SigNets based on different data completeness thresholds of the U54-NET datasets were generated as described above.

**Comparison of MI methods.** To compare the effect of different MI estimators, the HSM priors were used as ground truth, as provided by the vespa.db R-package. Based on the U54-NET datasets, subsets were generated with ≥ 20%, ≥ 40%, ≥ 60%, ≥ 80% and 100% data completeness. To compute hpMI and dMI, the sparse input matrices were used, to compute iMI, missing values were imputed row-wise, as previously described. To compute dMI, "vespa.aracne" was extended to support dMI (Git branch "depletion_support"; revision 470944f). "vespa.aracne" was run as described above, but without stDPI/DPI and using 100 bootstraps. Only significant interactions (<5% FDR) were considered. The overlap of these interactions with HSM was used to compute the summed score.

**DPI benchmarking.** stDPI, DPI, and noDPI-based SigNets were generated from U54-NET as previously described. Interactions were selected as ground truth (positive gold standard) if they were identified as ST-K → S pairs based on HSM analysis with PDZ, SH3, WH1, and WW domains, since these represent the primary determinants of specific ST-K interactions with serine and threonine phosphopeptides. As a negative gold standard, we used candidate TK → S interactions with an HMS-predicted, phosphotyrosine-specific PTB, PTP and SH2 domain interaction. This is because the dataset used in the benchmark (U54-NET) is not enriched for phosphotyrosine peptides and should thus result in no such interactions. This produces a context-specific reference dataset identifying the most and least likely direct and indirect interactions, thus providing a suitable framework for relative methodological comparisons. Receiver-Operating-Characteristics (ROC) were generated using the pROC R-package (version 1.17.0.1) and default parameters for each signaling network separately. Two-tailed $P$-values for ROC curve comparisons were also computed using pROC by DeLong's test and using default parameters. Precision-recall curves (PRC) and corresponding metrics were computed using the PRROC R-package (version 1.3.1).

**mVESPA Benchmarking.** To benchmark mVESPA, we used the baseline (untreated) phosphoproteomic profiles from the six cell lines in the U54-BL dataset. We downloaded the curated GDSC[36] drug sensitivity dataset and the primary target list from the original INKA publication[24] (Dataset_EV6.xlsx). The phosphostate- and activity-level signalons used for the benchmark were generated as described above. For the analysis, we used the "viperSignature" of the "viper" R-package to compute the differential activity of relevant KP-enzymes in sensitive vs. resistant or insensitive cell lines, with default parameters.

We modified the INKA[24] benchmark strategy to use the differential activity of KP-enzymes representing established drug targets in sensitive vs. resistant cells, rather than the absolute protein activity in sensitive cells. The analysis was performed independently for each cell line (Fig. 2c). The GDSC identifies sensitive (low z-score) vs. resistant (high z-score) cell lines, based on the compound's log(IC$_{50}$), as measured across 1000 cell lines. For this benchmark, we thus selected compounds eliciting the greatest differential sensitivity (i.e., z-score ≤ −1.0 and ≥ 1.0 for resistant and sensitive cell lines, respectively), when assessed for all possible CRC cell line pairs (Fig. 2d). For each selected drug, we used mVESPA to assess the activity of the target enzyme in sensitive vs. resistant cells, using the CRC SigNets (Fig. 2c). Finally, we assessed the method's sensitivity using an empirical score, as proposed by INKA. Specifically, let's define $DP_i$ as the differential activity of the $i$th protein, ranked based on their differential activity from the most to the least significant one, and $w_i$ as a weight representing the sensitivity of a cell line (C) to a specific drug (D); then the empirical score for a specific inhibitor and cell line is defined as the integration of the product $w_i DP_i$ over the $n$ most differentially active proteins.

$$S_{D,C}(n) = \sum_{i=1}^{N} w_i DP_i \qquad (3)$$

ROC metrics were computed as described previously (see Supplemental Notes: "Precision-specificity analysis using ROC curve")[59] and individual ROC curves were averaged. Statistical comparison of the differential comparison AUC metrics was conducted using unpaired, right tailed Wilcox' tests (R-package "stats", version 4.2.1).

**Comparative analysis of previously published algorithms.** To benchmark the algorithm against previously published ones, we applied the KSTAR benchmark suite[38], downloaded from (https://github.com/NaegleLab/KSTAR_Applications/tree/95563ddc57d39c200f06dd78a2c3672cd2d04bf2), according to the instruction of the original authors. We used mVESPA to predict KP-activity using either the Johnson or and KSTAR-benchmark-supplied PSP reference networks. KP-enzymes with <5 substrates were excluded from the analysis, and signalons with >500 substrates were trimmed to 500, as previously discussed. Crosstalk correction was not used in the analysis and only phosphostate-level signalons were used. $P$-values were obtained from VESPA NES values and corrected for multiple-testing by the Benjamini-Hochberg (FDR) approach[125].

**Application of VESPA to decryptM dataset.** The A431 kinase inhibition profiles for Afatinib, Gefitinib and Dasatinib were obtained from the original decryptM publication[21]. Due to the absence of suitable datasets for the generation of signaling networks for A431, we used a dVESPA-generated signaling network based on the CPTAC Lung Squamous Cell Carcinoma (LSCC) Discovery Study[41], which was generated as described above, but optimized for the CPTAC instead of the A431 profiles.

mVESPA was applied as described above, using the $t$-statistic reported by the decryptM dataset instead of peptide abundance. mVESPA was applied with default parameters and the integrated results were used for further visualization.

### Representation of CRC subtypes by cell line models
**Cell line selection.** Cell lines were selected based on their ability to recapitulate the activity of the top 50 most differentially active proteins (i.e., candidate Master Regulators) in TCGA CRC samples, as implemented by the OncoMatch methodology[42,45], at a conservative statistical significance threshold ($p < 10^{-5}$). To rank matching cell lines for each cluster, we used the OncoMatch scoring function[45] to select six CRC cell lines, representative of five out of eight subtypes (for cell lines with the top 5 OncoMatch scores) and all eight subtypes (for cell lines with the top 10 scores), as identified by MOMA analysis of the TCGA CRC cohort[45].

**MSI classification.** Information on the MSI status of CPTAC S045 samples was obtained from the original publication[34]. Information on the MSI status of the six cell lines was obtained from CCLE[126].

**CMS transcriptome-level classification.** Preprocessed RNA-seq profiles for CPTAC S045[34] were obtained from the original publication. Preprocessed RNA-seq profiles for the six CRC cell lines were obtained from CCLE[46]. Counts were normalized to TPM for both datasets and identifiers were mapped to be compatible with CMS. Only transcripts measured in both datasets were used for downstream analysis. The CMS classifier[47] was then applied using the RandomForest predictor and default parameters to assess subtype membership.

**VESPA analysis.** KP-enzyme activities were inferred by VESPA using the CRC-specific signalons, as described above. Because the phosphoproteomic profiles of CPTAC-S045 and U54-BL had very different levels of missing values, the profiles were first randomly subsampled, to ensure that phosphopeptide detectability was equivalent in both datasets. The two datasets were then combined and rank-normalized, first column-wise, then row-wise, as described previously[45]. The "viper"

function was applied to compute phosphostate- activity-, and integrated-level-based KP-enzyme activities, including crosstalk correction.

**Cluster analysis.** Phosphostate- and activity-level VESPA matrices were clustered by the k-medoids approach, prioritizing cluster robustness, as previously described[45].

**Gene set enrichment analysis.** GSEA analysis was performed using the R-package "fgsea" (version 1.14.0) to analyze enrichment of Reactome pathways (version 75), reduced to include only KP-enzymes (downstream pathway "R-HSA-162582"). Only statistically significant results (adj. $p < 0.05$), in at least one sample were reported.

**Feature selection.** To select the top 50 most important features for subtype classification we used the Random Forest recursive feature elimination method from the R-package "caret" (version 6.0-86). For simplicity, Fig. 3b only shows the cumulative most important features of the CMS and pVC classification systems, grouped according to pVC. Supplementary Figs. 7, 8 show the full results, whereas Supplementary Fig. 9 depicts the data underlying Fig. 3b, grouped according to CMS.

**Visualization.** Heatmaps were generated using the "*pheatmap*" (version 1.0.12) R-package. Hierarchical clustering on row-level was conducted using the default R "hclust" function with default parameters.

**Targeted drug perturbations of CRC cell lines**
**VESPA analysis.** The 336 perturbed U54-DP phosphoproteomic profiles were preprocessed to impute missing values using the row-wise minimum as described above. The peptide abundances of each sample were normalized by the corresponding DMSO controls, separately for each cell line. Time point values were averaged using a sliding window including the preceding and following time point, if available. E.g., the 15 min time points were normalized using the average of the 5 min, 15 min and 1 h time points from the corresponding DMSO treated cells. Log$_2$ fold changes were then used as input for all downstream steps. KP-enzyme activity was inferred by VESPA using the stDPI/DPI CRC signalons as previously described. The "viper" function was applied at the phosphostate- and activity-level, using a bootstrapped "viperSignature" null model based on the DMSO controls, with 1000 permutations. Crosstalk correction was included as previously described.

**Drug/cell line sensitivity analysis.** Drug sensitivity data from GDSC[36] was obtained and $z$-score was transformed per drug and GDSC dataset over all covered cell lines. Drug/cell line pairs with z-score < −1.0 were defined as sensitive, while those with z-score > 1.0 were defined as insensitive. Violin plots were generated using the "geom_violin" function with default parameters of the R-package "ggplot2" (version 3.4.0).

**Visualization.** Heatmaps were generated using the "*pheatmap*" (version 1.0.12) R-package. Hierarchical clustering on row and column-level was conducted using the default R "hclust" function with default parameters.

**Temporal dynamics of primary drug targets.** Known primary targets for the drug compounds were obtained from DrugBank[64] and ProteomicsDB[65]. Only the top five most downregulated target proteins per drug compound were visualized.

**Context-specific wiring of signaling pathways**
**VESPA analysis.** The 336 perturbed U54-DP phosphoproteomic profiles were preprocessed as described above. The "viper" function was applied separately for each cell line on phosphostate- and

activity-levels using a rank-normalized matrix[45] and including crosstalk correction. Phosphostate- and activity-level VESPA results were integrated as described above.

**DeMAND analysis.** DeMAND assesses the dysregulation of individual PPIs using the Kullback-Leibler divergence, by computing changes in mutual information across drug perturbations at different time points and/or drug concentrations vs. vehicle control-treated samples[59]. Enrichment of dysregulated PPIs (edges) originating on the same protein (node) in the network can then be used to identify proteins most dysregulated by a drug. The DeMAND[59] (version 1.18.0) algorithm was used to assess context-specific wiring of signaling pathways. DeMAND was applied on both phosphostate- and activity-level VESPA analysis results. First, phosphostate-level VESPA scores were used with the corresponding signalons. Second, activity-level VESPA scores were used in combination with STRING PPI DB (version 11) as reference interaction database, including only interactions with probability > 0.5. The results were then combined using Stouffer's method. To generate non-subtype-specific DeMAND MoA profiles, for each drug perturbation, the temporal profiles of all cell lines were compared against the DMSO controls. To generate subtype-specific DeMAND MoA profiles, the temporal profiles generated for each cell line/drug pair were compared to the matched DMSO controls, used as null distribution. Edge and node $p$-values were integrated using Fisher's method and BH-adjusted for multiple testing.

**Visualization.** Heatmaps were generated using the "*pheatmap*" (version 1.0.12) R-package. Hierarchical clustering on row and column-level was conducted using the default R "hclust" function with default parameters.

**Cytoscape.** To visualize the interaction networks, Cytoscape (version 3.8.2) was used. Nodes indicate the most affected KP-enzymes with the inner circle colors indicating cell line type and the outer circle color and node size indicating VESPA activity. Edges indicate dysregulated, undirected interactions between the KP-enzymes as inferred by DeMAND. Line thickness indicates the significance of dysregulation. Dysregulated nodes (BH-adjusted $p < 0.05$) and known primary targets are colored. Grey nodes indicate connecting dysregulated nodes (BH-adjusted $p < 0.1$).

**Context-specific adaptive stress resistance mechanisms**
To identify the mechanism of adaptive resistance for each cell line/drug combination, we assessed the effect of drug perturbation vs. vehicle control treated samples at the late time points using a time-point-paired $t$-test for differential testing of the VESPA inferred protein signaling activities (Supplemental Data 17). The $p$-values of all conditions were then integrated by Stouffer's method to select significant, increased activity of candidate resistance factors across all conditions ($q < 0.05$, mean($t$-statistic) > 0) (Fig. 7a, Supplemental Data 18).

**VESPA differential testing.** Time series were used to investigate the adaptive response of KP-enzymes to drug perturbations, by comparing their late (24 h, 48 h, 96 h) differential activity, compared to DMSO-treated samples, using a paired, one-tailed $t$-test (R version 4.2.1). To select candidates for visualization, $p$-values were integrated by Stouffer's method across all conditions and corrected for multiple testing ($q < 0.05$).

**Protein abundance differential testing.** For quantitative phosphoprotein abundance inference, the R-package "iq"[121] (version 1.9), implementing the MaxLFQ algorithm[104] for DIA-based datasets, was used with default parameters. Differential testing was then conducted identically as for VESPA.

**Identification of essential genes using DESeq2.** Alignment of NGS with sgRNA guides was performed using the "ShortRead" R-package (1.54.0). Essential genes for the DMSO vs. T0 comparison were obtained from a published repository[127]; CRC-specific essential genes were obtained from DepMap[128], and filtered to the 10% quantile of the gene effect. Differential expression analysis was performed separately for each gRNA guide with the "DESeq2" R-package (1.36.0). *P*-values were integrated using Stouffer's method and corrected for multiple-testing by the Benjamini-Hochberg FDR approach[125].

**Receiver operating characteristics.** ROC curves and statistics were generated using the R-package "pROC" (version 1.18.0). Significant (FDR < 0.01) CRISPRko results were used as ground truth values (negative beta: true; positive beta: false) and the VESPA (t-statistic), differential abundance (t-statistic), or VESPA-DeMAND ($-\log10$(De-MAND BH$-$adjusted $p-$value) * sign(VESPA t$-$statistic)) scores were used as predictors. ROC *p*-values were computed using the function "roc.area" from the R-package "validation" (version: 1.42).

**Correlation analysis.** Correlation analysis was performed by comparing the t-statistic of the differential VESPA analysis with the significant (FDR < 0.01) log-fold-changes reported by DESeq2. Correlation statistics were computed using a one-tailed Spearman correlation test (R version 4.2.1).

**Exclusion of tumor suppressor genes.** For the analyses excluding tumor suppressor genes, all genes present in TSGene 2.0 database[94] were excluded.

**Visualization.** Heatmaps were generated using the "*pheatmap*" (version 1.0.12) R-package. The t-statistic values of the described above are visualized. Hierarchical clustering on row- and column-level was conducted using the default R "hclust" function with default parameters.

**Reporting summary**

Further information on research design is available in the Nature Portfolio Reporting Summary linked to this article.

## Data availability

The CRC mass spectrometry proteomic and phosphoproteomic raw and preprocessed data generated in this study have been deposited to the ProteomeXchange Consortium via the MassIVE partner repository with the data set identifiers MSV000091204/PXD039859 [https://doi.org/10.25345/C5R20S61Q]. The CRISPRko RNA-seq raw and pre-processed data discussed in this publication have been deposited in NCBI's Gene Expression Omnibus[129] and are accessible through GEO Series accession number GSE224396 [https://www.ncbi.nlm.nih.gov/geo/query/acc.cgi?acc=GSE224396]. The VESPA analysis results for selected CPTAC datasets are available from Zenodo[130] [https://doi.org/10.5281/zenodo.8220610]. Supplemental Data 1-20 is available from Zenodo[131] [https://doi.org/10.5281/zenodo.10925250]. CPTAC mass spectrometry proteomic and phosphoproteomic raw and pre-processed data (PDC000116 and PDC000117) was obtained from Proteomic Data Commons (PDC) under the Creative Commons CC-BY 4.0 licensing terms: [https://proteomic.datacommons.cancer.gov/pdc/]. Source data are provided with this paper.

## Code availability

VESPA is available as modular platform-independent open-source software under a non-commercial usage license. VESPA consists of five different modules, which are provided as versioned source code, binaries or docker containers. The "vespa" R-package for signaling protein activity inference is available from GitHub (https://github.com/califano-lab/vespa) and Zenodo[132] (https://doi.org/10.5281/zenodo.10731059). The "vespa.db" R-package providing preprocessed reference networks is available from GitHub (https://github.com/califano-lab/vespa.db) and Zenodo[133] (https://doi.org/10.5281/zenodo.10731069). The "vespa.aracne" algorithm is available from GitHub (https://github.com/califano-lab/vespa.aracne) and Zenodo[134] (https://doi.org/10.5281/zenodo.10731065). The "vespa.net" Snakemake workflow to generate context-specific signalons from one or multiple datasets is available from GitHub (https://github.com/califano-lab/vespa.net) and Zenodo[135] (https://doi.org/10.5281/zenodo.10731073). A tutorial describing the full analysis workflow with example data is available from GitHub (https://github.com/califano-lab/vespa.tutorial) and Zenodo[136] (https://doi.org/10.5281/zenodo.10731075).

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

## Acknowledgements

This study was supported by NCI U54CA274506 (Center for Cancer Systems Therapeutics, CaST), a supplemental grant to NCI U54 CA209997 (Cancer Systems Biology Consortium), the NCI Office of Cancer Target Discovery and Development (CTD2) award U01CA272610, and the NIH Shared Instrumentation Grants S10 OD012351 and S10 OD021764 all to A.C. G.R. was supported by grants P2EZP3_175127 and P400PB_183933 from the Swiss National Science Foundation. Y.L. was supported by the National Institute of General Medical Sciences (NIGMS), NIH through grant R01GM137031. B.H. was supported by NIH R35GM1395858 and NCI U54 CA209997 (Cancer Systems Biology Consortium).

## Author contributions

G.R.: Conceptualization, Methodology (VESPA, Benchmarking, Analysis, Integration), Software (VESPA, Benchmarking, Analysis, Integration), Validation, Writing – Original Draft, Visualization, Funding Acquisition W.L.: Methodology (Phosphoproteomics), Writing – Review & Editing M.T.: Methodology (CRISPRko), Validation (CRISPRko), Writing – Review & Editing J.H.: Methodology (hpMI, stDPI), Software (VESPA/ARACNe), Writing – Review & Editing P.S.S.: Methodology (Cell culture, Drug sensitivity assays, Drug perturbation assays), Writing – Review & Editing S.P.: Methodology (Cell culture, Drug sensitivity assays, Drug perturbation assays), Writing – Review & Editing A.T.G.: Methodology (CRISPRko), Software (DESeq2-based analysis), Writing – Review & Editing C.K.: Methodology (Cell culture, Drug sensitivity assays, Drug perturbation assays), Writing – Review & Editing P.K.: Methodology (CRISPRko), Validation (CRISPRko), Writing – Review & Editing D.M.: Conceptualization, Writing – Review & Editing, Project Administration, Funding Acquisition B.H.: Conceptualization, Writing – Review & Editing, Funding Acquisition Y.L.: Conceptualization, Methodology (Phosphoproteomics), Writing – Review & Editing, Supervision, Funding Acquisition A.C.: Conceptualization, Writing – Original Draft, Supervision, Funding Acquisition

## Competing interests

A.C. is the founder, equity holder, and consultant of DarwinHealth Inc., a company that has licensed some of the algorithms used in this manuscript from Columbia University. Columbia University is also an equity holder in DarwinHealth Inc. and assignee of patent US10,790,040 ("Virtual inference of protein activity by regulon enrichment analysis"), which covers some components of the algorithms used in this manuscript. The other authors declare no competing interests.
