## [Peer Review File · Nature Communications]

REVIEWER COMMENTS

Reviewer #1 (Remarks to the Author): Expert in computational systems biology, signalling networks, and phosphoproteomics

In this manuscript, Rosenberger and colleagues present a system biology method, which they named VESPA, that takes phosphoproteomics data and knowledge of kinase-substrate interactions to infer changes in kinase networks induced by perturbations. The method was tested with the analysis of a phosphoproteomic time series dataset from colorectal cancer cells treated with kinase inhibitors. The adaptive response that this analysis suggested was experimentally investigated using a CRISPR ko assay. While the topic and approaches are of interest to the field of kinase signaling and cancer therapeutics, the low perceived novelty of the study, and the unproven accuracy of the approach limit enthusiasm for this work.

Specific comments

1. The main issue with this study is that VESPA does not seem to perform well. From the perturbation data in Figure 5, one would conclude that the method fails to quantify changes in kinase activities induced by kinase inhibitors. PIK3CA, EGFR and MAP2K1/2 scores seem to increase or remain unchanged by treatments with the respective inhibitor in most cases.

2. Another issue is that the study does not seem to be overtly novel, and ignores several papers that are a precedent to this work. For example:

- When discussing the study of cancer resistance mechanisms, the authors mention that “so far the focus has been on the discovery of genetic events leading to selection of drug resistant clones”. Contrary to this claim, there are several papers that have used phosphoproteomic approaches to investigate drug response mechanisms (e.g. PMID: 36688815) and adaptation mediated drug resistance (e.g., PMID: 26060313).

- The comment about “systematic, proteome-wide elucidation of tissue-specific signaling networks has trailed the study of regulatory interactions” (page 3), while technically correct, ignores important contributions to this field (e.g., PMID: 27909043)

- The statement “phosphoproteomic data for all CCL cell lines was not available” is inaccurate because a relatively comprehensive CRC phosphoproteomics data has been available for this purpose for several years (PMC5583477)

- The claim that this work “resulted in one of the largest context-specific targeted drug perturbation phosphoproteomic datasets” ignores studies, such as PMID: 31959955 and PMID: 36926954, which profiled larger number of drug perturbed phosphoproteomes than in this study.

3. There are contradictions in the approach used by the authors. A feature of dVESPA is that it eliminates indirect ES interactions, yet in order to estimate tyrosine kinase activity, the authors seem to rely on measuring the activity of Ser/Thr kinases that are downstream of tyrosine kinases. In other words, the approach relies in measuring indirect ES interactions to estimate tyrosine kinase activation.

4. It is not clear how the method quantifies PIK3CA activity, as shown in Figure 5, if indirect ES interactions are eliminated from the analysis. PIK3CA is a lipid kinase without significant protein kinase activity. This is indeed a puzzling feature of the approach given that one of the steps of the method is to

eliminate indirect interactions (page 13, line 267).

5. The manuscript is riddled with exaggerations and overblown comments. For example, the authors claim that they “designed a comprehensive longitudinal drug perturbation experiment (page 19)”. This is an exaggeration because the experiment used seven compounds at single dose for seven time-points, which is clearly not comprehensive. Similarly, the claim that mVESPA likely quantified “an almost complete subset of the KP-enzymes expressed in CRC cells” is overly speculative and another exaggeration. It would be relatively trivial to determine whether this is correct by simply comparing the results with databases of expression data.

6. It is claimed “that VESPA substantially outperforms established approaches” (page 5) but a direct comparison with those other approaches is not clear from reading the paper. To benchmark mVESPA the authors compare its performance with the method in reference 19 but the data on which of these methods performs best is not shown. Similarly, in the temporal analysis shown in figure 6, it is not clear whether established methods to quantify kinase activity from phosphoproteomics data would produce similar or more reliable results. In addition, the correlation between sensitivity to kinase perturbation and cognate kinase activity (as measured by mVESPA) does not seem to be significant for most cell lines. It would be interesting to see if established methods to infer kinase activities from phosphoproteomics data perform similarly to mVESPA in this regard.

7. The supplementary tables and datasets are impossible to review because there are no captions or any other type of explanation (meaning of headings, etc.) on the data that these files contain.

8. There are not details on the source of KP-enzyme direct and indirect substrate relationships used in the study.

Reviewer #2 (Remarks to the Author): Expert in cancer proteomics, bioinformatics, and colorectal cancer

This manuscript presents VESPA (Virtual Enrichment-based Signaling Protein activity Analysis), a computational method for dissecting enzyme-substrate interactions and measuring signaling protein activity. The authors also applied VESPA to investigate post-translational cell adaptation mechanisms associated with colorectal cancer's resistance to targeted drugs. There are multiple strengths of the manuscript, including its focus on an important and challenging question in phosphoproteomics data analysis, a computational algorithm built upon the widely used VIPER algorithm previously developed by the group, a unique and large context-specific targeted drug perturbation phosphoproteomic dataset with 354 phosphoproteomic profiles to demonstrate the application of the algorithm, and experimental validation of the hypotheses generated from the exploratory study. The manuscript should be of interest to bioinformatics, phosphoproteomics, and cancer research communities.

Major concerns:

1. The primary concern I have is the complexity of the manuscript. The authors have undertaken a substantial amount of work, which I believe warrants two separate publications. The first publication

could focus on the development, evaluation, and benchmarking of the computational method. The second publication could center on the application of the already benchmarked method to the analysis of drug resistance in colon cancer. This division would make the manuscript less dense and more reader-friendly, while allowing for a more comprehensive description of the computational method and its potential applications.

In particular, I recommend expanding the computational section to include descriptions of additional potential use cases for the VESPA algorithm. It would be helpful to clarify the required and optional input files for VESPA, such as whether it necessitates a CPTAC-type dataset and a time course phosphoproteomics dataset. Additionally, addressing whether VESPA is applicable to a single case-control phosphoproteomics dataset and how its performance compares to other algorithms mentioned by the authors would be valuable information for potential users. Providing such details could greatly enhance the impact of the algorithm and increase its usability.

If the authors choose to maintain the current arrangement, I suggest simplifying the text to make it more accessible to the broader audience of the journal. Some content in the results section could be moved to the methods section.

2. Another concern pertains to the normalization of phosphoproteomics data using global proteomics data. In the manuscript, the authors introduced a second CPTAC dataset where the CPTAC-S045 phosphosite abundances were normalized by the corresponding whole protein abundances, with the goal of addressing confounded kinase-phosphosite relationships. However, it is unclear which version of the data was ultimately used for downstream analysis. The authors observed that the normalized version yielded fewer optimized signalons. Does this reduction indicate that the normalization procedure eliminated biological signal? Additionally, if normalization is crucial, why was it not employed in the cell line study? The question of whether and how to normalize phosphoproteomics data remains unresolved, and I hope the authors could use their unique methods and datasets to shed light on this important question.

3. Benchmarking the MI estimator: Figure 2a, please clarify the ground truth and the benchmarking metric used for benchmarking in the Results section, together with detailed ground truth information and justification of choice.

4. Benchmarking indirect interaction removal: Figure 2b, as dVESPA can incorporate HSM information, please clarify whether HSM information is used in the dVESPA analysis. Moreover, considering the large number of negative space, is AUROC 0.693 good enough for inference? Adding AUPRC might be helpful? How do these methods compare to commonly used methods for such inference?

Minor:

1. Figure 1 is not very helpful for understanding the method.

2. Figure 4 does not really highlight any useful information other than the clusters. The giant heatmap with no clearly visible detail is better suited for the supplement.

3. Line 52-53, "systematic, proteome-wide elucidation of tissue-specific signaling networks has trailed the study of regulatory interactions" is somewhat confusing. Clear definitions of signaling networks and regulatory interactions would be helpful.

4. Supplementary Fig 1c, please clarify whether it is true that all connections for a kinase will be on the positive side, and for a phosphatase will be on the negative side.

5. mVESPA infers both phosphosite-specific and whole-protein signalons. Why whole-protein signalons are needed? Is it used in downstream analysis? If so, is it recommended to include this in other applications?
6. Line 197-199. "Kinase inhibitors and other drugs inhibit the activity of their targets without affecting their phospho-state but rather by binding to the protein's active site." I think many kinases are activated by phosphorylation, and thus the kinases' active sites are closely related to their phosphostates. Please clarify.
7. Line 218-220, cite a reference on the KP-enzyme fraction expected to be expressed in any specific cellular context.
8. One of the most exciting results is that "VESPA could measure enzymatic activity for 158 of 371 (42.6%) of all KP-enzymes in the CRC SigNet that completely lacked phosphostate information thus almost doubling the amount of critical available information.". It would be great to specifically assess the quality of these inferences. Are these inferences as good as the ones made for KP-enzymes with phosphostate information?
9. The conclusion "These analyses show the effect of both each individual improvement of mVESPA as well as their cumulative effect, the latter resulting in the best overall performance and a significant improvement over the current state-of-the-art (Fig. 2d)". It is not clear to me what is the state-of-the-art algorithm here. Please clarify. It would be helpful to include a few commonly used methods in this benchmarking, e.g., KSEA + PSP/NetWorkin substrates, as well as the recently published Kinase library-based activity inference (<https://www.nature.com/articles/s41586-022-05575-3>). Please also report the AUROC values.
10. Please describe MSI status of the cell lines, and check whether it is consistent with the corresponding subtypes.
11. It is unclear what is the relationship between the 8 distinct clusters defined in reference 39 and the CMS, and why the former was used to identify cell lines and the later was used for analysis in Figure 2.
12. I don't understand why data was generated for 7 time points (which I consider a key strength of the study) and only a reduced dataset with 3 time points was used for dVESPA for SigNet inference. Early time points should be very informative for direct interactions.
13. Is it possible to quantify the inhibition effect in Figure 5? At least highlight the cell lines in which a target is considered effectively inhibited. It would also be useful to highlight which are considered as adaptation activation (as suggested by the authors), and which as non-responsive. It would also be useful to discuss what is the expected timeframe of response (both inhibition and adaptation activation), e.g., minutes, hours, or days.

Reviewer #3 (Remarks to the Author): Expert in functional cancer genomics, drug and CRISPR screening

Rosenberger et al. addresses an important challenge, identifying signaling pathways underlying drug response / resistance, through the generation of novel phosphoproteomics datasets and methods. The reconstruction of kinase-substrate networks is a longstanding challenge. Novel large-scale phosphoproteomics datasets and methodologies are important and of general interest to the community. This work is comprehensive, rigorous and necessary support for the conclusions and claims

is provided, although their comparison with other independent studies and methods is limited.

The methodology utilized in this study is interesting, as it incorporates several techniques from gene regulatory networks while also making appropriate statistical modifications to accommodate the complexities of incomplete and continuous phosphoproteomics measurements. However, the current version of the manuscript does not provide sufficient evidence to demonstrate the distinct advantages of this approach in addressing the research questions posed, nor does it adequately compare its results with those of more recent investigations and datasets.

I believe the following points could improve the manuscript:

1. As previously mentioned, several other approaches exist to infer kinase-substrate networks and the authors allude to them in the introduction but do not make any strong comparison with existing independent approaches. Particularly, it would be interesting to compare and integrate a recent large-scale reconstruction of kinase networks in cancer cells using phosphoproteomics (PMID: 31959955).
2. Baseline phosphoproteomics of cancer cell lines and tumor patient samples is highly confounded with protein abundance. Have the authors tried to account for this, for example, by removing (regressing-out) protein abundance from each phosphosite measured to see the potential impact in their results?
3. The manuscript is at times dense and hard to follow. It would greatly benefit from significant simplifications of the text. Even for experts in the field the terminology and complexity of the sentences become too confusing. For example, this sentence in Figure 1a) legend "The signaling network reconstruction module uses this matrix together with optional priors from reference networks to assess regulatory relationships by computing the mutual information between enzymatic regulator and target phosphopeptides or phosphosites and the signal transduction Data Processing Inequality (stDPI) to generate signalons for each regulator consisting of interaction probabilistic weight and mode of regulation (kinase activation: red, phosphatase deactivation: blue) with substrate targets."
4. Whilst this is provided in the text, it is important to provide more details in the figures, for example, quantify AUC in the ROC curve plots; Fig2 a), spearman correlation between what? This information would help follow the manuscript.
5. Fig 2b, while the approach outperforms the other methods the AUC is still modest (stDPI = 0.693). Is the benchmark dataset balanced?

Reviewer #4 (Remarks to the Author): Expert in colorectal cancer genomics, functional genomics, drug and CRISPR screening

The authors develop a phospho-proteomic methodology to infer the activity of kinases and phosphatases in proteomic datasets by utilising existing and developing novel inferencing methods on

existing proteomics and phosphoproteomics data. These methods are applied in a context-dependent manner to identify context-specific K/P-substrate relationships, specifically in colorectal cancer as demonstrated by the authors. The methodology is timely, conceptually interesting and will add a new tool to interrogate adaptive and context-dependent signaling pathways in cancer and other disease. However, the biological and translational impact of these studies seem somewhat premature and in its current state, the paper may be better suited for a bio-informatics specialty journal.

Specific comments below:

Validation using only 6 CRC cell lines is limiting. Can findings be extended to patient derived samples such as CRC organoids and more diverse cell line panels beyond CRC.

Furthermore, it is stated that: HCT-15, HT115, LS1034, MDST8, NCI-H508 and SNU-61 are representative CRC cell lines. Can the authors be more specific on how these cell lines are chosen and their mutational profiles (e.g. KRAS, BRAF mutations etc).

Availability of phosphoproteomic data across hundreds of cell lines - CPTAC data across multiple lineages are also available to infer lineage-specific kinase-substrate networks. It would be useful to extend validation studies to 1-2 additional cancer types to broaden the validity of the model.

How do the contextual effects of drugs relate to sensitivity? For example, in Figure 6 where Osimertinib is highlighted, were HCT-15 and HT115 differentially sensitive to the drug? How did the cell line specific factors identified (BUB1, ERBB2, LYN, PRKCZ) relate to response of each line to Osimertinib? Addition of the CRISPR study here would be of great interest to increase the relevance of these findings. Similarly, for ralimetinib, continuing these as specific examples to demonstrate biological relevance would be interesting.

For figure 7b, is there an underlying reason behind the relatively poorer performance in NCI-H508 compared to HCT-15? Is this a function of the drug concentration chosen for the screen or model performance?

Can the authors make a comment on the false negatives identified through CRISPR screens in supp figure 18-19.

In supp figure 18-19 a head-to-head comparison to VESPA activity and peptide intensity values for identifying resistance factors (vs CRISPR ground truth) would clarify the increase in predictability from using this model.

Figure 7, Supplementary Figure 18 and Supplementary Figure 19 are a bit confusing to understand. Would be helpful to show a volcano plot (e.g. logFC or Mageck beta scores drug vs control along the x-axis), and log-transformed p values along the y axis, with kinases predicted to be drivers of resistance marked out by a distinct colour.

Network-based elucidation of colon cancer drug resistance by phosphoproteomic time-series analysis

Point-by-point response

George Rosenberger^{1,†}, Wenxue Li^{2,†}, Mikko Turunen^{1,†}, Jing He^{1,3,†}, Prem S Subramaniam¹, Sergey Pampou^{1,4}, Aaron T Griffin^{1,5}, Charles Karan^{1,4}, Patrick Kerwin¹, Diana Murray¹, Barry Honig^{1,6,7,8}, Yansheng Liu^{2,9}, Andrea Califano^{1,4,6,7,10,11}

- 1 Department of Systems Biology, Columbia University Irving Medical Center, New York, NY, USA
 - 2 Yale Cancer Biology Institute, Yale University, West Haven, CT, USA
 - 3 Present address: Regeneron Genetics Center, Tarrytown, NY, USA
 - 4 J.P. Sulzberger Columbia Genome Center, Columbia University Irving Medical Center, New York, NY, USA
 - 5 Medical Scientist Training Program, Columbia University Irving Medical Center, New York, NY, USA
 - 6 Department of Medicine, Columbia University Irving Medical Center, New York, NY, USA
 - 7 Department of Biochemistry & Molecular Biophysics, Columbia University Irving Medical Center, New York, NY, USA
 - 8 Zuckerman Mind Brain and Behavior Institute, Columbia University, New York, NY, USA
 - 9 Department of Pharmacology, Yale University School of Medicine, New Haven, CT, USA
 - 10 Herbert Irving Comprehensive Cancer Center, Columbia University Irving Medical Center, New York, NY, USA
 - 11 Department of Biomedical Informatics, Columbia University Irving Medical Center, New York, NY, USA
- † Equal Contribution

Correspondence to: yansheng.liu@yale.edu and ac2248@cumc.columbia.edu

Reviewer Comments

Reviewer #1 (Remarks to the Author): Expert in computational systems biology, signalling networks, and phosphoproteomics

In this manuscript, Rosenberger and colleagues present a system biology method, which they named VESPA, that takes phosphoproteomics data and knowledge of kinase-substrate interactions to infer changes in kinase networks induced by perturbations. The method was tested with the analysis of a phosphoproteomic time series dataset from colorectal cancer cells treated with kinase inhibitors. The adaptive response that this analysis suggested was experimentally investigated using a CRISPR ko assay. While the topic and approaches are of interest to the field of kinase signaling and cancer therapeutics, the low perceived novelty of the study, and the unproven accuracy of the approach limit enthusiasm for this work.

Specific comments

1. The main issue with this study is that VESPA does not seem to perform well. From the perturbation data in Figure 5, one would conclude that the method fails to quantify changes in kinase activities induced by kinase inhibitors. PIK3CA, EGFR and MAP2K1/2 scores seem to increase or remain unchanged by treatments with the respective inhibitor in most cases.

We apologize for any confusion in the presentation, as the original benchmarks, which are now significantly extended, combined with our orthogonal CRISPRko validation experiments, show that VESPA outperforms previously published algorithms. Our comparisons, especially considering the new benchmark approaches, show that VESPA surpasses all competing reference networks when analyzing large-scale cohorts representing a common cellular context (lines 294-344, Fig. 2d) and performs competitively with other algorithms even when independent single sample analyses are conducted (lines 345-386, Supplemental Fig. 5). Further, our experimental validation results are based on actual experimental observations and are consistent with the highly heterogeneous response of cells to pharmacologic perturbations.

A possible confusion in the interpretation of the result is that the drug perturbation experiment was designed to study drug mechanism of action and cell adaptive resistance mechanisms under physiologically relevant drug perturbations. Specifically, consistent with several other publications by our lab (Mundi *et al.* 2023 Cancer Discovery; Obradovic *et al.* 2023 Cancer Cell; Vasciaveo *et al.* 2023 Cancer Discovery; Alvarez *et al.* 2018 Nat Genet), we used sub-lethal drug concentrations to avoid confounding effects arising from cell stress or death pathway activation. It is thus unreasonable to expect that every primary drug target will show a clearly measurable response upon drug perturbation. However, this experimental design allows studying adaptive response and network rewiring also for perturbations where the primary drug target was not strongly affected, as shown throughout the analysis and experimental validation sections.

We agree with the reviewer that the current experimental design—based on single, low dosages drug perturbations—is not ideally suited to differentiating between a biological (insufficient dosage to measure primary target perturbation) and a technical (VESPA hypothetically not performing well) rationale. Thus, to assess whether VESPA has specific biases or limitations for the specific kinases highlighted by the reviewer (i.e., PIK3CA, EGFR, and MAP2K1/2), we conducted an additional analysis, applying the main mVESPA benchmark to only those four targets (Fig. R1). This analysis shows clear selectivity and sensitivity of the algorithm to differential PIK3CA, EGFR

and MAP2K1/2 activity. Taken together, the proposed additional analysis shows that VESPA provides improved performance to competing algorithms and reference networks for these selected targets. We also conclude that several of the drug perturbations we performed using drugs targeting these proteins did not fully abrogate PIK3CA, EGFR, and MAP2K1/2 activity, which was explicitly not the intention of our experimental design.

Figure R1: mVESPA benchmark limited to comparisons targeting PIK3CA, EGFR and MAP2K1/2. The Pseudo-ROC network is generated comparing the GDSC reference drug sensitivities and differential comparisons of the U54BL dataset. Details are provided in the main text and methods section.

2. Another issue is that the study does not seem to be overtly novel, and ignores several papers that are a precedent to this work. For example:

- When discussing the study of cancer resistance mechanisms, the authors mention that “so far the focus has been on the discovery of genetic events leading to selection of drug resistant clones”. Contrary to this claim, there are several papers that have used phosphoproteomic approaches to investigate drug response mechanisms (e.g. PMID: 36688815) and adaptation mediated drug resistance (e.g., PMID: 26060313).

We have clarified the citation from the review article linked in the main text to highlight that several prior studies focused primarily on genetic events. We have also added the two proposed references as examples of more recent network-based adaptation studies (lines 637-641). PMID36688815 was published after the initial submission of our manuscript and could thus only be added now.

- The comment about “systematic, proteome-wide elucidation of tissue-specific signaling networks has trailed the study of regulatory interactions” (page 3), while technically correct, ignores important contributions to this field (e.g., PMID: 27909043)

We have added the references to the main text (lines 51-56).

- The statement “phosphoproteomic data for all CCLE cell lines was not available” is inaccurate because a relatively comprehensive CRC phosphoproteomics data has been available for this purpose for several years (PMC5583477)

We originally investigated whether the referenced dataset from Roumeliotis *et al.*, could be used for several aspects of our analysis. Unfortunately, we found the quantitative consistency to be insufficient for the quantitative comparisons conducted by VESPA. Although Roumeliotis *et al.* acquired their dataset by a similar label-based quantification strategy as CPTAC, we found both proteome coverage and quantification consistency to be substantially lower than in the CPTAC-S45 and U54BL datasets. This is illustrated by two key characteristics when comparing 100 randomly sampled pairs of two samples. Figure R2 illustrates the substantially lower number of shared phosphosites for Roumeliotis *et al.*, compared to U54BL and CPTAC-S45. Figure R3 illustrates the substantially lower data completeness expressed as ratio between phosphosites measured in one sample versus the total. Adjusting for label batches unfortunately substantially lowered the number of valid comparisons that could be drawn, thus limiting the usefulness of the dataset for this study. Since substrate identification is critical to the performance of VESPA analyses, lack of completeness represented a substantial challenge to the use of these data.

Figure R2: Density distribution of absolute number of overlapping phosphosites between 100 randomly sampled binary comparisons for each dataset.

Figure R3: Relative data consistency for phosphosite quantification between 100 randomly sampled binary comparisons for each dataset. Due to the label-based approaches used by the 50 COAD and CPTAC-S45 studies, some binary comparison generate perfect overlaps of identified sites.

- The claim that this work “resulted in one of the largest context-specific targeted drug perturbation phosphoproteomic datasets” ignores studies, such as PMID: 31959955 and PMID: 36926954, which profiled larger number of drug perturbed phosphoproteomes than in this study.

We now acknowledge this previous work and have added the two references (lines 751-755). In particular, PMID36926954 was published after the initial submission of our manuscript and could thus only be added now.

3. There are contradictions in the approach used by the authors. A feature of dVESPA is that it eliminates indirect ES interactions, yet in order to estimate tyrosine kinase activity, the authors seem to rely on measuring the activity of Ser/Thr kinases that are downstream of tyrosine kinases. In other words, the approach relies in measuring indirect ES interactions to estimate tyrosine kinase activation.

We apologize for the confusion. The **direct substrates** of S/T kinases are first used to estimate their activity. Then the **direct substrates** of a tyrosine kinase are used to assess its activity, based on their activity rather than their phosphostate. So, the only difference in the second step is that instead of using the phosphostate of a kinase's substrates we use their activity, as inferred from their direct substrates. Using stDPI and DPI, dVESPA attempts to find the most direct interactions, however, due to incomplete phosphoproteome coverage of all high-throughput assays, a certain fraction of indirect interactions will always remain present. We have now improved the description of the method and its limitations in the manuscript (lines 147-164, 174-183).

4. It is not clear how the method quantifies PIK3CA activity, as shown in Figure 5, if indirect ES interactions are eliminated from the analysis. PIK3CA is a lipid kinase without significant protein kinase activity. This is indeed a puzzling feature of the approach given that one of the steps of the method is to eliminate indirect interactions (page 13, line 267).

dVESPA is designed to eliminate indirect interactions via the data processing inequality (Methods, lines 1170-1191). However, if one of the two interactions forming an indirect interaction cannot be detected, then the indirect interaction will emerge as the most statistically significant. For instance, if $A \rightarrow B$ and $B \rightarrow C_i$ are direct interactions inducing an indirect but significant interaction between A and C AND either $A \rightarrow B$ or $B \rightarrow C_i$ are undetectable, then the indirect $A \rightarrow C_i$ interactions will be identified by the algorithm. Although these interactions are indirect, they can be useful for activity inference because they represent the least indirect path between an enzyme and its downstream effectors, as supported by the data. This is analogous to the signatures used by algorithms such as PTM-SEA. We have modified the main text to better highlight the relative nature of direct vs. indirect interaction removal (lines 155-160).

Basically, the DPI does not guarantee that only direct interactions are identified but rather that only the "least indirect" interactions are identified among those detectable by mutual information analysis. While this eliminates most indirect interactions, many signalons will still contain some fraction of indirect interactions; these are, however, still highly reflective of the information flow in signaling networks and can be effectively used to infer enzymatic activity.

We thus agree with the reviewer that the signaling activity of PIK3CA is not well characterized, having no associated substrates in databases, such as PhosphoSitePlus or Johnson *et al.*, 2023. To assess how dVESPA generated the signalon for PIK3CA, we investigated the K/P-S interactions in more detail. In total, 53 substrates were found with varying likelihood and strength of mode of regulation. Comparing this set of substrates to the predicted K/P-S relationships of LinkPhinder, we found an overlap of 24. Further, the LinkPhinder confidence score distribution of VESPA-identified (24) vs. non-identified hits (2271), suggests that the VESPA-identified K/P-S relationships have a higher score distribution (KS-test; $p = 0.016$).

We hope that future studies by Johnson *et al.* will also include kinases such as PIK3CA and provide experimental evidence for inferred interactions of less well investigated K/P-enzymes. In absence of these reference points, we believe that the overlap with prediction databases at least

supports that the inferred interactions are representative of associated functions or mechanisms, even though the degree of directness could not yet be assessed.

5. The manuscript is riddled with exaggerations and overblown comments. For example, the authors claim that they “designed a comprehensive longitudinal drug perturbation experiment (page 19)”. This is an exaggeration because the experiment used seven compounds at single dose for seven time-points, which is clearly not comprehensive. Similarly, the claim that mVESPA likely quantified “an almost complete subset of the KP-enzymes expressed in CRC cells” is overly speculative and another exaggeration. It would be relatively trivial to determine whether this is correct by simply comparing the results with databases of expression data.

We apologize if the claim that our study is comprehensive was deemed offensive. We did not mean that the study covered all possible cell lines, drugs, time points, and concentrations, a feat that would not be possible in any case, but only that it covered a large subset of these possible cases. We have thus modified the text by removing the word “comprehensive” (lines 454-457).

The study in PMID36926954, which was published after our initial submission, comprises a larger number of tested drug compounds, cell lines, and perturbation time points. As a result, our dataset is no longer the largest in terms of tested conditions. Yet, within the specific context of colon cancer, our study remains one of the largest of its kind, as we are not aware of published colon cancer datasets providing the same combination of phosphoproteome coverage and quantitative consistency. We thus hope that the reviewer will find the revised description more acceptable.

With respect to the second point by the reviewer, We have compared the inferred set of K/P-enzymes with the expected number of expressed K/P-enzymes and clarified the statement in the main text (lines 236-241, Methods), specifically:

“Overall, despite the well-known sparseness of peptides and phosphopeptides detected by proteomic assays, mVESPA quantitatively assessed the activity of 371 KP-enzymes—*i.e.*, around half of all known human KP-enzymes and around 66.7% of the KP-enzymes estimated to be expressed in CRC cells (Methods). In contrast, phosphostate information was available for only 42.7% of expressed KP-enzymes.”

As a result of this analysis, we agree with the reviewer that the statement can be misleading. We have thus revised it to state that we characterized the majority of K/P-enzymes that could be detected by phosphoproteomic assays, representing about 43% of all K/P-enzymes.

6. It is claimed “that VESPA substantially outperforms established approaches” (page 5) but a direct comparison with those other approaches is not clear from reading the paper. To benchmark mVESPA the authors compare its performance with the method in reference 19 but the data on which of these methods performs best is not shown. Similarly, in the temporal analysis shown in figure 6, it is not clear whether established methods to quantify kinase activity from phosphoproteomics data would produce similar or more reliable results. In addition, the correlation between sensitivity to kinase perturbation and cognate kinase activity (as measured by mVESPA) does not seem to be significant for most cell lines. It would be interesting to see if established methods to infer kinase activities from phosphoproteomics data perform similarly to mVESPA in this regard.

We agree with the reviewer that a more comprehensive benchmarking of the algorithm is required to compare it to preexisting methods. As a result, we have significantly increased the breadth of the original benchmark by leveraging recently published methodologies and databases.

To assess the performance of the full algorithm, including the reverse engineering component, we compared its performance to using large, cell context-free reference networks published by Hijazi *et al.* 2020, and Johnson *et al.* 2023. Even though these networks are substantially more comprehensive than previous databases, such as PhosphoSitePlus, the use of dVESPA-derived interaction networks significantly improved performance and coverage.

In addition, to show that mVESPA performs at least as well or better than other algorithms proposed for the analysis of signaling networks we leveraged the recently published KSTAR benchmark (Crowl *et al.*, 2022) to compare mVESPA to a number of established algorithms. This has allowed comparing mVESPA to KSTAR, KSEA, PTM-SEA, KARP and KEA3. For these comparisons, the dVESPA-based crosstalk correction and hierarchical inference could not be applied due to sample size. Despite this critical limitation, which negatively affects VESPA performance, the analysis still shows that mVESPA outperformed the other algorithms, except for TKs, where mVESPA was applicable only in combination with the PhosphoSitePlus reference network (lines 345-386, Supplemental Fig. 5). It should be noted though, that the optimal performance can only be achieved when dVESPA and mVESPA can be used together.

Regarding the correlation between kinase perturbation sensitivity and cognate kinase activity, we again refer to the experimental design of our study, which was geared towards identifying resistance factors and investigating cell adaptive response. For this reason, we do not expect the primary targets of each targeted inhibitor to be silenced to the levels associated with the compound IC50 but rather for their activity to be modulated, thus allowing detection as well as triggering of adaptation and drug resistance mechanisms. Indeed, VESPA-based analysis of our dataset helped identify resistance factors and mechanisms of adaptive response, which were either independently reported by other studies or were validated using the CRISPR/Cas9-mediated KO experimental design at a statistically significant level.

7. The supplementary tables and datasets are impossible to review because there are no captions or any other type of explanation (meaning of headings, etc.) on the data that these files contain.

We apologize for this issue. We now provide Microsoft Excel XLSX-type supplemental tables that include a description of the data on the first page.

8. There are not details on the source of KP-enzyme direct and indirect substrate relationships used in the study.

We are not sure if we understood the concern of the reviewer correctly. dVESPA does not use any previously reported K/P-S interactions. Rather, KP-S interactions are inferred *de novo* by analyzing large-scale, cell context-specific phosphoproteomics profiles. The specific KP-S interactions inferred by dVESPA and used by mVESPA to infer kinase activity are included in Supplemental Data 1 and we now provide additional networks for several CPTAC datasets on the Zenodo repository linked in "Data Availability".

We have shown that existing K/P-S reference databases (e.g. HSM, LinkPhinder or similar) can be integrated with dVESPA predictions (lines 161-164). However, our benchmarks show that the optimal performance is achieved when dVESPA networks are used natively and not integrated with prior databases (Supplemental Fig. 2-3). It is, of course, possible to only use mVESPA with any signaling network (e.g., PhosphoSitePlus or Johnson *et al.*, 2023) (Fig. 2d, Supplemental Fig. 5). However, VESPA's performance is significantly degraded when dVESPA-inferred networks are not used.

Reviewer #2 (Remarks to the Author): Expert in cancer proteomics, bioinformatics, and colorectal cancer

This manuscript presents VESPA (Virtual Enrichment-based Signaling Protein activity Analysis), a computational method for dissecting enzyme-substrate interactions and measuring signaling protein activity. The authors also applied VESPA to investigate post-translational cell adaptation mechanisms associated with colorectal cancer's resistance to targeted drugs. There are multiple strengths of the manuscript, including its focus on an important and challenging question in phosphoproteomics data analysis, a computational algorithm built upon the widely used VIPER algorithm previously developed by the group, a unique and large context-specific targeted drug perturbation phosphoproteomic dataset with 354 phosphoproteomic profiles to demonstrate the application of the algorithm, and experimental validation of the hypotheses generated from the exploratory study. The manuscript should be of interest to bioinformatics, phosphoproteomics, and cancer research communities.

Major concerns:

1. The primary concern I have is the complexity of the manuscript. The authors have undertaken a substantial amount of work, which I believe warrants two separate publications. The first publication could focus on the development, evaluation, and benchmarking of the computational method. The second publication could center on the application of the already benchmarked method to the analysis of drug resistance in colon cancer. This division would make the manuscript less dense and more reader-friendly, while allowing for a more comprehensive description of the computational method and its potential applications.

We agree with the reviewer that the manuscript has become complex. We will take up and discuss this issue with the editor. We have also tried to simplify the manuscript components, as suggested by all reviewers, and to provide additional details about the methodology in the materials section.

The major issue is that the performance of VESPA, as further shown in this resubmission, is inextricably related to both of its components. As a result, it is difficult to find an appropriate way to split the manuscript.

In particular, I recommend expanding the computational section to include descriptions of additional potential use cases for the VESPA algorithm. It would be helpful to clarify the required and optional input files for VESPA, such as whether it necessitates a CPTAC-type dataset and a time course phosphoproteomics dataset. Additionally, addressing whether VESPA is applicable to a single case-control phosphoproteomics dataset and how its performance compares to other algorithms mentioned by the authors would be valuable information for potential users. Providing such details could greatly enhance the impact of the algorithm and increase its usability.

We agree with the reviewer that potential users will be interested in the applicability of VESPA to scenarios outside these boundaries. We have thus added a new section “Applicability to different dataset types”, where we describe the requirements in more detail (lines 187-197).

In addition, to further support the performance of the algorithm, we have significantly extended the benchmarks and comparison to competing algorithms. First, we show that when the dVESPA network is replaced with recently published, large, cell context-free reference networks—including by Hijazi *et al.* 2020, and Johnson *et al.* 2023—mVESPA performance is significantly degraded.

Indeed, even though these networks are substantially more comprehensive than previous databases, such as PhosphoSitePlus, dVESPA still provides significantly better performance and coverage. Then, we leveraged the published KSTAR benchmark (Crowl *et al.*, 2022) to compare mVESPA to KSTAR, KSEA, PTM-SEA, KARP and KEA3. In this benchmark, we could not use dVESPA-inferred networks because the dataset size was not sufficient to support the information theory-based inference. Despite this critical limitation, which negatively affects VESPA performance, the analysis still shows that mVESPA outperformed the other algorithms, except for TKs, where mVESPA was applicable only in combination with the PhosphoSitePlus reference network (lines 345-386, Supplemental Fig. 5). Taken together, these two benchmarks show that the combination of mVESPA and dVESPA substantially outperforms any of the previously published algorithms. This is largely due to the cell context-specific network inference by dVESPA, even though mVESPA also provides state-of-the-art performance.

We hope the reviewer will agree that the results of these two independent benchmarks justify combining the two algorithms into a single publication and that VESPA outperforms existing methods when context-specific networks can be reconstructed using suitable disease- or condition-specific sample cohorts.

If the authors choose to maintain the current arrangement, I suggest simplifying the text to make it more accessible to the broader audience of the journal. Some content in the results section could be moved to the methods section.

We agree with the reviewer that the manuscript has become complex. We have tried our best to simplify the manuscript by moving critical but less foundational details to the methods section.

2. Another concern pertains to the normalization of phosphoproteomics data using global proteomics data. In the manuscript, the authors introduced a second CPTAC dataset where the CPTAC-S045 phosphosite abundances were normalized by the corresponding whole protein abundances, with the goal of addressing confounded kinase-phosphosite relationships. However, it is unclear which version of the data was ultimately used for downstream analysis. The authors observed that the normalized version yielded fewer optimized signalons. Does this reduction indicate that the normalization procedure eliminated biological signal? Additionally, if normalization is crucial, why was it not employed in the cell line study? The question of whether and how to normalize phosphoproteomics data remains unresolved, and I hope the authors could use their unique methods and datasets to shed light on this important question.

This is an important issue and we have clarified the rationale and idea behind all components employed by our approach in more detail in the methods section (lines 1082-1113):

“Protein abundance level normalization can be very important for clinical datasets, such as CPTAC, where some proteins might have more variable protein abundance distributions, which could potentially confound corresponding phosphopeptide abundances and thus lead to wrong associations of co-regulated proteins. VESPA can incorporate protein abundances at three different stages of the workflow:

Protein abundance normalization: Of note, as opposed to CPTAC samples, we did not measure baseline peptide abundances suitable for protein abundance inference for the U54 samples, since this would have doubled the number of required LC-MS/MS runs. However, by normalizing each drug-treated sample against the corresponding vehicle control-treated ones, we expect that this will not significantly affect results. Further, since

most signalons comprise of phosphosites representing dozen to hundreds of independent phosphoproteins, the analysis is robust against changes in protein abundance of individual proteins. As such, we recommend using protein abundance-normalized profiles only if the investigated mechanisms are expected to be extremely confounded, e.g. the auto-phosphorylation feedback loops of tyrosine kinases when comparing drug perturbed to baseline samples.

Protein abundance as proxy for KP-enzyme signalons: Instead of using KP-enzyme phosphopeptides as proxy for the enzyme component of the signalon, protein abundances can optionally be used for the inference of signaling networks. This is particularly useful when variability in gene expression influences the activity of a KP-enzyme, for which phosphopeptides were not measured, for example tyrosine kinases. This mode should thus only be used instead of “protein abundance normalization” described above, but not together. The file format is similar to PVM; however, the columns “peptide_id” (free text unique protein identifier), “site_id” (unambiguous combination of gene_id, protein_id and “PA” (protein abundance), separated by “:”, e.g. “EGFR:P00533:PA”), “modified_peptide_sequence” (free text unique protein identifier), “peptide_sequence” (free text unique protein identifier), “phosphosite” (“PA” (protein abundance)), and “peptide_intensity” (float log2-transformed protein intensity from upstream software) are different.

Signalon optimization: Since protein abundance measurements are themselves noisy and thus have limited accuracy, the normalization step may introduce additional bias. Since dVESPA supports the use of *de novo*, inferred signalons, both protein-abundance normalized and unnormalized phosphoproteomic profiles can be included in the network dissection step, thus allowing optimal signalon selection on an individual KP-enzyme basis.”

For downstream analysis, we used a mixture between all three datasets CPTAC-S045, U54-NET, and CPTAC-S045N, accounting each for 47.2%, 43.4% and 9.4%, respectively. This means, that for 9.4% of all covered kinases and phosphatases, normalizing against protein abundance as confounding factor was beneficial. In contrast, for the other KP-enzymes, either the U54-NET signalons performed better, or the unnormalized signalons of the CPTAC-S045 dataset were less affected by protein abundances and more by potential biases introduced through the normalization procedure (lines 223-231).

3. Benchmarking the MI estimator: Figure 2a, please clarify the ground truth and the benchmarking metric used for benchmarking in the Results section, together with detailed ground truth information and justification of choice.

We have extended and clarified the description of the MI estimator benchmark in the main text (lines 242-263).

4. Benchmarking indirect interaction removal: Figure 2b, as dVESPA can incorporate HSM information, please clarify whether HSM information is used in the dVESPA analysis. Moreover, considering the large number of negative space, is AUROC 0.693 good enough for inference? Adding AUPRC might be helpful? How does these methods compare to commonly used methods for such inference?

We have clarified that dVESPA was run without any prior network information and further modified AUROC analysis to AUPRC comparison to better illustrate the differences between the methods (lines 264-293). Since the positive HSM gold standard is context-unspecific, the benchmark

performance might be lower due to this gold standard artefact, however, we believe that the conclusions for relative comparisons of the methods are fulfilled.

Minor:

1. Figure 1 is not very helpful for understanding the method.

We agree with the reviewer that the original figure might have been too convoluted. We have redesigned Figure 1 and hope that it will better help to understand the methodology.

2. Figure 4 does not really highlight any useful information other than the clusters. The giant heatmap with no clearly visible detail is better suited for the supplement.

We agree that Figure 4 is too complex to directly interpret the data. However, we believe that it can be suitable to understand the experimental design (cell lines, drug perturbations, time points) and the nature of the inferred K/P-enzyme activities. We have modified the figure to include selected examples linked to Fig. 5 to connect these two primary data visualization approaches.

3. Line 52-53, “systematic, proteome-wide elucidation of tissue-specific signaling networks has trailed the study of regulatory interactions” is somewhat confusing. Clear definitions of signaling networks and regulatory interactions would be helpful.

We have clarified this sentence in the main text (lines 51-56).

4. Supplementary Fig 1c, please clarify whether it is true that all connections for a kinase will be on the positive side, and for a phosphatase will be on the negative side.

We have clarified this sentence in the figure legend (lines 954-955).

5. mVESPA infers both phosphosite-specific and whole-protein signalons. Why whole-protein signalons are needed? Is it used in downstream analysis? If so, is it recommended to include this in other applications?

Most downstream applications and data integration methods can be applied only to protein-level metrics and can't make use of phosphosite-level data. Further, while single phosphosites of kinases and phosphatase can sometimes be associated with enzymatic activity, frequently an interplay of multiple factors, including other sites, protein structure, inhibitors and activators needs to be considered. For this reason, the integrated, protein-level summarization of VESPA can be useful to screen global protein activities (lines 1192-1198). We further demonstrate that site-level information is also interesting and can provide even more mechanistic insights but requires further assessment (lines 545-565).

6. Line 197-199. “Kinase inhibitors and other drugs inhibit the activity of their targets without affecting their phospho-state but rather by binding to the protein's active site.” I think many kinases are activated by phosphorylation, and thus the kinases' active sites are closely related to their phosphostates. Please clarify.

We intended to indicate that many kinase inhibitors (e.g. ATP-competitive inhibitors) allosterically modulate the active sites of targets rather than directly dephosphorylate the specific phosphosites. In many cases, the enzymes are thus rendered inactive independent of the abundance of the active phosphosites, although exceptions, including the autophosphorylation

mechanisms of some kinases certainly exist. We have modified the sentence and hope that this clarifies the intended meaning (lines 198-199).

7. Line 218-220, cite a reference on the KP-enzyme fraction expected to be expressed in any specific cellular context.

We have compared the inferred set of K/P-enzymes with the expected number of expressed K/P-enzymes and clarified the statement in the main text (lines 236-241, Methods):

“Overall, despite the well-known sparseness of peptides and phosphopeptides detected by proteomic assays, mVESPA quantitatively assessed the activity of 371 KP-enzymes—*i.e.*, around half of all known human KP-enzymes and around 66.7% of the KP-enzymes estimated to be expressed in CRC cells (Methods). In contrast, phosphostate information was available for only 42.7% of expressed KP-enzymes.”

8. One of the most exciting results is that “VESPA could measure enzymatic activity for 158 of 371 (42.6%) of all KP-enzymes in the CRC SigNet that completely lacked phosphostate information thus almost doubling the amount of critical available information.”. It would be great to specifically assess the quality of these inferences. Are these inferences as good as the ones made for KP-enzymes with phosphostate information?

That is a great suggestion and we have added the following results to the main text (lines 330-339):

“Benchmarking only signalons with U54BL-measured phosphopeptides indicates that VESPA performs very similar on this subset when assessing all kinases in comparison to the full dataset, although with lower sensitivity. Further, it should be noted, that this result could also be confounded due to the bias of the benchmark towards well studied or experimentally better accessible KP-enzymes. While 83.9% of all comparisons of the benchmark cover targets with U54BL-measured phosphopeptides, the fraction of CRC signalons that cover directly measured regulator phosphopeptides is only 57.4%. Interestingly, when only considering TKs, inclusion of signalons without measured regulator phosphoproteins, expectedly increased substantially (Supplemental Fig. 4).”

9. The conclusion “These analyses show the effect of both each individual improvement of mVESPA as well as their cumulative effect, the latter resulting in the best overall performance and a significant improvement over the current state-of-the-art (Fig. 2d).”. It is not clear to me what is the state-of-the-art algorithm here. Please clarify. It would be helpful to include a few commonly used methods in this benchmarking, e.g., KSEA + PSP/NetWorkin substrates, as well as the recently published Kinase library-based activity inference (<https://www.nature.com/articles/s41586-022-05575-3>). Please also report the AUROC values.

We agree with the reviewer, that potential users will be interested in the applicability of VESPA to scenarios outside these boundaries. We have added a new benchmark and comparison to competing algorithms that covers the single case-control use case. Using the KSTAR benchmark (Crowl *et al.*, 2022), we have added a comparison to KSTAR, KSEA, PTM-SEA, KARP and KEA3. Although the dVESPA, crosstalk correction and hierarchical inference methods could not be applied, we demonstrate competitive or superior performance of mVESPA with reduced functionality compared to competing methods, as assessed by the performance metrics proposed by the KSTAR authors (lines 345-386, Supplemental Fig. 5). We have further extended the mVESPA benchmark to test the large context-free reference networks by Hijazi *et al.* 2020, and

Johnson *et al.* 2023. Even though these networks are substantially more comprehensive than previous databases such as PhosphoSitePlus, dVESPA provides significantly better performance and coverage. We thus believe that these two independent comparisons, and particular the independent KSTAR benchmark demonstrate the improvements of VESPA over preexisting methods, when context-specific networks can be reconstructed using suitable disease- or condition-specific sample cohorts.

10. Please describe MSI status of the cell lines, and check whether it is consistent with the corresponding subtypes.

We have added information regarding MSI status of the cell lines and CPTAC samples to Figure 3. The information is consistent with the three subtypes, where VC2 consists of only MSS and VC1 and VC3 consist of MSI-high type cell lines and clinical samples.

11. It is unclear what is the relationship between the 8 distinct clusters defined in reference 39 and the CMS, and why the former was used to identify cell lines and the later was used for analysis in Figure 2.

Based on our previous study (Paull *et al.*, 2021 Cell), classifying clinical CRC subtypes into 8 distinct clusters based on a multi-omic data integration framework, we selected the six cell lines as most representative models. As independent validation of our selection, we further applied the CMS classification system on the CPTAC and cell line transcriptome profiles, however, we consider our previous classification as more fine-grained, due to the additional incorporated data.

12. I don't understand why data was generated for 7 time points (which I consider a key strength of the study) and only a reduced dataset with 3 time points was used for dVESPA for SigNet inference. Early time points should be very informative for direct interactions.

The information theoretic framework of dVESPA requires independent data points to work best. This assumption is given for clinical cohorts, however, for our time series experiment, samples are processed and perturbed together, thus close individual time points are closely correlated. The reduction to only three time points, which are more distant in time provides more independent data points.

13. Is it possible to quantify the inhibition effect in Figure 5? At least highlight the cell lines in which a target is considered effectively inhibited. It would also be useful to highlight which are considered as adaptation activation (as suggested by the authors), and which as non-responsive. It would also be useful to discuss what is the expected timeframe of response (both inhibition and adaptation activation), e.g., minutes, hours, or days.

mVESPA quantifies K/P-enzyme activities using the aREA framework which is conceptually similar to GSEA but can also account for directionality of interaction and probabilistic weight. Inferred activities are reported as z-scores or normalized enrichment scores (NES), which allow to identify differential activities but can not provide absolute parameters such as those used in the field of enzyme kinetics.

For our main drug perturbation experiment, it is important to consider that the primary goal of our study was to study adaptive resistance mechanisms, which develop as part of low concentration drug perturbations rather than extremely strong knock-out perturbations of primary drug targets. For this reason, we do not expect all primary drug targets to show a clearly measurable response upon drug perturbation. Because the primary target responses visualized in Figure 5 show

perturbation profiles which are (expectedly) different to complete knock-out profile of kinase activities and the applied drug concentrations were selected to be well below IC20/IC50 levels, we tried to avoid arbitrary classification based on strict NES cutoffs or profiles. For example, Osimertinib targeting BTK and EGFR induces consistent deactivation of the kinases across multiple time points, but typically only with an NES of ~ -1.0 . Trametinib on the other hand deactivates MAP2K1 at time points 6h (MAP2K1) and 24h (HT115) with an NES < -2.0 .

While we believe that our experimental design would not allow us to accurately compare or classify the primary target responses, we are convinced that our data allows to study adaptive response and rewiring also for those data points, where the primary drug target was seemingly not perturbed, as we demonstrate throughout the analysis and experimental validation. However, since these analyses were focused on the comparison of interactions across multiple time points, we decided to highlight only the relevant primary targets in the DeMAND analysis, rather than as part of the raw VESPA activity profiles (Fig. 6).

Regarding the timeframe of adaptive response, we have found substantial heterogeneity between different cell lines. For this reason, we compared only early (5min, 15min, 1h) with late (24h, 28h, 96h) timepoints.

Reviewer #3 (Remarks to the Author): Expert in functional cancer genomics, drug and CRISPR screening

Rosenberger et al. addresses an important challenge, identifying signaling pathways underlying drug response / resistance, through the generation of novel phosphoproteomics datasets and methods. The reconstruction of kinase-substrate networks is a longstanding challenge. Novel large-scale phosphoproteomics datasets and methodologies are important and of general interest to the community. This work is comprehensive, rigorous and necessary support for the conclusions and claims is provided, although their comparison with other independent studies and methods is limited.

The methodology utilized in this study is interesting, as it incorporates several techniques from gene regulatory networks while also making appropriate statistical modifications to accommodate the complexities of incomplete and continuous phosphoproteomics measurements. However, the current version of the manuscript does not provide sufficient evidence to demonstrate the distinct advantages of this approach in addressing the research questions posed, nor does it adequately compare its results with those of more recent investigations and datasets.

I believe the following points could improve the manuscript:

1. As previously mentioned, several other approaches exist to infer kinase-substrate networks and the authors allude to them in the introduction but do not make any strong comparison with existing independent approaches. Particularly, it would be interesting to compare and integrate a recent large-scale reconstruction of kinase networks in cancer cells using phosphoproteomics (PMID: 31959955).

We fully agree with the reviewer. To address this issue, we have added a new benchmarking section to compare VESPA to other published algorithms. Specifically, we used the KSTAR benchmark (Crowl *et al.*, 2022), to compare VESPA to KSTAR, KSEA, PTM-SEA, KARP and KEA3. Although the dVESPA, crosstalk correction and hierarchical inference methods could not

be applied, for reason explained above, we demonstrate competitive or superior performance of mVESPA with reduced functionality compared to competing methods, as assessed by the performance metrics proposed by the KSTAR authors (lines 345-386, Supplemental Fig. 5). We have further extended the mVESPA benchmark to test the large context-free reference networks by Hijazi *et al.* 2020, and Johnson *et al.* 2023. Even though these networks are substantially more comprehensive than previous databases such as PhosphoSitePlus, dVESPA provides significantly better performance and coverage. We thus believe that these two independent comparisons, and particular the independent KSTAR benchmark demonstrate the improvements of VESPA over preexisting methods, when context-specific networks can be reconstructed using suitable disease- or condition-specific sample cohorts.

2. Baseline phosphoproteomics of cancer cell lines and tumor patient samples is highly confounded with protein abundance. Have the authors tried to account for this, for example, by removing (regressing-out) protein abundance from each phosphosite measured to see the potential impact in their results?

This is an important issue and we have clarified the rationale and idea behind our approach in more detail in the methods section (lines 1082-1113):

“Protein abundance level normalization can be very important for clinical datasets, such as CPTAC, where some proteins might have more variable protein abundance distributions, which could potentially confound corresponding phosphopeptide abundances and thus lead to wrong associations of co-regulated proteins. VESPA can incorporate protein abundances at three different stages of the workflow:

Protein abundance normalization: Of note, as opposed to CPTAC samples, we did not measure baseline peptide abundances suitable for protein abundance inference for the U54 samples, since this would have doubled the number of required LC-MS/MS runs. However, by normalizing each drug-treated sample against the corresponding vehicle control-treated ones, we expect that this will not significantly affect results. Further, since most signalons comprise of phosphosites representing dozen to hundreds of independent phosphoproteins, the analysis is robust against changes in protein abundance of individual proteins. As such, we recommend using protein abundance-normalized profiles only if the investigated mechanisms are expected to be extremely confounded, e.g. the auto-phosphorylation feedback loops of tyrosine kinases when comparing drug perturbed to baseline samples.

Protein abundance as proxy for KP-enzyme signalons: Instead of using KP-enzyme phosphopeptides as proxy for the enzyme component of the signalon, protein abundances can optionally be used for the inference of signaling networks. This is particularly useful when variability in gene expression influences the activity of a KP-enzyme, for which phosphopeptides were not measured, for example tyrosine kinases. This mode should thus only be used instead of “protein abundance normalization” described above, but not together. The file format is similar to PVM; however, the columns “peptide_id” (free text unique protein identifier), “site_id” (unambiguous combination of gene_id, protein_id and “PA” (protein abundance), separated by “:”, e.g. “EGFR:P00533:PA”), “modified_peptide_sequence” (free text unique protein identifier), “peptide_sequence” (free text unique protein identifier), “phosphosite” (“PA” (protein abundance)), and “peptide_intensity” (float log2-transformed protein intensity from upstream software) are different.

Signalon optimization: Since protein abundance measurements are themselves noisy and thus have limited accuracy, the normalization step may introduce additional bias. Since dVESPA supports the use of *de novo*, inferred signalons, both protein-abundance normalized and unnormalized phosphoproteomic profiles can be included in the network dissection step, thus allowing optimal signalon selection on an individual KP-enzyme basis.”

3. The manuscript is at times dense and hard to follow. It would greatly benefit from significant simplifications of the text. Even for experts in the field the terminology and complexity of the sentences become too confusing. For example, this sentence in Figure 1a) legend “The signaling network reconstruction module uses this matrix together with optional priors from reference networks to assess regulatory relationships by computing the mutual information between enzymatic regulator and target phosphopeptides or phosphosites and the signal transduction Data Processing Inequality (stDPI) to generate signalons for each regulator consisting of interaction probabilistic weight and mode of regulation (kinase activation: red, phosphatase deactivation: blue) with substrate targets.”

We agree with the reviewer that Figure 1 was too complex. We have restructured and simplified the figure to provide a better accessible introduction to the VESPA method. We have also revised the manuscript to be more approachable by moving some complex details to the methods section.

4. Whilst this is provided in the text, it is important to provide more details in the figures, for example, quantify AUC in the ROC curve plots; Fig2 a), spearman correlation between what? This information would help follow the manuscript.

We have added the performance metrics to the panels of Figure 2. We have further extended the legend of Figure 2a to describe to what datapoints the metrics relate:

“MI was measured and Spearman correlation was computed for each KP-enzyme/target pair using data from the CPTAC-S45 dataset (Methods).”

5. Fig 2b, while the approach outperforms the other methods the AUC is still modest (stDPI = 0.693). Is the benchmark dataset balanced?

We have modified the AUROC analysis of Figure 2b to an AUPRC comparison to better illustrate the differences between the methods (lines 282-293). Since the positive HSM gold standard is context-unspecific, the benchmark performance might be lower due to this gold standard artefact, however, we believe that the conclusions for relative comparisons of the methods are fulfilled.

Reviewer #4 (Remarks to the Author): Expert in colorectal cancer genomics, functional genomics, drug and CRISPR screening

The authors develop a phospho-proteomic methodology to infer the activity of kinases and phosphatases in proteomic datasets by utilising existing and developing novel inferencing methods on existing proteomics and phosphoproteomics data. These methods are applied in a context-dependent manner to identify context-specific K/P-substrate relationships, specifically in colorectal cancer as demonstrated by the authors. The methodology is timely, conceptually interesting and will add a new tool to interrogate adaptive and context-dependent signaling pathways in cancer and other disease. However, the biological and translational impact of these

studies seem somewhat premature and in its current state, the paper may be better suited for a bio-informatics specialty journal.

Specific comments below:

Validation using only 6 CRC cell lines is limiting. Can findings be extended to patient derived samples such as CRC organoids and more diverse cell line panels beyond CRC.

We respectfully disagree with this statement. We are convinced that validating reverse engineering algorithms in six cell lines is already above accepted standards. Moreover, cell lines other than colon cancer cannot be used because the primary data used for the predictions was based on one of the most extensive phosphoproteomic datasets generated yet specific to colon cancer. The goal here is not to move to clinical translation but only to introduce an algorithm that, by improving the inference of kinase substrates, can help better characterize kinase activity under a diverse range of perturbations. The validation data is provided as a proof of concept and not as the basis for clinical translation. Much additional work will need to be performed on each individual candidate mediator of resistance before these approaches may be used to improve drug sensitivity in patients.

Furthermore, it is stated that: HCT-15, HT115, LS1034, MDST8, NCI-H508 and SNU-61 are representative CRC cell lines. Can the authors be more specific on how these cell lines are chosen and their mutational profiles (e.g. KRAS, BRAF mutations etc).

Our lab perspective, now supported by many publications, see (Paull *et al.* 2021 *Cell*; Zeleke *et al.* 2023, *Nat Cancer*; Mundi *et al.* 2023 *Cancer Discovery*; Alvarez *et al.* 2018 *Nat Genetics*), is that individual genetic alterations are poor determinants of cellular phenotype and that only the canalization of the effects induced by the full repertoire of mutations by a small subset of proteins dubbed Master Regulators can define the actual molecular and phenotypic cell state.

In particular, the (Paull *et al.* 2021 *Cell*) manuscript, which identifies 8 subtypes of colorectal cancer based on Master Regulator analysis, represents the foundation for the choice of the cell lines used in this study. Specifically, we used our recently published OncoMatch algorithm (Vasciaveo *et al.* 2023 *Cancer Discovery*) to identify CCLE cell lines representing the highest fidelity models for these patient derived subtypes, based on the conservation of their Master Regulator protein activity with those of the patients in the 8 subtypes reported in the *Cell* manuscript. To address the potential confusion, we have described the process in more detail in the main text (lines 387-401).

Availability of phosphoproteomic data across hundreds of cell lines - CPTAC data across multiple lineages are also available to infer lineage-specific kinase-substrate networks. It would be useful to extend validation studies to 1-2 additional cancer types to broaden the validity of the model.

As part of the VESPA tutorial, we applied our algorithm to a CPTAC LUAD dataset (<https://github.com/califano-lab/vespa.tutorial>). This approach is further directly applicable to many other CPTAC datasets following the steps described therein and we provide precomputed signaling networks and VESPA activities for ccRCC, LUAD, PBT, HBV-HCC, LSCC and PDAC CPTAC datasets (see Data Availability). However, we are convinced that data interpretation for these additional datasets would go well beyond the scope of this manuscript.

We are currently also working on other applications of VESPA, including the application to an HCC organoid collection derived from clinically representative samples. However, the HCC

dataset will be released as a separate publication, since it was conducted in collaboration with a different research group.

How do the contextual effects of drugs relate to sensitivity? For example, in Figure 6 where Osimertinib is highlighted, were HCT-15 and HT115 differentially sensitive to the drug?

The experimental design of our study was optimized for our primary goal, which was the identification of molecular resistance mechanisms and adaptive response. In this context, the selected drug concentrations were chosen to be far below otherwise typically applied “knock-out” dosages as selected by typical experiments designed to identify the IC₅₀ concentration of a drug. Indeed, we have shown that to optimally elucidate drug mechanism of action and potential cell adaptive mechanisms, cells must be perturbed using the highest sublethal concentration of a drug, thus avoiding activation of cell stress and cell death pathways that would dramatically compromise the analysis, see (Alvarez *et al.* 2018 Nat Genet; Woo *et al.* 2014 Cell; Mundi *et al.* 2023 Cancer Discovery; Vasciaveo *et al.* 2023 Cancer Discovery; Obradovic *et al.* 2023 Cancer Cell).

For Osimertinib, HCT-15 IC₂₀ and IC₅₀ were determined by us and GDSC to be at 2.55 and 4.29 μ M, respectively. However, FDA sets C_{max} (maximum tolerated serum concentration in vivo) at 1.68 μ M, which was still above our internal threshold concentration of 0.5 μ M, which was also applied in this combination. For HT115, which displayed a very different response than HCT-15 in our analysis (Fig. 6), our experiment did not even yield a confident IC₂₀ value, indicating that the cell line could be insensitive to the drug perturbation. We thus believe that within the context of our experimental design, IC₂₀ and IC₅₀ drug sensitivities are difficult to relate to low dosage induced adaptive response.

How did the cell line specific factors identified (BUB1, ERBB2, LYN, PRKCZ) relate to response of each line to Osimertinib? Addition of the CRISPR study here would be of great interest to increase the relevance of these findings. Similarly, for ralimetinib, continuing these as specific examples to demonstrate biological relevance would be interesting.

Due to the limited throughput and suitability of cell lines models for the CRISPRko screening experiment (lines 671-688), our validation experiment only covered HCT-15 Linsitinib and Trametinib and NCI-H508 Trametinib perturbations. Validation of the Osimertinib-specific results identified by DeMAND was thus not possible with the current dataset.

However, we added a new validation of the DeMAND identified candidates using the CRISPRko experiment to our manuscript (lines 728-734, Supplemental Figs. 22-23):

“We further used the CRISPRko validation experiments to assess the VESPA-DeMAND-predicted resistance factors, as well as measured phosphoprotein abundances. While the VESPA-DeMAND-predicted resistance factors achieved almost similar performance to the results obtained only by VESPA (Supplemental Fig. 22-23), we found measured differential phosphoprotein abundance to not be predictive or correlate with the CRISPRko validation experiment, supporting the increased predictive power of VESPA inferred K/P-enzyme activities over phosphoprotein abundances (Supplemental Fig. 24-25).”

For figure 7b, is there an underlying reason behind the relatively poorer performance in NCI-H508 compared to HCT-15? Is this a function of the drug concentration chosen for the screen or model performance?

Indeed, we had to select lower drug concentrations for NCI-H508 (C1: 0.005 μ M, C2: 0.01 μ M) due to drug toxicity manifestation after the 96h time point, whereas we applied the FDR C_{\max} concentration of 0.036 μ M in the phosphoproteomic drug perturbation experiment. We believe that this discrepancy could explain the lower performance of this component of the validation experiment. We added a corresponding statement to the manuscript (lines 720-725):

“ROC was found to be particularly significant for HCT-15 perturbed by linsitinib and trametinib (AUC = 0.81, $p = 9e-04$; AUC = 0.74, $p = 7.8e-3$, respectively), yet lower significance for trametinib treated NCI-H508 cells (AUC = 0.67; $p = 0.0962$), potentially caused by the differences in drug concentrations in the two CRISRP-ko experiments (C1: 0.005 μ M, C2: 0.01 μ M) vs. the drug perturbation assays used to generate the phosphoproteomic profiles (C_{\max} : 0.036 μ M).”

Can the authors make a comment on the false negatives identified through CRISPR screens in supp figure 18-19.

While we believe that VESPA's low rate of false positive predictions support our claims regarding the improvements of our methods, we believe that the larger number of false negative predictions is caused by several confounding factors that could include a) experimental biases or discrepancies, such as the described bias towards tumor suppressors of CRISPRko screens (lines 694-712) , b) still incomplete coverage of all phosphoproteomic methods, including the one used by us, in comparison to CRISRPko screening, which covers all known kinases and phosphatases, or c) the involvement of false negative predicted proteins in adaptive response or survival mechanisms independent of the protein's primary kinase/phosphatase activity.

We believe that the last hypothesis is further supported by our new analysis, assessing predictability of differential phosphoprotein abundance (Supplemental. Fig. 24-25). Although many fewer TP candidates were identified, a similarly large fraction of FN candidates was confidently identified as being differentially abundant.

For example, MAP3K7 was found to have both lower VESPA activity and phosphoprotein abundance, while being an essential gene. This discrepancy could potentially be explained by the centrality of MAP3K7 as regulator of cell death, being involved in both NF- κ B and in NF- κ B-independent pathways such as oxidative stress and receptor-interacting protein kinase 1 (RIPK1) kinase activity-dependent pathways (PMID:25146924).

We have added a corresponding statement to the manuscript (lines 705-712).

In supp figure 18-19 a head-to-head comparison to VESPA activity and peptide intensity values for identifying resistance factors (vs CRISPR ground truth) would clarify the increase in predictability from using this model.

This is an excellent suggestion and we have added the corresponding analysis results to the manuscript (lines 728-732, Supplemental Figs. 24-25):

“We further used the CRISPR-ko validation experiments to assess the VESPA-DeMAND-predicted resistance factors, as well as measured phosphoprotein abundances. While the VESPA-DeMAND-predicted resistance factors achieved almost similar performance to the results obtained only by VESPA (Supplemental Fig. 22-23), we found measured differential phosphoprotein abundance to not be predictive or correlate with the

CRISPR-ko validation experiment, supporting the increased predictive power of VESPA inferred K/P-enzyme activities over phosphoprotein abundances (Supplemental Fig. 24-25).”

Figure 7, Supplementary Figure 18 and Supplementary Figure 19 are a bit confusing to understand. Would be helpful to show a volcano plot (e.g. logFC or Mageck beta scores drug vs control along the x-axis), and log-transformed p values along the y axis, with kinases predicted to be drivers of resistance marked out by a distinct colour.

We agree that the established visualization can be insightful as well. We have added color-coded volcano plots to Supplemental Figures 18-23.

REVIEWER COMMENTS

Reviewer #1 (Remarks to the Author):

The authors present an improved manuscript and the study has strong support from all reviewers, including this one. However, it is a pity that no direct evidence is presented showing that Vespa outputs are readouts of KP activities, as they are understood by people in the field of kinase signalling. The conclusion from the authors is that “drug perturbations we performed using drugs targeting these proteins did not fully abrogate PIK3CA, EGFR, and MAP2K1/2 activity, which was explicitly not the intention of our experimental design”. The authors explain that this is because low drug concentrations were used for this experiment. However, at the times tested (5 min, 15 and 60 minutes) “adaptive resistance mechanisms” are not yet activated and the activity of drug targets and pathway components should decrease by treatment. This should not be controversial because other phosphoproteomics methods have demonstrated this. Therefore, another interpretation of the results presented in the manuscript is that Vespa does not provide readouts of kinase activity. This does not mean that the information provided by the algorithm is not useful, as it seems that the readouts correlate with drug sensitivity, but the authors cannot claim to be measuring KP activities until this is rigorously demonstrated. It would be relatively trivial to carry out an experiment using one drug at a concentration known to completely inhibit the target (as determined the measurement of bona fide activity markers by WB or LC-MS/MS). Such experiment could then be analysed by Vespa to demonstrate that the readouts of the algorithm do indeed provide measures of KP activities. Alternatively, it is recommended that the authors do not claim that Vespa provide readouts of KP activities because this is not demonstrated in the paper.

Reviewer #2 (Remarks to the Author):

The authors have made substantial improvements in response to prior feedback. The manuscript remains intricate, which may pose a challenge for readers. However, this complexity may be inherent to the nature of the study, and I believe the authors have already tried their best to simplify the content in the Results section.

Reviewer #3 (Remarks to the Author):

The authors have put in considerable effort to respond to all the comments from the reviewers. They have successfully addressed some of my concerns, such as the issue of protein abundance as a confounding effect. However, I still have some doubts about the model's performance. The incorporation of additional datasets and the comparison with other methods is a good step forward but the conclusions are perplexing. The superiority of mVESPA over other methods is not evident. For instance, Supplementary Figure 5, which the authors use as a primary comparison to other methods, indicates that mVESPA either underperforms or is not suitable compared to other methods. Furthermore, the

manuscript's structure for review is quite poor, with figures lacking labels and numbers and legends being separated from both the figures and the text.

Reviewer #4 (Remarks to the Author):

The authors do a good job of addressing most points, with the exception of the extending their data to clinically relevant samples. I believe an opportunity has been missed, the data will be more impactful if the authors can extend their findings beyond a set of cell lines to more sophisticated models. Their response regarding the lack of significance of genetic mutations in colorectal cancer is inaccurate. Patients with colorectal cancer are routinely assessed based on their genetic mutation status (e.g. KRAS) and methylation status (MSI..), not transcriptional signatures. As of today, this is the gold standard for colorectal cancer classification in the clinical setting.

Network-based elucidation of colon cancer drug resistance by phosphoproteomic time-series analysis

Point-by-point response

George Rosenberger^{1,†}, Wenxue Li^{2,†}, Mikko Turunen^{1,†}, Jing He^{1,3,†}, Prem S Subramaniam¹, Sergey Pampou^{1,4}, Aaron T Griffin^{1,5}, Charles Karan^{1,4}, Patrick Kerwin¹, Diana Murray¹, Barry Honig^{1,6,7,8}, Yansheng Liu^{2,9}, Andrea Califano^{1,4,6,7,10,11}

- 1 Department of Systems Biology, Columbia University Irving Medical Center, New York, NY, USA
 - 2 Yale Cancer Biology Institute, Yale University, West Haven, CT, USA
 - 3 Present address: Regeneron Genetics Center, Tarrytown, NY, USA
 - 4 J.P. Sulzberger Columbia Genome Center, Columbia University Irving Medical Center, New York, NY, USA
 - 5 Medical Scientist Training Program, Columbia University Irving Medical Center, New York, NY, USA
 - 6 Department of Medicine, Columbia University Irving Medical Center, New York, NY, USA
 - 7 Department of Biochemistry & Molecular Biophysics, Columbia University Irving Medical Center, New York, NY, USA
 - 8 Zuckerman Mind Brain and Behavior Institute, Columbia University, New York, NY, USA
 - 9 Department of Pharmacology, Yale University School of Medicine, New Haven, CT, USA
 - 10 Herbert Irving Comprehensive Cancer Center, Columbia University Irving Medical Center, New York, NY, USA
 - 11 Department of Biomedical Informatics, Columbia University Irving Medical Center, New York, NY, USA
- † Equal Contribution

Correspondence to: yansheng.liu@yale.edu and ac2248@cumc.columbia.edu

Reviewer Comments

Reviewer #1 (Remarks to the Author):

The authors present an improved manuscript and the study has strong support from all reviewers, including this one.

We thank the reviewer for the positive feedback.

However, it is a pity that no direct evidence is presented showing that Vespa outputs are readouts of KP activities, as they are understood by people in the field of kinase signalling. The conclusion from the authors is that “drug perturbations we performed using drugs targeting these proteins did not fully abrogate PIK3CA, EGFR, and MAP2K1/2 activity, which was explicitly not the intention of our experimental design”. The authors explain that this is because low drug concentrations were used for this experiment. However, at the times tested (5 min, 15 and 60 minutes) “adaptive resistance mechanisms” are not yet activated and the activity of drug targets and pathway components should decrease by treatment.

We believe that the mechanisms of action of kinase inhibitors are frequently more complex than explored by typical pharmacological assays and, in reality, strongly depend on (a) cellular context and state, (b) drug specificity (c) drug concentration and (d) time-dependent cell adaptation mechanisms. Systematic chemo-proteomic studies (Klaeger *et al.*, Science, 2017; Zecha *et al.*, Science, 2023) provide strong support that kinase inhibitors can have very variable effects on kinases and their signaling pathways under different conditions. However, results based on binding affinity assays fail to report on the complex interplay between multiple high and low affinity targets and, more importantly, on cell adaptation mechanisms that can buffer the inhibitory effects of the drug on the target kinase activity. This is important because VESPA reports on a drug-targeted kinase activity based on the phospho-state of its primary substrates and not on whether the drug is binding the kinase. Given the highly pleiotropic regulation of phospho-target regulation, it is quite possible that the targets of an inhibited kinase may bounce back due to activation of alternative pathways. Indeed, while our data shows clear inhibition effects for some combinations of drugs and cell lines at the early time points—e.g., Fig. 5 Osimertinib (all cell lines), Ralimetinib (LS1034, SNU-61)—our data also suggest that this cannot be reasonably expected for all conditions and time points, especially due to the cell adaptive responses that clearly emerge from time-dependent profiles. This is also especially relevant given the relatively low drug concentrations used in our experiment (see also further data below).

This should not be controversial because other phosphoproteomics methods have demonstrated this. Therefore, another interpretation of the results presented in the manuscript is that Vespa does not provide readouts of kinase activity. This does not mean that the information provided by the algorithm is not useful, as it seems that the readouts correlate with drug sensitivity, but the authors cannot claim to be measuring KP activities until this is rigorously demonstrated.

We agree with the reviewer. VESPA uses the phospho-state K/P-enzyme substrates to assess its activity. Thus, if alternative compensatory pathways change the phospho-state of these substrates, due to cell adaptation mechanisms, for instance, the readout may incorrectly reflect the kinase activity. However, this is an intrinsic limitation of all conceptually related approaches, and not a VESPA-specific limitation. Further, as suggested by the reviewer, we believe that this represents an important readout because, ultimately, kinase activity is reflected in the phospho-state of their substrates and if these are rebalanced in the cell it is, for all practical purposes,

equivalent to differentially regulated kinase activity. We also note that, currently, there are no universally accepted approaches to benchmark kinase activity inference tools. Experimental validation using selected kinase inhibitors is typically extremely biased towards specific and well-known compounds and their targets, leading to poor generalization performance to larger signaling systems (Crowl *et al.*, Nat Commun, 2022).

We believe the methodologies developed for InKA (Beekhof *et al.*, Mol Syst Biol., 2019) and KSTAR (Crowl *et al.*, Nat Commun, 2022) represent a substantial advance for the field, because although they still are hampered by intrinsic biases and limited data quality and quantity, they are systematic and account for much more diverse sets of drugs than are usually covered. In our study, we not only extend and use the InKA and KSTAR benchmark to demonstrate the advantages and unique properties of VESPA compared to the state-of-the-art, but we also experimentally validate VESPA's predictions by orthogonal, kinome/phosphatome-wide CRISPRko assays. We are thus convinced that VESPA's metrics closely represent inferred kinase activities with state-of-the-art performance.

It would be relatively trivial to carry out an experiment using one drug at a concentration known to completely inhibit the target (as determined the measurement of bona fide activity markers by WB or LC-MS/MS). Such experiment could then be analysed by Vespa to demonstrate that the readouts of the algorithm do indeed provide measures of KP activities. Alternatively, it is recommended that the authors do not claim that Vespa provide readouts of KP activities because this is not demonstrated in the paper.

We fully agree with the reviewer, that given appropriate experimental conditions, VESPA should effectively report on drug-mediated kinase inhibition. For this purpose, we analyzed the recently published decryptM dataset (Zecha *et al.*, Science, 2023), where A431 epidermoid carcinoma cells (dependent on EGFR expression) were perturbed by Afatinib (targeting EGFR), Gefitinib (targeting EGFR), and Dasatinib (targeting SRC- and EPH-family proteins) with 10 different drug concentrations (Supplemental Fig. 5). Since epidermoid carcinoma is not included in CPTAC, we used a dVESPA-generated signaling network based on the CPTAC Lung Squamous Cell Carcinoma (LSCC) Discovery Study (Satpathy *et al.*, Cell, 2021), with the potential caveat that our networks might not be fully representative of A431 cell lines. We then used the full VESPA approach to infer kinase activities for all covered KP-enzymes and focused interpretation on the known targets as listed by DrugBank. We considered a VESPA NES (z-score) of NES < -1.65 (p -value < 0.05) to be the threshold for significant inhibition.

Our analysis shows significant inhibition of EGFR for both Afatinib and Gefitinib treatments with median z-scores of -3.49 (p -value = 0.0002) and -2.03 (p -value = 0.02), respectively (Supplemental Fig. 5). ERBB2 was also significantly inhibited by Afatinib, resulting in a median z-score of -2.24 (p -value = 0.01). Interestingly, only concentrations equal to or higher than 1M induced significant inhibition of the primary targets. For Dasatinib, 11 out of 15 covered DrugBank targets showed negative activity, with only MAPK14 being significantly inhibited (z-score = -2.09; p -value = 0.02). Using orthogonal assays (kinobeads), the original authors of the decryptM study observed a wider distribution of drug-target affinities for Dasatinib than for Afatinib and Gefitinib, supporting the notion that not all known drug targets might be effectively inhibited in all cellular contexts.

Reviewer #2 (Remarks to the Author):

The authors have made substantial improvements in response to prior feedback. The manuscript remains intricate, which may pose a challenge for readers. However, this complexity may be inherent to the nature of the study, and I believe the authors have already tried their best to simplify the content in the Results section.

We thank the reviewer for the positive feedback.

Reviewer #3 (Remarks to the Author):

The authors have put in considerable effort to respond to all the comments from the reviewers. They have successfully addressed some of my concerns, such as the issue of protein abundance as a confounding effect.

We thank the reviewer for the positive feedback.

However, I still have some doubts about the model's performance. The incorporation of additional datasets and the comparison with other methods is a good step forward but the conclusions are perplexing. The superiority of mVESPA over other methods is not evident. For instance, Supplementary Figure 5, which the authors use as a primary comparison to other methods, indicates that mVESPA either underperforms or is not suitable compared to other methods.

We apologize for any confusion in the presentation. VESPA achieves its performance by combining context-specific networks reconstruction (dVESPA) with K/P-enzyme activity inference (mVESPA). In this regard, our algorithm works very differently than other options and cannot be fairly benchmarked, especially for datasets where the dVESPA network reconstruction—which we have shown to provide a major improvement compared to the use of static, non-context-specific networks—cannot be performed. Specifically, dVESPA requires ~100 independent samples to reconstruct a network, whereas mVESPA can be applied to individual phosphoproteomic samples. As such, when a dVESPA network cannot be generated, it is unfair to compare to algorithms that rely on static, context-free reference networks, operate on absolute protein abundance estimates (InKA), or can only assess inhibited or activated K/P-enzymes in a single step (KSTAR). Despite these limitations, we have shown that mVESPA performs as well and generally better than the other algorithms even when a context-specific network is not available.

Our initial dVESPA/mVESPA benchmark requires differential comparisons between sensitive and insensitive cell lines and activity inference for both up- and down-regulated K/P-enzymes. To our knowledge, this functionality is not supported in many competing algorithms, which would require extensions or workarounds that by themselves might make fair comparison difficult. For example, a core feature of InKA is the incorporation of sites activated by tyrosine kinase autophosphorylation, a property that can impossibly be supported by our data types. KSTAR, on the other hand, requires prior definition of a threshold to identify directed (up- or down-regulated) differentially regulated phospho-sites, which makes KSTAR only suitable for analyses, where the effect direction and magnitude is well understood. For this reason, we compared dVESPA/mVESPA against a context-free reference network implementation (mVESPA/PC), which we believe is representative of prior approaches (Dugourd *et al.*, Mol Syst Biol, 2021). Our analysis demonstrates significantly better performance when using dVESPA networks (Fig. 2d).

The comparison against state-of-the-art algorithms using the KSTAR benchmark is much more biased against VESPA, since only parts of the mVESPA algorithm can be assessed using context-free reference networks. The reason is that the underlying data encompasses studies consisting of very few samples, representing very different sample types, for which no reference networks are available, and/or are not very quantitative. Nevertheless, we show that even such a limited incarnation of the algorithm outperforms all other algorithms in predicting primary serine/threonine drug targets (Supplemental Fig. 5a of last revision). Since tyrosine kinases are not represented in mVESPA/Johnson, benchmarking is not possible, however, mVESPA/PSP achieves the second-best result after KSTAR, which we believe also illustrates the applicability of VESPA to such datasets, given the right data. This is a result that can be greatly improved when dVESPA networks can be generated.

We agree with the reviewer though that showing the full results of the KSTAR benchmark is too complex for our purpose. Although the authors of the original KSTAR publication described all assumptions and methods in detail, reading their paper to understand our results goes beyond what can be expected by the readers of our paper. We have thus decided to remove the full KSTAR benchmark data and to only show the aggregated findings as part of Fig. 2e. We have replaced Supplemental Fig. 5 with the analysis of the decryptM dataset and modified the manuscript accordingly (Zecha *et al.*, Science, 2023).

Furthermore, the manuscript's structure for review is quite poor, with figures lacking labels and numbers and legends being separated from both the figures and the text.

We apologize for this issue. We have now submitted a combined PDF where all figures and legends are integrated.

Reviewer #4 (Remarks to the Author):

The authors do a good job of addressing most points, with the exception of the extending their data to clinically relevant samples. I believe an opportunity has been missed, the data will be more impactful if the authors can extend their findings beyond a set of cell lines to more sophisticated models. Their response regarding the lack of significance of genetic mutations in colorectal cancer is inaccurate. Patients with colorectal cancer are routinely assessed based on their genetic mutation status (e.g. KRAS) and methylation status (MSI..), not transcriptional signatures. As of today, this is the gold standard for colorectal cancer classification in the clinical setting.

In collaboration with a different research group, we have been working on extending VESPA to hepatocellular carcinoma (HCC) in the context of clinical tissue samples obtained from biopsies and corresponding organoid models. The efforts required to acquire the corresponding phosphoproteomic datasets has dramatically surpassed our already considerable efforts to acquire the present cell line data. Further, more than one reviewer has already complained that the paper scope is too large. As a result, we believe that a first manuscript focusing on the methodology will foster follow-up studies, where VESPA can be applied to clinically relevant data. Please note that the same approach was demonstrated for the VIPER algorithm. It was first introduced as a pure methodology in (Alvarez *et al.* 2016 Nat Genet) and then used in a very large number of translational manuscripts where clinically relevant cohorts were studied, such as (Alvarez *et al.* 2018 Nat Genet; Zeleke *et al.* 2023 Nat Cancer; Obradovic *et al.* 2021 Cell; Obradovic *et al.* 2023 Cancer Cell; Vasciaveo *et al.* 2023 Cancer Discovery; Mundi *et al.* 2023

Cancer Discovery). We believe that VESPA can have a similar impact on dissecting the role of signaling proteins in clinically relevant cohorts as well as their response to drug perturbations.

REVIEWERS' COMMENTS

Reviewer #1 (Remarks to the Author):

The revised manuscript shows new results but the evidence that Vespa provides readouts of K/P activities is not strong because such proof is limited to just two kinases (according to Suppl. Figure 5). While it is correct that “alternative compensatory pathways change the phospho-state of these Substrates”, this is not seen at short time points, as demonstrated by numerous phosphoproteomic studies, where kinase activities can be seen to robustly decrease in cells treated with the respective kinase inhibitor.

Reviewer #2 (Remarks to the Author):

The new results in Supplementary Figure 5 are supportive. However, I believe the drug concentration unit labeled in the figure (M) is wrong.

Reviewer #3 (Remarks to the Author):

The authors have clarified my remaining concerns. The presentation of the methods and the different method modalities (dVESPA/mVESPA) is still dense and confusing at times. A concrete application of the approach (e.g. clinically relevant samples as proposed by another reviewer) showing an advantage over other methods would have strengthened the manuscript.

Reviewer #3 (Remarks on code availability):

The method is implemented and provided within a well-established software platform.

Network-based elucidation of colon cancer drug resistance by phosphoproteomic time-series analysis

Point-by-point response

George Rosenberger^{1,†}, Wenxue Li^{2,†}, Mikko Turunen^{1,†}, Jing He^{1,3,†}, Prem S Subramaniam¹, Sergey Pampou^{1,4}, Aaron T Griffin^{1,5}, Charles Karan^{1,4}, Patrick Kerwin¹, Diana Murray¹, Barry Honig^{1,6,7,8}, Yansheng Liu^{2,9}, Andrea Califano^{1,4,6,7,10,11}

- 1 Department of Systems Biology, Columbia University Irving Medical Center, New York, NY, USA
 - 2 Yale Cancer Biology Institute, Yale University, West Haven, CT, USA
 - 3 Present address: Regeneron Genetics Center, Tarrytown, NY, USA
 - 4 J.P. Sulzberger Columbia Genome Center, Columbia University Irving Medical Center, New York, NY, USA
 - 5 Medical Scientist Training Program, Columbia University Irving Medical Center, New York, NY, USA
 - 6 Department of Medicine, Columbia University Irving Medical Center, New York, NY, USA
 - 7 Department of Biochemistry & Molecular Biophysics, Columbia University Irving Medical Center, New York, NY, USA
 - 8 Zuckerman Mind Brain and Behavior Institute, Columbia University, New York, NY, USA
 - 9 Department of Pharmacology, Yale University School of Medicine, New Haven, CT, USA
 - 10 Herbert Irving Comprehensive Cancer Center, Columbia University Irving Medical Center, New York, NY, USA
 - 11 Department of Biomedical Informatics, Columbia University Irving Medical Center, New York, NY, USA
- † Equal Contribution

Correspondence to: yansheng.liu@yale.edu and ac2248@cumc.columbia.edu

Reviewer Comments

Reviewer #1 (Remarks to the Author):

The revised manuscript shows new results but the evidence that Vespa provides readouts of K/P activities is not strong because such proof is limited to just two kinases (according to Suppl. Figure 5). While it is correct that “alternative compensatory pathways change the phospho-state of these Substrates”, this is not seen at short time points, as demonstrated by numerous phosphoproteomic studies, where kinase activities can be seen to robustly decrease in cells treated with the respective kinase inhibitor.

We believe that there might be confusion about the data depicted in Supplemental Fig. 5. These profiles depict kinase activate over drug concentration instead of perturbation time. We have selected this public dataset specifically to address the previous request for the demonstration of VESPA kinase activity inference using fully inhibiting drug concentrations:

“It would be relatively trivial to carry out an experiment using one drug at a concentration known to completely inhibit the target (as determined the measurement of bona fide activity markers by WB or LC-MS/MS). Such experiment could then be analysed by Vespa to demonstrate that the readouts of the algorithm do indeed provide measures of KP activities.”

Within this context, we believe our analysis demonstrates VESPA’s capability for identification of kinase activity inhibition.

Reviewer #2 (Remarks to the Author):

The new results in Supplementary Figure 5 are supportive. However, I believe the drug concentration unit labeled in the figure (M) is wrong.

We thank the reviewer for the positive feedback and for spotting this mistake. We have corrected the wrong concentration unit label to “nM”.

Reviewer #3 (Remarks to the Author):

The authors have clarified my remaining concerns. The presentation of the methods and the different method modalities (dVESPA/mVESPA) is still dense and confusing at times. A concrete application of the approach (e.g. clinically relevant samples as proposed by another reviewer) showing an advantage over other methods would have strengthened the manuscript.

We thank the reviewer for the positive feedback. As we have discussed in our previous rebuttal letter, we are convinced that an additional application to clinical sample is beyond the scope of this paper.